



# Tracking geothermal anomalies along a crustal fault using (U-Th)/He apatite thermochronology and REE analyses, the example of the Têt fault (Pyrenees, France)

Gaétan MILESI[1], Patrick MONIÉ[1], Philippe MÜNCH[1], Roger SOLIVA[1], Audrey TAILLEFER[1], Olivier BRUGUIER[1], Mathieu BELLANGER[2], Michaël BONNO[1], Céline MARTIN[1]

[1]Géosciences Montpellier, Université de Montpellier, CNRS, Université des Antilles, Campus Triolet, CC060, Place Eugène Bataillon, 34095 Montpellier Cedex05 France

[2]TLS Geothermics, 92 chemin de Gabardie, 31200 Toulouse France

*Correspondence to*: Gaétan Milesi (gaetan.milesi@umontpellier.fr)

**Abstract.** The Têt fault is a crustal scale major fault in the eastern Pyrenees along which 29 hot springs emerge mainly within the footwall damage zone of the fault. In this study, (U-Th)/He apatite (AHe) thermochronology is used in combination with REE analyses to investigate the imprint of hydrothermal activity nearby two main hot spring clusters and in between in an attempt to better define the geometry and intensity of the recent thermal anomalies along the fault and to compare them with previous results from numerical modelling. This study displays 99 new AHe ages and 63 REE analyses on samples collected in the hanging wall (18 to 43 Ma) and footwall (8 to 26 Ma) of the Têt fault. In the footwall, the results reveal AHe age resetting and apatite REE depletion due to hydrothermal circulation along the Têt fault damage zone, nearby the actual hot springs (Thuès-les-Bains and St-Thomas) but also in areas lacking actual geothermal surface manifestation. These age resetting and element depletions are more pronounced around Thuès-les-Bains hot spring cluster and are spatially restricted to a limited volume of the damage zone. Outside this damage zone, the modelling of thermochronological data in the footwall suggests the succession of two main phases of exhumation, between 30 and 24 Ma and a second one around 10 Ma. In the hanging wall, few evidences of hydrothermal imprint on AHe ages and REE signatures have been found and thermal modelling records a single exhumation phase at 35-30 Ma. Low-temperature thermochronology combined with REE analyses allows to identify the spatial distribution of a recent geothermal perturbation related to hydrothermal flow along a master fault zone in the eastern Pyrenees, opens new perspectives for the exploration of geothermal fields and provides at the regional scale new constraints on the tectonic uplift of the footwall and hanging wall massifs.

## 1 Introduction

World geothermal energy production is expected to grow rapidly over the next years due to its low environmental impact and its increasing technologic developments (Van der Zwaan and Della Longa, 2019). Geothermal exploration reveals a great potential in different geological contexts (Lund et al., 2011, Dobson, 2012) recognized for several years (Barbier, 2002),



which includes magmatic environments (e.g. Heffington et al., 1977) and fault systems due to crustal extension (Meixner et al., 2016). The heat production requires the presence of water such as provided by continental hydrothermal systems (Deming, 1994). Such a type of system is already exploited for the production of geothermal energy in the Basin and Range province (Blackwell et al., 2000; Faulds et al., 2010) or in western Anatolia Turkey (Roche et al., 2018), with temperatures above 200°C (Bertani, 2012).

Several studies have shown the impact of faults adjacent to topography to control hydrothermal circulation, where the hot water flow is enhanced by forced convection (e.g. Forster and Smith, 1989; McKenna and Blackwell, 2004; Taillefer et al., 2017; Sutherland et al., 2017; Volpi et al., 2017; Jordan et al., 2018; Wanner et al., 2019). Even in poorly active tectonic context, high topography gradient and fracture permeability inherent to normal faults provide good conditions to allow efficient hydrothermal flow, as shown by multiple hot springs located along faults (Taillefer et al., 2017, 2018; Wanner et al., 2019). In
this case, meteoric fluids infiltrate from high reliefs in the footwall, heat up at depth and raise in areas of high permeability around the faults (Grasby and Hutcheon, 2001; Craw et al., 2013). Thus, the circulation of hydrothermal fluids is essentially controlled by topography, permeability of the host rocks and the fault zone (Caine and Tomusiak, 2003; Bense et al., 2013), fracture mineralization (Eichhubl et al., 2009; Griffiths et al., 2016) and tectonic background of the area (Faulds and Hinz, 2015).

In places where heat flow data are not available (no bore hole), geothermal exploration mainly relies on surface manifestations of hot hydrothermal fluids (hot springs, tuffa deposits, recent hydrothermal mineralisation, …). It has however recently been proposed that Low Temperature (LT) thermochronology can be used as a low-cost tool to support or extend the exploration of such geothermal systems (Gorynski et al., 2014; Milesi et al., 2019). Actually, alongside these thoughts, the past decades revealed an increasing amount of studies about the influence of hydrothermal circulations on LT
thermochronometers, rendering difficult their interpretation in terms of exhumation history along faults (e.g. Whipp and Ehlers, 2007; Wölfler et al., 2010; Gorynski et al., 2014; Danišík et al., 2015; Valla et al., 2016; Louis et al., 2019). In this study, we propose an extended analysis of (U-Th)/He on apatite (AHe) thermochronometer (a LT thermochronometer sensitive for a range of temperature between 30 and 90 °C, Gautheron et al., 2009; Shuster and Farley, 2009), in association to Rare Earth Elements (REE) analysis of dated apatites, to track the effects of surface and hidden thermal systems along the Têt normal
fault (Pyrenees). We showed in a preliminary work (Milesi et al., 2019) that the analysis of REE mobility can help to interpret AHe ages scattering and thus evidence hydrothermal imprints on measured ages. In this previous work, we focused on an area located in the vicinity of an active hot spring cluster along the dormant Têt fault in the eastern Pyrenees. The present work investigates a 10 km-long segment of the Têt fault in order to test this tool both in areas lacking current surface manifestations of hot hydrothermal fluids and nearby two hot spring clusters along the fault with hot water temperature above 60°C.





## 2 Geological setting

### 2.1 Cenozoic Têt fault activity and related exhumation of the eastern Pyrenees

The Pyrenees form a mountain range 1000 km long and 150 km wide, at the boundary between the Eurasian and Iberian plates (Fig. 1A). The maximum of shortening occurred during the Eocene in the main part of Pyrenees (Vergès et al., 1995), and between Eocene and early Oligocene in the eastern part (Burbank et al., 1992) with the emplacement of south verging nappes rooted in the northern part of the range (Vergès et al., 1995; Sibuet et al., 2004; Teixell et al., 2016). At the end of Oligocene, faults with a NE-SW trend such as the Têt and Tech faults are (re-) activated with normal motion in response to the rifting of the Gulf of Lion (Séranne et al., 1995). Later, ) two minor tectonic events were recorded on these faults: a Miocene dextral strike-slip phase along the Têt fault leads to the formation of pull-apart basins (Fig. 1B) such as Cerdagne and Roussillon basins (Cabrera et al., 1988; Gibert et al., 2007) and a second extensional tectonic event during the Plio-Quaternary (Carozza and Baize, 2004; Lacan and Ortuno, 2012; Petit and Mouthereau, 2012; Clauzon et al., 2015).

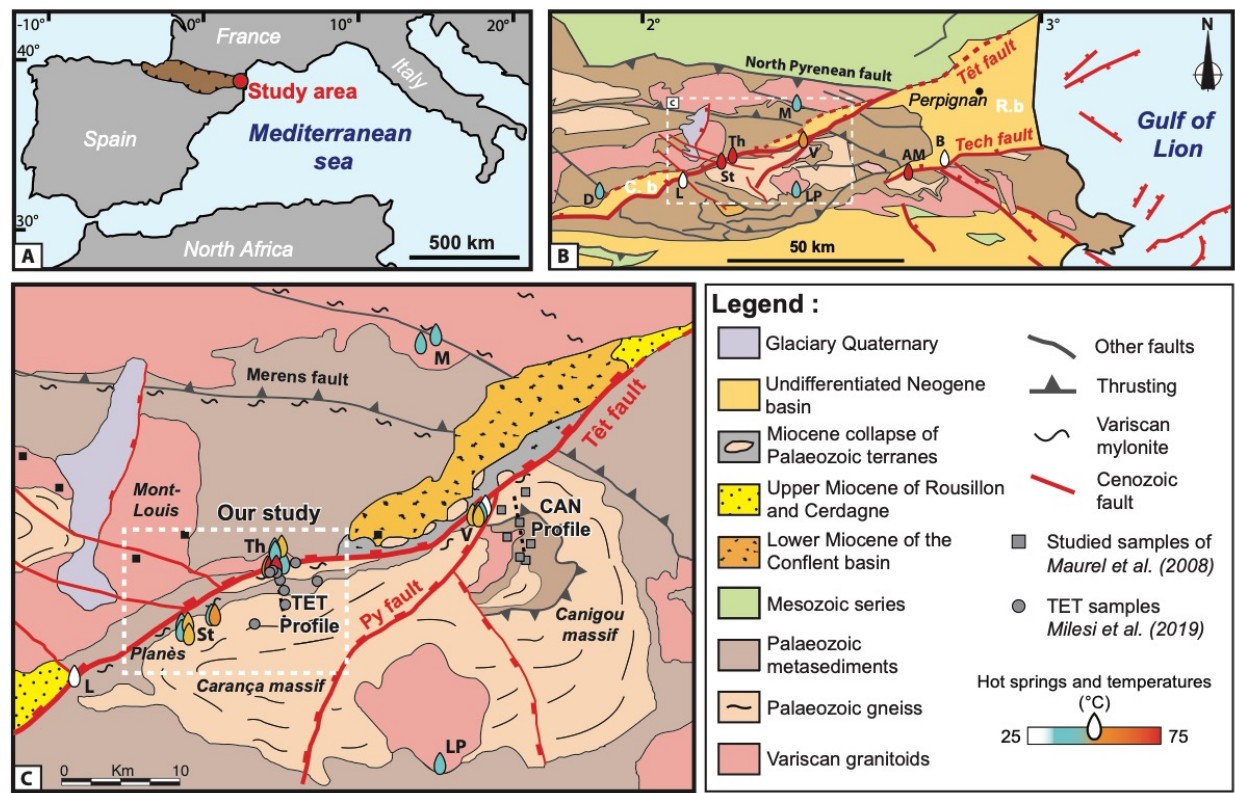

**Figure 1. A) Regional map at the scale of the Mediterranean basin: the Pyrenees are reported in brown and the study area is indicated with a red dot. B) Structural map with principal faults and units, hot spring clusters are reported with coloured drops (D: Dorres, L: Llo, St: St-Thomas, Th: Thuès-les-bains, V: Vernet-les-bains, LP: La Preste-les-bains, AM: Amélie-les-bains, B: Le Boulou, M: Molitg-les-bains ). Roussillon Basin: RB, Cerdagne Basin: CB (modified from Taillefer et al., 2017). C) Local map with AHe samples previously published and location of the studied area, hot springs are also represented by a drop (modified from Maurel et al., 2008).**



The Têt fault (eastern Pyrenees) is a 100 km long accident that runs across Palaezoic rocks from the Axial Zone and that separates two domains with different long-time thermal evolution and exhumation history (Fig. 1B). Previous studies, at the scale of the eastern Pyrenees, using various thermochronometers such as muscovite and biotite $^{40}$Ar/$^{39}$Ar, zircon fission track (ZFt), (U-Th)/He on zircon (ZHe), apatite fission track (AFT) and (U-Th)/He on apatite (AHe) provide an important age database to constrain the regional thermal history of eastern Pyrenees (Maurel et al., 2008; Gunnell et al., 2009; Milesi et al., 2019).

In the hanging wall of the Têt fault, $^{40}$Ar/$^{39}$Ar ages of hornblende (300.3 ± 3.1) and biotite (291.2 ± 2.8) from the granitic Mont-Louis massif record only the Variscan cooling history. By contrast K-feldspar displays a discordant age spectrum increasing from about 50 Ma to 295 Ma which was interpreted through multiple diffusion domain (MDD) modelling to record the exhumation and cooling effects related to the pyrenean compressive tectonics in the Eocene. This is consistent with the fission tracks and zircon (U-Th)/He ages in the whole massif. Apatites yielded a large range of AHe ages from about 50 to 30 Ma that are indicative of a long residence in the partial retention zone from the middle Eocene to the lower Oligocene (Maurel et al., 2008; Gunnell et al., 2009).

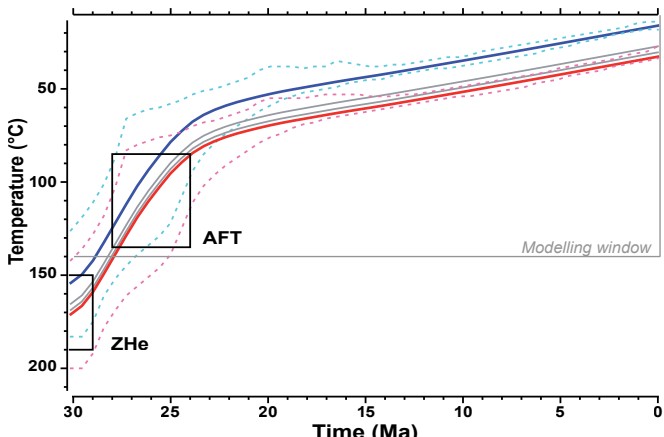

**Figure 2. Regional thermal history of the Carança Canigou massifs, in the Têt fault footwall (grey line). Thermal modelling was computed with QTQt software (Gallagher, 2012) and Gautheron et al. (2009) diffusion parameters using AHe from Milesi et al. (2019) and AFT and ZHe data from Maurel et al. (2008). Blue and red T-t paths correspond to the computed thermal history of the most and least elevated samples, respectively. Dashed lines: 95% confidence interval of T-t paths.**

In the footwall of the Têt fault, samples from the Canigou massif display younger and discordant biotite and muscovite $^{40}$Ar/$^{39}$Ar ages that range from about 90 to 280 Ma and were interpreted to record a thermal event contemporary to the thinning and associated metamorphism of the North Pyrenean zone (Maurel et al., 2008) and/or the thermal effects associated with the widespread hydrothermal activity that affected the Eastern Pyrenees in the Cretaceous (Boulvais et al. 2007; Poujol et al. 2010; Fallourd et al. 2014; Boutin et al. 2016). The Canigou massif could then have occupied a deeper position with respect to the Mont-Louis massif, with a differential estimated at ~ 4 km (Maurel et al., 2008). Low temperature thermochronometers revealed that the Canigou massif was mainly exhumed, and rapidly cooled, between ~30–20 Ma in relation with the Têt fault normal activity (Fig. 2) (Maurel et al., 2008; Milesi et al., 2019). The Cenozoic exhumation of the Carança and Canigou



massifs is estimated at about 2000 m from a large consensus but last period of relief edification and Têt fault activity is still debated. The last vertical movements are related either to the Miocene tectonic phase with a maximum of 300 m of footwall uplift (Cabrera et al., 1988; Mauffret et al., 2001) or to the Plio-Quaternary tectonic phase with up to 500 m of exhumation

(Carozza and Baize, 2004; Lacan and Ortuno, 2012; Clauzon et al., 2015). It must be stressed that some authors consider that any vertical displacement occurred during the Plio-Quaternary (Petit and Mouthereau, 2012). Numerous NW-SE faults, inherited from the Hercynian tectonics (Barbey et al., 2001), are described as brittle faults with post Miocene activity. These faults are well expressed near the St-Thomas hot spring cluster (Fig. 3, Taillefer et al., 2017 and 2018). In the late to middle Neogene, a global regional uplift of ~1.5 to 2 km associated to thermal erosion of the subcrustal lithosphere is evidenced on

the basis of a multi-disciplinary study (Gunnell et al., 2009) and thermochronological study (Fitzgerald et al., 1999).

      Today, the Têt fault is quiescent without clear evidence of seismic activity (Souriau and Pauchet, 1998) and no vertical movement is recorded by GPS in the eastern Pyrenees (Masson et al., 2019). Cosmogenic radionuclide data on karst sediments (Sartégou et al., 2018) and on Têt valley fluvial terraces (Delmas et al., 2018) suggest low incision rates in a range of 1 and 25 m/m.y. since 6 Ma.

**2.2 Têt fault damage zone and associated hydrothermal systems**

      The Têt fault is characterized by a thick sequence of deformed rocks on both sides of the main fault that can be divided in two distinct zones, the Core Zone (CZ) and the Damage Zone (DZ) (Caine et al., 1996; Shipton and Cowie, 2001; Billi et al., 2003; Kim et al., 2004; Berg and Skar, 2005). The CZ is generally characterised by fault rocks composed of breccias, cataclasites and gouges (e.g. Sibson, 2000; Fossen and Rotevatn, 2016). The CZ of the Têt fault is ~10-m thick and essentially

composed of incohesive breccias supported by clay gouges, cataclasites and crush breccias formed at the expense of Palaeozoic rocks. Locally the Têt fault displays a multi-core pattern (Thuès-les-Bains, Llo) with poorly deformed gneiss lenses within the CZ (see Supplement Section S.1 for a description of Thuès-les-Bains CZ).

  The DZ is a rock volume adjacent to the CZ characterised by an important amount of fractures (Caine et al., 1996; Sibson, 2000; Kim et al., 2004; Agosta et al., 2007; Faulkner et al., 2010). Close to Thuès-les-Bains, on the basis of field measurements

of fractures and small faults in the Carança valley, the half-thickness of the DZ is estimated at 400 m with an inner highly fractured DZ of 75 m (Milesi et al., 2019). The half-thickness of the Têt fault DZ is estimated between 200 m and 700 m using lineament analysis of satellite SPOT images with 5 m resolution (Taillefer et al. 2017). These values are coherent with fault displacement / damage zone thickness laws assuming a symmetrical DZ for the fault (Mayolle et al., 2019). In the DZ, fractures are filled with quartz and/or carbonates and some of them from the inner DZ show chlorite, muscovite and iron oxides in veins

crosscutting fault rocks.

      The Têt valley concentrates 29 hot springs with temperatures between 25°C and 75°C, without seasonal temperature variation, and with relatively homogeneous chemical composition of spring waters (Petit et al., 2014; Supplement Section S.2). In the studied area, hot springs are distributed in four main clusters, from east to west (Fig. 1B): Vernet-les-bains (Vernet, Tw= 36-56°C), Thuès-les-bains (Thuès, Tw=35-73°C), St-Thomas-Prats-Balaguer (St-Thomas, Tw=38-60°C) and Llo (Tw around





30°C). Stable isotope analyses of hot spring waters indicate a meteoric origin, with altitude of infiltration above 2000 meters for Thuès and St-Thomas, above 1800 meters for Vernet and in between for Llo hot springs (Petit et al., 2014). Various geothermometers (silica, chalcedony, Na/K, Na/Li, Na-K-Ca) show a range of maximum water temperature at depth between 70°C and 130°C consistent with the surface temperature of emerging water of each cluster (Krimissa, 1992; Taillefer et al., 2018). Hot springs are mainly located in the footwall DZ of the Têt fault except for Thuès-les-Bains area where 3 hot springs

occur in the hanging wall too (Canaveilles, Tw=52-60°C). The location of hot springs mainly in the footwall may be related to the occurrence in surface of impermeable metasediments in the hanging wall compartment (Taillefer et al., 2018) or to the fault CZ which can act as a barrier of permeability for hydrothermal fluids (e.g. Fisher and Knipe, 1998; Ballas et al., 2012). The local presence of NW-SE faults (e.g. Prats fault near St Thomas) cross-cutting or linked to the Têt fault can also increase the permeability and localised channelized fluid upflows.

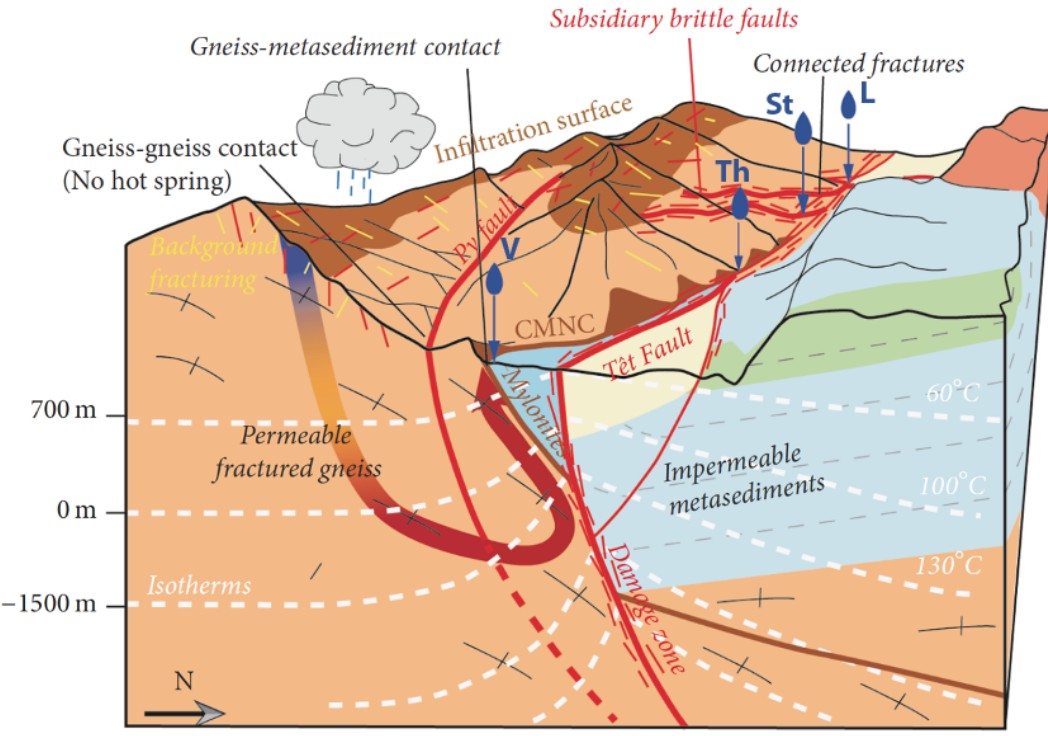

**Figure 3. Conceptual model of hydrothermal circulations in the footwall of the dormant Têt fault (hot spring cluster: L: Llo, St: St-Thomas, Th: Thuès-les-bains, V: Vernet-les-bains; after Taillefer et al., 2017).**

Figure 4 shows the result of a numerical modelling of fluid circulations along the Têt fault, between Thuès-les-Bains and Planès, that takes into account the combined effects of permeability, topography and structural discontinuities but do not consider a potential fluid contribution from the hanging wall (Taillefer et al., 2018). It shows the presence of a ~7 km-long

surface thermal anomaly along the Têt fault. In this model, the simulated geothermal gradient can reach up to 90°C/km around Thuès-les-Bains and three main zones of hot surface water temperatures have been identified. Two of them are located at the tips of this anomaly, on the hot springs clusters of Thuès-les-Bains and St Thomas, and the third is in the center of this anomaly,



located 2 km west of Thuès-les-Bains at the foot of the Gallinàs peak (2461 m), in a place where no hot spring is observed (Taillefer et al., 2018). In the latter location, the numerical model simulates elevated surface water temperature mainly in

relation with the topography gradient along and outside the fault from both the footwall and the hanging wall, which promotes deep infiltration of meteoric waters and resurgence in the valley, near the Têt fault. To the west, near the Planès plateau, abnormally low surface water temperatures have been modelled, suggesting that this area could be interpreted as a recharge zone with infiltration of meteoric fluids (Taillefer et al., 2018).

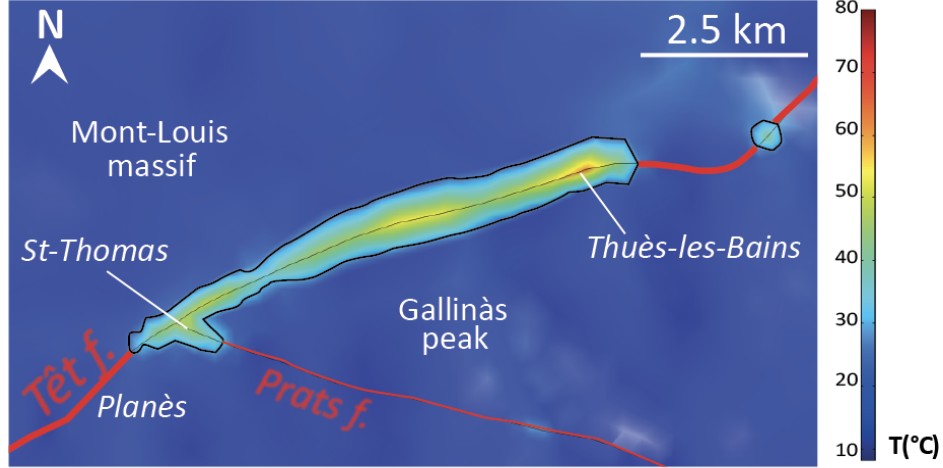

**Figure 4. Map view of the modelled water temperatures at the surface taking into account a permeability of 7.10-15 m2 and 5.10-15**
**m2 for the Têt fault zone and Prats fault zone, respectively (modified after Taillefer et al., 2018).**

Near Thuès, we recently showed that hydrothermal interactions account for AHe age dispersion (Milesi et al., 2019). Indeed, AHe ages are scattered and rejuvenated or aged in a consistent way with respect to the structural position of samples. In the inner DZ adjacent to the CZ, where fluid flows were concentrated, AHe ages display mainly an apatite ageing with a large scatter between $2.7 \pm 0.3$ Ma to $41.2 \pm 1.9$ Ma. Samples from the outer DZ mainly show apatite rejuvenation with less

dispersed ages between $0.9 \pm 0.1$ Ma to $21.1 \pm 1.1$ Ma. Moreover, we showed that the rejuvenation of apatites in the DZ correlates to a decrease of their REE contents, while their ageing is associated with enriched REE contents compared to samples taken outside the DZ (Milesi et al., 2019).

The present thermochronological study allows to extend to west the study area along the footwall Têt fault, up to the Planès recharge zone. This enables to explore the location of the recent thermal anomalies with LT thermochronology

combined to REE analyses along the fault in a different way than numerical modelling. We also study some samples from the hanging wall to compare with.



## 3 Methodology

### 3.1 Sampling

23 new samples of Paleozoic gneisses, Variscan mylonites and granites were collected both in the footwall and hanging wall of the Têt fault (Fig. 5) representing 99 single grain AHe ages and 63 REE analyses. To study the distribution of the thermal anomaly along the Têt fault, sampling was realized roughly perpendicular to the fault: (1) in the vicinity of Thuès-les-bains (TET profile) and St-Thomas (ST profile) hot spring clusters, (2) in areas without hot springs between St-Thomas and Thuès-les-bains, i.e. the Carança valley (CAR) and the foot of the Gallinàs peak (GAL profile). Farther west, in the Planès area (PLA) which is a topographic hight along the fault considered as a recharge zone. The aim of this sampling was to track

the effects of recent hydrothermal circulations, mainly in the footwall, adjacent to the fault on the behaviour of AHe ages and REE. Our final goal is to provide a first order map of recent thermal anomalies along the Têt fault and to compare it with that deduced from numerical modelling (Fig. 4). For all the samples, we selected the freshest rocks, up to the highly fractured inner DZ, in order to get the most suitable apatite grains for AHe dating.

### 3.1.1 Sampling nearby active hot spring clusters: TET and ST profiles

The TET profile is located in the vicinity of the Thuès-les-bains cluster showing the hottest water temperatures (Fig. 5). This profile has already been studied (Milesi et al., 2019) and we completed the sampling of this profile with two samples from the footwall inner DZ (TET1 and TET8) and one from the hanging wall at the transition between inner and outer DZ (TETHW). Samples TET1 and TET8 were taken within hydrothermalized, chloritized and fractured gneisses, within the hot spring cluster and 500 m to the east, respectively. Sample TETHW is a well-preserved, non-fractured augen gneiss collected

at a distance of 100 m from the Têt fault to evaluate the influence of a potential hydrothermal activity in this block.

ST samples are located on a ~3500 meter long profile crossing St-Thomas hot spring cluster. The hot spring emergences of St-Thomas are located in the vicinity of a NW-SE fault network connected to the Têt fault (Taillefer et al., 2017). The dimension of DZ in St-Thomas is larger than in Thuès-les-Bains due to the presence of this fault network with a width of 700 m. We collected 10 samples on this profile including gneisses, granites and mylonites (Figs. 6 A, B and C)

between an elevation of 1107 m (ST16) and 1480m (ST8). In the footwall DZ, samples ST15 and ST16 are highly fractured mylonites collected close to the fault at 15 m and 100 m (inner DZ), with evidences of chlorite and silica precipitation (Fig. 6, G, H and I). Samples ST2 and ST3 are located around St-Thomas hot springs (< 100 m) in the outer DZ at 250 m and 400 m from the Têt fault, respectively. Sample ST2 is a fractured and chloritized gneiss showing some traces of oxidation, while gneiss ST3 does not display any fracturation or alteration. Outside the footwall DZ, samples ST4 and ST8 were collected at a

distance of 835 m and 1750 m from the Têt fault. They look fresh and do not display neither any evidence of hydrothermal alteration nor fractures. In the hanging wall, samples ST1, ST11, ST12 and ST14 are collected respectively at 5 m, 160 m, 1900 m and 175 m from the Têt fault and correspond to an altitude variation of 500 m. Sample ST1 is a highly fractured gneiss from the inner DZ and shows an important density of silica veins. In the outer DZ, sample ST14 is also a gneiss showing few



fractures filled by chlorite and sample ST11 is an unaltered and non-fractured granite (Fig. 6B). ST12 corresponds to a well-
preserved and undeformed outcrop of the Mont-Louis granite.

**Figure 5. A) Stuctural map modified from Taillefer et al. (2017) showing the four sampling transects. In grey, samples from previous studies (Maurel et al., 2008; Milesi et al., 2019); in white, new samples of this study. B) Topographic cross sections of the four transects with the projection of the dated samples (vertical scale exaggerated two times). 1) Planès, 2) St-Thomas, 3) Gallinàs and 4)**





**Thuès-les-Bains. White samples are new samples of this study, grey samples from Thuès-les-Bains are from Milesi et al. (2019).**
**Projected samples are above topographic lines.**

### 3.1.2 Free hot spring areas: GAL and PLA profiles, CAR valley

The GAL profile has been centered on the thermal anomaly simulated in an area devoid of actual hot springs (*Taillefer et al., 2018*), approximately midway between the TET and ST profiles. Samples GAL3, GAL6 and GAL7 are fine-grained gneisses located on a steep slope of the Têt fault footwall, respectively at a distance of 920 m, 330 m and 200 m of the fault
core, with an elevation ranging between 1323 m and 1065 m. GAL3 sample is a well-preserved fine-grained gneiss outside the DZ of the Têt fault. In the outer DZ, samples GAL6 and GAL7 have the same lithology than GAL3 but show many quartz and calcite filled fractures and locally the occurrence of chlorite and iron oxides. Samples GAL1, GAL2 and GAL8 are gneisses from the hanging wall, at a distance of 425 m, 115 m, and 40 m from the Têt fault core (Fig. 5). Samples GAL1 and GAL2 from the outer DZ are weakly fractured whereas sample GAL8 from inner DZ is highly fractured and highly chloritized and
silicified.

We also collected samples from the footwall, 1.5 km to the west of the Thuès hot spring cluster near the Carança valley (Fig. 5), but only one sample (CAR7) from the outer DZ at 200 m south of the the Têt fault core yields apatites suitable for (U-Th)/He dating. It is a variscan mylonite that exhibits a well-preserved fine-grained assemblage with little evidence of retrogression (scarce growth of sericite after feldspar and of chlorite after biotite) and the presence of quartz-calcite filled
fractures at the scale of the outcrop (Figs. 6D and E).

On the PLA profile, the three dated samples are located on a small plateau above 1500 m of elevation and 1.5 km west of the St-Thomas hot spring cluster (Fig. 5). Gneisses PLA2 and PLA5 do not show any evidence of alteration, and sample PLA3 is fractured with locally the presence of iron oxides (Fig. 5). In the footwall, samples PLA3 and PLA2 are located at a distance of 170 m (within the outer DZ) and 920 m (outside DZ) from the principal Têt fault trace and at rather the same
elevation (1622 m and 1682 m). The sample PLA5 is located within the outer DZ of the hanging wall at a distance of 600 m from the Têt fault.



**Figure 6. Samples outside the Damage Zone (DZ): A) Typical Canigou augen gneiss; B) Thin section of undeformed Mont-Louis granite, ST12 sample; C) Variscan mylonite unaffected by fractures. Samples from the outer DZ: D) Silica-filled fractures in mylonites close to CAR7 sample; E) Thin section view of a silica-filled fracture; F) Fracture filled with a micro-breccia near St-Thomas (ST) hot spring cluster. Samples from the inner DZ : G) Intense fracturation in the footwall near the Têt fault on ST profile close to samples ST15 and ST16; H) Thin section of a heterogeneously fractured and altered gneiss (close to sample ST15) showing gneiss lenses less affected by hydrothermal alteration; I) Close view on apatite grains within a gneiss lens (enlargement of H); J) Thin section of cataclasite next to the Têt fault core; K) Thin section of proto-cataclasite next to the Têt fault core.**





## 3.2 Analytical method

### 3.2.1 (U-Th)/He analyses

Apatite (U-Th)/He analyses were conducted in the noble gas laboratory of Géosciences Montpellier. All samples were crushed, and apatite mineral concentration was operated with heavy liquids method. Inclusion free apatite crystals with less evidence of fractures were handpicked (Supplement Section S.3 for apatite grains photos) under a binocular microscope and

grains with equivalent radius (Rs) above 40 μm were selected (see Tables in section 4). Single grains were packaged in Pt tubes, placed under vacuum and heated at 900°C for 5 min with a 1090 nm fibre laser operating at 20W. After $^3$He spiking, gas purification was achieved with a cryogenic trap and two SAES AP-10-N getters and helium content was measured on a Quadrupole PrismaPlus QMG 220. The $^4$He content was determined by the peak height method and is 10–100 times above typical blank levels. Second heating with the same procedure was systematically operated to check that more than 99% of $^4$He

was extracted during first heating. After helium extraction, Pt tubes were retrieved from the mass spectrometer and put in a 2ml polypropylene conical tube. Samples were doubly spiked (230Th and 233U) and dissolved in 100μl 5N HNO3 (60°C for 2,5 hour). The resulting solutions were diluted with 900μl 1N HNO3 and U (233U and 238U) and Th (230Th and 232Th) were measured by using isotope dilution ICP-MS (for more details about the analytical procedure see Wu et al., 2016). For age calculation, alpha ejection correction (Farley et al., 1996) was calculated using the Ft software (Gautheron and Tassan-Got,

2010; Ketcham et al., 2011). Durango apatite replicates were analysed each four unknown grains and yielded a mean age of 31.31 ± 1.82 Ma during the course of this study. This is consistent with the reference age of 31.02 ± 1.01 Ma given by McDowell et al. (2005).

### 3.2.2 REE analyses

REE and trace elements (LILE, HFSE) analyses on single apatite grains were performed at Géosciences Montpellier

using the equipment available at the AETE-ISO analytical platform of the OSU OREME (University of Montpellier). Analyses were conducted using the same solutions that were previously used for U and Th isotopic analyses following a technique published in earlier reports (e.g. Bruguier et al., 2003). Analyses were performed using an Agilent 7700x quadrupole ICP-MS in the no-gas mode using the pulse counting mode (three points per peak) and a 50 μl/mn self-aspirating nebulizer. Concentrations were determined by external calibration using multi-element calibration solutions prepared from pure, 10 ppm,

single element solutions. Nb and Ta were measured by surrogate calibration using Zr and Hf respectively, following the method outlined by Jochum et al. (1990) for spark source mass spectrometry. Polyatomic interferences were reduced by optimizing the system to an oxide production level < 1.5% measured on Ce and corrections were applied using yields for MO+ and MOH+ determined during the same run by analysing batches of synthetic solutions containing interfering elements (Light REE: LREE and Ba) but free of interfered elements (Middle REE and Heavy REE: HREE). Accuracy was checked by repeated

measurements of aliquots of the Durango apatite which are available in Supplement Section S.4. The results are consistent





with previously published values by Chew et al. (2016). Due to a technical problem, some apatites have not been analysed for their REE (see Supplement Section S.4 for REE content table).

### 3.2.3 Thermal modelling

Inverse thermal history modelling was performed with QTQt software (Gallagher, 2012) in order to test if He thermal
diffusion models (Flowers et al., 2009; Gautheron et al., 2009) can reproduce measured AHe ages of a single sample. We have also modelised T-t paths using the new samples collected to complete the thermal history of Milesi et al. (2019) established along the TET profile footwall and to set up a new thermal model for the hanging wall.

## 4 Results

### 4.1 Footwall samples

AHe ages and REE patterns from apatite grains of the footwall of the Têt normal fault are summarized in Table 1 and Figures 7 and 8. No AHe ages - eU values relationship can be evidenced (Supplement Section S.5). All analysed apatite grains from samples either outside or inside the DZ show a typical wing-shaped REE pattern with a marked negative anomaly in Europium (Fig. 8) characteristic of apatites in S-type granites (Sha and Chappell, 1999), except those from fine-grained gneisses on the GAL profile where this anomaly is less pronounced. Outside the DZ, apatites GAL3 have consistent lower
REE contents and flatter shape patterns with a slight Europium negative anomaly very similar to that reported by Henrichs et al. (2018) for apatites retrieved from medium grade metapelites. These effects are attributed to growth of co-genetic epidote, which is the dominant carrier phase of the REE (with Y, Th and U) and which is observed in our samples in variable quantity, thus controlling the shape of the REE patterns of apatites and their REE contents at different scales.



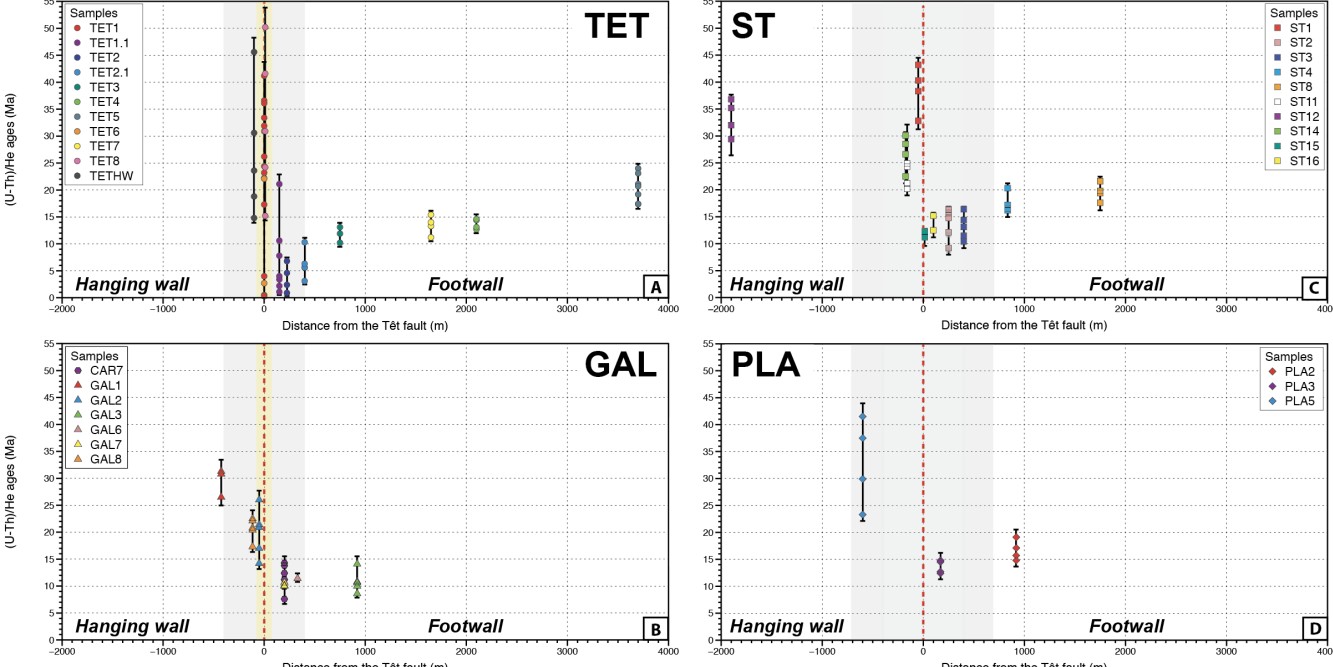

**Figure 7. AHe ages as a function of distance from the Têt fault. Grey area: the Damage Zone of the Têt fault. Inner DZ (yellow) is only distinguished for TET and GAL profiles according to Milesi et al. (2019).**

### 4.1.1 Nearby active hot spring clusters: TET and ST profiles

Along the TET profile, new apatite grains from the previously dated sample TET1 (Milesi et al., 2019) were dated. This sample is a highly fractured and chloritized gneiss from the inner DZ and within the Thuès-les-Bains hot spring cluster. Two apatite grains yield AHe ages of $0.5 \pm 0.2$ Ma and $0.3 \pm 0.1$ Ma that are younger than those previously published on the same sample, i.e. between $4.0 \pm 0.2$ Ma and $41.2 \pm 1.9$ Ma (Milesi et al., 2019). The AHe age dispersion is then very important in this sample. However, the eU value (U+0.235*Th) of newly analysed grains varies between 18.8 and 39.3 ppm and falls in the range of the previously dated grains. We also dated apatite grains from another gneiss from the inner DZ (TET8) that also shows a very high intra-sample AHe age dispersion, i.e. between $15.2 \pm 1.2$ Ma and $50.2 \pm 4.0$ Ma, with rather low eU values between 8.8 and 26.8 ppm.

Along the ST profile, apatites from highly fractured variscan mylonites within the inner DZ (ST15 and ST16) display AHe ages that range between $11.2 \pm 1.1$ Ma and $15.2 \pm 1.2$ Ma with low eU values between 8.2 and 26.4 ppm. However, the moderate intra-sample AHe age dispersion and the mean ages (Table 1) have to be considered with caution because of the low number of analysed grains in each sample: 2 grains in ST15 and 3 in ST16. Gneiss samples ST2 and ST3 from the outer DZ but close to the St-Thomas hot springs show more dispersed AHe ages, i.e. between $16.4 \pm 1.5$ Ma and $9.2 \pm 0.5$ Ma and between $16.4 \pm 1.3$ Ma and $10.4 \pm 0.8$ Ma, respectively and eU values comprised between 14.9 and 59.7 ppm. In ST2, two grains (ST2-5 and ST2-6, table 2) show anomalously high Th contents that could be indicative of the occurrence of optically undetected tiny Th-rich inclusions within these apatites. By contrast with co-existing apatites, these Th-rich apatites are also





enriched in light REE (Fig. 8) consistent with the presence of tiny monazite or allanite inclusions which are common light
REE-rich phases in granites (e.g. Förster, 1998). Therefore, these disturbed REE patterns are not taken into account in the
discussion and the corresponding AHe ages are discarded. A mean age of $15.8 \pm 0.8$ Ma is retained for sample ST2. Outside
the footwall DZ, samples ST4 and ST8, at a distance of 835 m and 1750 m from the Têt fault, respectively, show older mean
AHe ages of $17.3 \pm 0.8$ Ma and $19.6 \pm 0.8$ Ma with a limited intra-sample dispersion. eU values are higher for the leucogranite
ST4 (76.7 to 176.6 ppm) than for the gneiss ST8 (29.3 and 70.1 ppm).

### 4.1.2 Free hot spring areas: GAL and PLA profiles, CAR valley

On the GAL profile, samples GAL3 and GAL7 are fine grained gneisses respectively from the outside and outer Têt
fault DZ and both show a limited AHe age dispersion with mean ages of $10.8 \pm 0.9$ Ma and $10.3 \pm 0.2$ Ma. However, eU values
are higher and more variable in sample GAL7 (16.5 and 41.0 ppm) than in sample GAL3 (9.6 to 17.5 ppm). A single apatite
retrieved from a fractured gneiss (GAL6), next to the sample GAL7, displays an AHe age of $11.4 \pm 0.9$ Ma with an eU value
of 21 ppm. With the exception of one heavy REE enriched apatite in sample GAL7, apatite grains from the outer DZ and
outside have the same REE patterns (Fig. 8), thus indicating similar geochemical and paragenetic conditions of growth during
medium grade metamorphism of pelitic rocks (Henrichs et al., 2018).

In the Carança valley, sample CAR7 from the outer DZ displays relatively scattered AHe ages, i.e. between $7.6 \pm 0.4$
Ma and $14.3 \pm 0.5$ Ma, and scattered eU values between 6.0 and 49.5 ppm. Apatites from CAR7 display consistent REE
patterns but variable REE contents (Fig. 8). It is noteworthy that apatites with the lowest REE contents yield younger AHe
ages of 7.6 and 10.4 Ma compared to those with higher REE contents with ages between 11.2 and 14.3 Ma (Fig. 8). A similar
tendency was previously observed on apatites sampled in the vicinity of Thuès hot springs and was proposed to be related to
combined $^4$He loss and REE depletion during hydrothermal fluids circulation (TET profile; Milesi et al., 2019).

Farther west, on the PLA profile, gneiss samples PLA3, sampled outside the DZ, and PLA2, from outer DZ, show
limited intra-sample age dispersion with mean AHe ages of $16.7 \pm 0.9$ Ma and $13.6 \pm 0.6$ Ma, respectively. eU values vary
between 15.8 and 55.7 ppm for these samples. Apatites from the outer DZ have lower REE and more variable contents
compared to apatites from outside the DZ (Fig. 8). However, no systematic correlation can be observed between AHe ages
and REE contents by contrast with the previous sample CAR7.







**Figure 8.** Chondrite normalised REE patterns (Sun and McDonough, 1989) for the Têt fault footwall samples, respectively outside and inside the DZ. Fd : fault distance.





**Table 1 AHe ages of footwall samples**

| Profile | Fd | Sample/ | Rs | U | Th | eU | Th/U | 4He | ± s | Ft | Corrected age | Error |
|---|---|---|---|---|---|---|---|---|---|---|---|---|
| | m | grain | μm | ppm | ppm | ppm | | ncc/g | ncc/g | | Ma | ±1σ (Ma) |
| **TET profile** | 5 | TET1 (42.52819N 2.24912E 776m) Highly fractured augen gneiss with chlorite | | | | | | | | | | |
| | | TET1-11 | 62.8 | 18.1 | 3.1 | 18.8 | 0.2 | 860.6 | 43.0 | 0.82 | 0.5 | 0.2 |
| | | TET1-12 | 61.3 | 39.1 | 0.7 | 39.3 | <0.1 | 1154.8 | 46.2 | 0.80 | 0.3 | <0.1 |
| | 5 | TET8 (42.52957N 2.25050E 850m) Augen gneiss with chlorite | | | | | | | | | | |
| | | TET8-1 | 70.8 | 13.4 | 13.7 | 16.6 | 1.0 | 51869.3 | 1556.1 | 0.84 | 30.9 | 2.5 |
| | | TET8-2 | 58.7 | 20.4 | 27.0 | 26.8 | 1.3 | 61359.9 | 1840.8 | 0.78 | 24.2 | 1.9 |
| | | TET8-3 | 58.1 | 10.9 | 16.7 | 14.9 | 1.5 | 58875.8 | 1766.3 | 0.79 | 41.6 | 3.3 |
| | | TET8-4 | 51.9 | 16.7 | 41.6 | 26.7 | 2.5 | 126713.5 | 1267.1 | 0.79 | 50.2 | 4.0 |
| | | TET8-5 | 76.8 | 7.9 | 3.9 | 8.8 | 0.5 | 13892.5 | 694.6 | 0.858 | 15.2 | 1.2 |
| **CAR sample** | 200 | CAR7 (42.520623N 2.22180E 900m) Fine-grained mylonite with few silica veins | | | | | | | | | | |
| | | CAR7-1* | 59.2 | 42.7 | 4.5 | 43.8 | 0.1 | 46774.3 | 935.48 | 0.79 | 11.2 | 0.5 |
| | | CAR7-2* | 78.1 | 5.7 | 1.2 | 6.0 | 0.2 | 4599.2 | 137.97 | 0.84 | 7.6 | 0.4 |
| | | CAR7-3* | 79.1 | 11.4 | 1.3 | 11.7 | 0.1 | 12480.9 | 374.4 | 0.85 | 10.4 | 0.5 |
| | | CAR7-4* | 70.0 | 23.1 | 3.2 | 23.9 | 0.1 | 29276.0 | 878.28 | 0.82 | 12.4 | 0.5 |
| | | CAR7-5* | 56.5 | 9.8 | 2.4 | 10.3 | 0.2 | 13434.4 | 403.02 | 0.78 | 13.8 | 0.7 |
| | | CAR7-6* | 68.0 | 48.0 | 6.3 | 49.5 | 0.1 | 70340.1 | 703.4 | 0.82 | 14.3 | 0.5 |
| **GAL profile** | 200 | GAL7 (42.51505N 2.19904E 1025m) Fractured fine grained gneiss with quartz and calcite veins and locally oxides | | | | | | | | | | |
| | | GAL7-1* | 55.2 | 14.7 | 7.1 | 16.5 | 0.5 | 16201.3 | 486.03 | 0.77 | 10.6 | 0.3 |
| | | GAL7-2* | 53.0 | 19.2 | 16.3 | 23.1 | 0.9 | 21563.9 | 431.26 | 0.77 | 10.0 | 0.4 |
| | | GAL7-3* | 56.9 | 33.8 | 11.9 | 36.6 | 0.4 | 36854.0 | 368.54 | 0.79 | 10.6 | 0.3 |
| | | GAL7-4* | 53.5 | 37.6 | 14.3 | 41.0 | 0.4 | 36945.2 | 369.45 | 0.74 | 10.1 | 0.3 |
| | | | | | | | | | | **Mean** | **10.3** | **0.2** |
| | 330 | GAL6 (42.51368N 2.19888E 1090m) Fine-grained gneiss with few quartz and calcite veines | | | | | | | | | | |
| | | GAL6-1 | 54.4 | 17.0 | 17.0 | 21.0 | 1.0 | 22342.1 | 446.8 | 0.77 | 11.4 | 0.9 |
| | 920 | GAL3 (42.51018N 2.20525E 1363m) Fine grained gneiss | | | | | | | | | | |
| | | GAL3-1* | 65.9 | 9.9 | 11.5 | 12.7 | 1.2 | 17196.9 | 343.9 | 0.80 | 14.1 | 1.1 |
| | | GAL3-2* | 62.5 | 12.4 | 5.8 | 13.8 | 0.5 | 14266.4 | 428.0 | 0.79 | 10.8 | 0.9 |
| | | GAL3-3* | 60.2 | 7.6 | 8.2 | 9.6 | 1.1 | 7861.2 | 314.4 | 0.79 | 8.6 | 0.7 |
| | | GAL3-4* | 72.1 | 9.9 | 8.0 | 11.8 | 0.8 | 12294.0 | 245.9 | 0.82 | 10.6 | 0.8 |
| | | GAL3-5 | 70.8 | 15.7 | 7.7 | 17.5 | 0.5 | 17324.0 | 519.7 | 0.82 | 10.0 | 0.8 |
| | | | | | | | | | | **Mean** | **10.8** | **0.9** |
| **ST profile** | 15 | ST15 (42.50357N 2.16602E 1123m) Highly fractured mylonite with silica veins | | | | | | | | | | |
| | | ST15-1* | 70.8 | 16.7 | 4.5 | 17.8 | 0.3 | 19703.4 | 985.2 | 0.82 | 11.2 | 1.1 |
| | | ST15-2* | 69.5 | 7.8 | 1.6 | 8.2 | 0.2 | 9959.6 | 597.6 | 0.82 | 12.3 | 1.5 |
| | | | | | | | | | | | **Mean** | **11.8** | **0.3** |
| | 100 | ST16 (42.50331N 2.16692E 1107m) Highly fractured fine grained mylonite with silica veins | | | | | | | | | | |
| | | ST16-1* | 71.5 | 11.2 | 1.7 | 11.6 | 0.1 | 14439.8 | 433.2 | 0.82 | 12.5 | 1.0 |
| | | ST16-2* | 59.2 | 22.3 | 3.8 | 23.2 | 0.2 | 33451.9 | 1003.6 | 0.79 | 15.2 | 1.2 |
| | | ST16-3* | 63.2 | 25.7 | 2.9 | 26.4 | 0.1 | 38790.1 | 775.8 | 0.80 | 15.2 | 1.2 |
| | | | | | | | | | | | **Mean** | **14.2** | **0.8** |
| | 250 | ST2 (42.50150N 2.16661E 1217m) Chloritized and locally oxidised gneiss | | | | | | | | | | |
| | | ST2-1* | 71.5 | 14.4 | 2.1 | 14.9 | 0.1 | 24260.5 | 1213.0 | 0.82 | 16.4 | 1.5 |
| | | ST2-2* | 49.4 | 30.8 | 3.6 | 31.7 | 0.1 | 41404.3 | 1656.2 | 0.73 | 14.8 | 1.3 |
| | | ST2-3* | 65.3 | 41.7 | 17.2 | 45.8 | 0.4 | 70061.7 | 2101.9 | 0.81 | 15.7 | 1.4 |
| | | ST2-4* | 55.6 | 31.2 | 6.0 | 32.7 | 0.2 | 50171.4 | 2006.9 | 0.78 | 16.3 | 1.5 |
| | | ST2-5* | 51.0 | 102.9 | 118.3 | 131.3 | 1.1 | 142063.2 | 1420.6 | 0.74 | 12.1 | 0.7 |
| | | ST2-6* | 66.5 | 27.2 | 72.2 | 44.5 | 2.7 | 39835.8 | 796.7 | 0.81 | 9.2 | 0.5 |
| | | | | | | | | | | | **Mean** | **15.8** | **0.8** |
| | 400 | ST3 (42.50001N 2.16697E 1174m) Unaltered gneiss with biotite | | | | | | | | | | |
| | | ST3-1 | 63.9 | 42.9 | 11.9 | 45.7 | 0.3 | 63845.3 | 1915.4 | 0.80 | 14.4 | 1.2 |
| | | ST3-2 | 74.3 | 15.1 | 2.6 | 15.8 | 0.2 | 18099.0 | 905.0 | 0.83 | 11.5 | 0.9 |
| | | ST3-3 | 59.3 | 57.4 | 9.6 | 59.7 | 0.2 | 89740.7 | 2692.2 | 0.76 | 16.4 | 1.3 |
| | | ST3-4 | 49.1 | 20.8 | 6.8 | 22.5 | 0.3 | 20848.7 | 1042.4 | 0.74 | 10.4 | 0.8 |
| | | ST3-5 | 50.1 | 52.9 | 25.5 | 59.0 | 0.5 | 70579.6 | 2117.4 | 0.76 | 13.1 | 1.0 |
| | 835 | ST4 (42.49567N 2.16784E 1249m) Muscovite leucogranite | | | | | | | | | | |
| | | ST4-1* | 62.5 | 150.6 | 3.6 | 151.4 | 0.0 | 239669.3 | 2396.7 | 0.79 | 16.5 | 1.4 |
| | | ST4-2* | 71.0 | 176.0 | 2.4 | 176.6 | 0.0 | 303270.8 | 2274.5 | 0.83 | 17.2 | 1.6 |
| | | ST4-3* | 61.5 | 138.2 | 2.5 | 138.8 | 0.0 | 218022.0 | 2180.2 | 0.79 | 16.5 | 1.2 |
| | | ST4-4* | 63.5 | 73.6 | 13.1 | 76.7 | 0.2 | 153280.5 | 1532.8 | 0.82 | 20.3 | 2.1 |
| | | ST4-5* | 67.9 | 80.4 | 1.6 | 80.7 | 0.0 | 132302.5 | 1323.0 | 0.84 | 16.2 | 1.3 |
| | | | | | | | | | | | **Mean** | **17.3** | **0.8** |
| | 1750 | ST8 (42.48856N 2.17261E 1480m) Augen gneiss with biotite | | | | | | | | | | |
| | | ST8-1* | 52.0 | 39.0 | 13.1 | 42.1 | 0.3 | 83719.3 | 837.2 | 0.76 | 21.6 | 1.7 |
| | | ST8-2* | 61.6 | 34.2 | 8.5 | 36.2 | 0.2 | 70052.1 | 1401.0 | 0.81 | 19.8 | 1.6 |
| | | ST8-3* | 57.2 | 27.2 | 8.7 | 29.3 | 0.3 | 54141.9 | 1082.8 | 0.79 | 19.4 | 1.5 |
| | | ST8-4* | 43.5 | 63.2 | 28.4 | 70.1 | 0.4 | 107043.6 | 1070.4 | 0.72 | 17.6 | 1.4 |
| | | | | | | | | | | | **Mean** | **19.6** | **0.8** |
| **PLA profile** | 170 | PLA3 (42.49343N 2.15462E 1622m) Fractured leucocratic gneiss and locally oxidised | | | | | | | | | | |
| | | PLA3-1* | 62.0 | 43.6 | 16.6 | 47.6 | 0.4 | 55098.6 | 551.0 | 0.77 | 12.4 | 1.0 |
| | | PLA3-2* | 76.1 | 14.9 | 3.9 | 15.8 | 0.3 | 20356.2 | 407.1 | 0.84 | 12.7 | 1.0 |
| | | PLA3-3* | 73.7 | 20.1 | 4.0 | 21.0 | 0.2 | 31039.5 | 620.8 | 0.82 | 14.8 | 1.2 |
| | | PLA3-4* | 52.7 | 34.5 | 12.5 | 37.5 | 0.4 | 49640.8 | 595.7 | 0.76 | 14.5 | 1.2 |
| | | | | | | | | | | | **Mean** | **13.6** | **0.6** |
| | 920 | PLA2 (42.48671N 2.15489E 1682m) Augen gneiss | | | | | | | | | | |
| | | PLA2-1* | 48.9 | 21.2 | 10.9 | 23.8 | 0.5 | 36356.6 | 727.1 | 0.74 | 17.1 | 1.4 |
| | | PLA2-2* | 52.6 | 11.5 | 6.4 | 13.1 | 0.6 | 17749.9 | 710.0 | 0.76 | 14.8 | 1.2 |
| | | PLA2-3* | 68.4 | 16.4 | 8.3 | 18.4 | 0.5 | 34659.4 | 519.9 | 0.82 | 19.1 | 1.5 |
| | | PLA2-4* | 51.5 | 23.4 | 14.5 | 26.8 | 0.6 | 38053.9 | 665.9 | 0.75 | 15.7 | 1.3 |
| | | | | | | | | | | | **Mean** | **16.7** | **0.9** |

Note : Samples with * are also analysed for REE / Fd : Fault distance





### 4.2 Hanging wall samples

AHe ages and REE patterns of apatites from samples of the hanging wall of the Têt normal fault are summarized in Table 2 and Figures 7 and 9. The first observation is that AHe ages are older than in the footwall, with an age difference of about 15-20 Ma for samples outside the DZ on the ST and GAL profiles. As in the footwall, no AHe ages-eU values relationship can be evidenced (Supplement Section S.5). Apatites from hanging wall display basically REE patterns similar to those of the footwall (S-type granites, Sha and Chappell, 1999).

### 4.1.1 TET and ST profiles

On the TET profile, AHe ages from sample TETHW, a gneiss lens within the inner DZ, are very dispersed, i.e. between $14.8 \pm 0.7$ Ma and $45.6 \pm 2.7$ Ma. eU values are highly dispersed, ranging from 48.8 to 328.8 ppm, and much higher than those of footwall apatites (this work and Milesi et al., 2019). A single grain from sample TETHW (TETHW-5) exhibit significantly lower contents of all REE compared to other apatites (Fig. 9). It is noticeable that this REE depleted apatite provides a much younger age than other co-existing grains, which is in line with previous results in the footwall of the same
profile (Milesi et al., 2019).

   On the ST profile, sample ST1 (gneiss) from the inner DZ shows old and dispersed AHe ages between $32.8 \pm 2.4$ Ma and $43.2 \pm 3.1$ Ma. Samples ST11 (gneiss) and ST14 (granite) from the outer DZ display weak intra-sample age dispersion, with mean ages of $22.7 \pm 1.6$ Ma and $26.9 \pm 2.6$, respectively. Sample ST12 (granite), outside the DZ yields moderately dispersed AHe ages from $29.4 \pm 2.3$ Ma to $36.8 \pm 2.9$ Ma with a mean age $32.6 \pm 1.5$ Ma. eU values of apatite grains appear
to be controlled mainly by the lithology, between 45.3 and 85.8 ppm for gneissic samples (ST1 and ST11) and between 11.6 and 37.7 ppm for granitic samples (ST12 and ST14). REE patterns of all apatite grains are very similar to each other independently of AHe age variations (Fig. 9).

   Apatite grains from granite ML1 show intra-sample AHe age dispersion between $26.2 \pm 2.1$ Ma and $34.1 \pm 2.7$ Ma with a mean age of $29.6 \pm 2.2$ Ma. This age is consistent with an AHe age of $29.8 \pm 1.5$ Ma previously obtained by Maurel et
al. (2008) on two apatite populations of 30 grains from the same sample. This age is also consistent with the mean AHe age of the granite ST12 at roughly the same elevation.

### 4.1.2 GAL and PLA profiles

   Sample GAL8 from the inner DZ displays dispersed AHe ages between $14.2 \pm 1.1$ Ma and $26.0 \pm 2.1$ Ma with eU values between 12.2 and 35.3 ppm. Samples GAL1 (outside the DZ) and GAL2 (outer DZ) from gneisses at a distance of 425
m and 115 m from the Têt fault, respectively, show moderate intra-sample AHe age dispersion with mean AHe ages of $29.5 \pm 1.6$ Ma and $20.6 \pm 0.9$ Ma, respectively. Sample GAL1 shows low eU values, i.e. between 4.3 and 9.1 ppm, compared to sample GAL2 with eU values between 23.6 to 53.1 ppm.



A single sample of gneiss (PLA5) from the outer DZ shows old and dispersed AHe ages between 23.3 ± 1.9 Ma and 41.5 ± 3.3 Ma and eU values from 26.9 and 55.7 ppm (see Supplement Section S.5 for AHe vs eU graph). One apatite grain (PLA5-

4) exhibits a different REE pattern with no light REE depletion (Fig. 8) probably due to the presence of Th-rich inclusion (Table 2, grain PLA5-4). Consequently, this apatite grain will be discarded in the discussion.

## Hanging wall samples

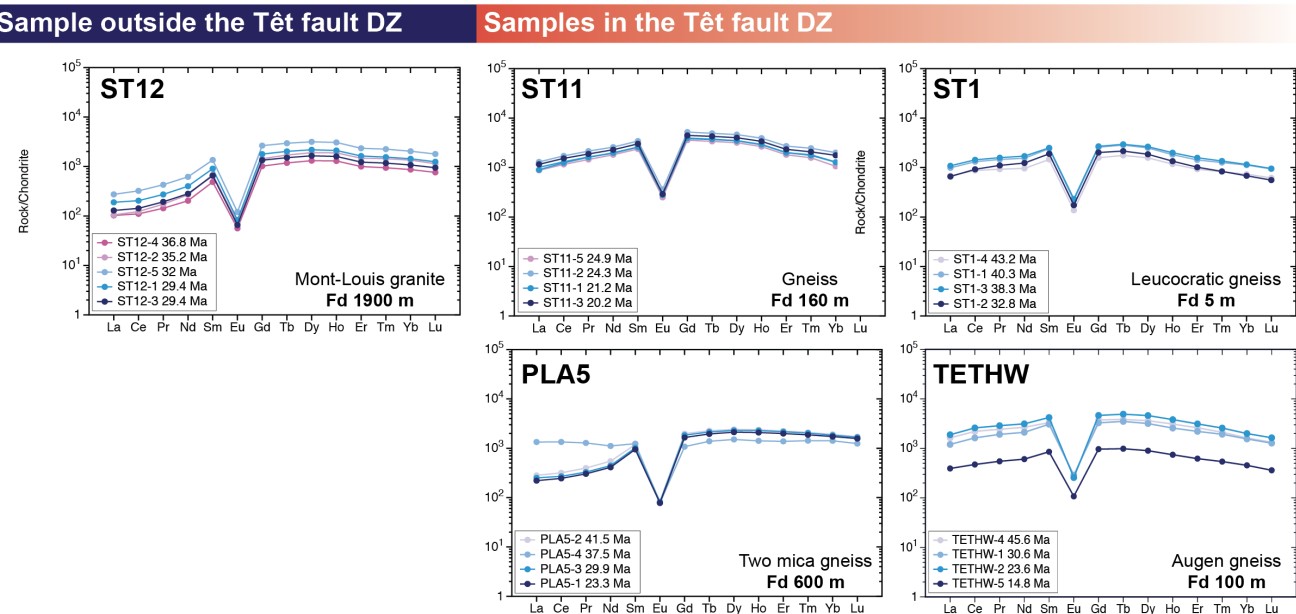

**Figure 9. Chondrite normalised REE patterns (Sun and McDonough, 1989) for the Têt fault hanging wall samples. REE patterns are reported for samples outside and inside the DZ. Fd : fault distance.**





**Table 2 AHe ages of hanging wall samples**

| Profile | Fd | Sample/ | Rs | U | Th | eU | Th/U | 4He | ± s | Ft | Corrected age | Error |
|---|---|---|---|---|---|---|---|---|---|---|---|---|
| | m | grain | μm | ppm | ppm | ppm | | ncc/g | ncc/g | | Ma | ±1σ (Ma) |
| **TET profile** | 100 | **TETHW** (42.52972N 2.24694E 770m) Gneiss | | | | | | | | | | |
| | | TETHW-1* | 52.1 | 324.3 | 18.4 | 328.7 | 0.1 | 882336.7 | 8823.4 | 0.73 | 30.6 | 1.8 |
| | | TETHW-2* | 54.7 | 133.4 | 18.7 | 137.9 | 0.1 | 298712.7 | 2987.1 | 0.76 | 23.6 | 1.6 |
| | | TETHW-3 | 62.6 | 64.4 | 9.0 | 66.5 | 0.1 | 117207.1 | 2344.1 | 0.78 | 18.8 | 1.1 |
| | | TETHW-4* | 61.8 | 102.1 | 21.6 | 107.3 | 0.2 | 471710.9 | 4717.1 | 0.80 | 45.6 | 2.7 |
| | | TETHW-5* | 49.6 | 47.4 | 5.8 | 48.8 | 0.1 | 66340.6 | 1990.2 | 0.76 | 14.8 | 0.7 |
| **GAL profile** | 40 | **GAL8** (42.51505N 2.19904E 905 m) Fine-grained gneiss chloritized and silicified paragneiss | | | | | | | | | | |
| | | GAL8-1 | 68.9 | 11.2 | 4.2 | 12.2 | 0.4 | 21079.6 | 421.6 | 0.84 | 17.0 | 1.4 |
| | | GAL8-2 | 57.7 | 32.0 | 13.7 | 35.3 | 0.4 | 68825.8 | 688.3 | 0.77 | 20.9 | 1.7 |
| | | GAL8-3 | 56.5 | 23.8 | 19.2 | 28.4 | 0.8 | 68655.2 | 686.6 | 0.77 | 26.0 | 2.1 |
| | | GAL8-4 | 55.1 | 11.8 | 5.8 | 13.1 | 0.5 | 17437.3 | 523.1 | 0.78 | 14.2 | 1.1 |
| | | GAL8-5 | 58.5 | 15.4 | 14.8 | 19.0 | 1.0 | 38387.2 | 767.7 | 0.78 | 21.4 | 1.7 |
| | 115 | **GAL2** (42.51707N 2.19633E 982m) Fine-grained gneiss | | | | | | | | | | |
| | | GAL2-1 | 58.0 | 48.9 | 17.5 | 53.1 | 0.4 | 101612.7 | 1016.1 | 0.78 | 20.4 | 1.6 |
| | | GAL2-2 | 71.7 | 22.4 | 4.6 | 23.6 | 0.2 | 50982.5 | 611.8 | 0.80 | 22.5 | 1.8 |
| | | GAL2-3 | 64.4 | 21.3 | 19.9 | 26.1 | 0.9 | 55346.4 | 664.2 | 0.80 | 22.1 | 1.8 |
| | | GAL2-4 | 59.5 | 24.7 | 23.0 | 30.2 | 0.9 | 49594.0 | 595.1 | 0.79 | 17.3 | 1.4 |
| | | GAL2-5 | 74.0 | 26.6 | 8.0 | 28.6 | 0.3 | 59317.2 | 889.8 | 0.83 | 20.8 | 1.7 |
| | | | | | | | | | | **Mean** | **20.6** | **0.9** |
| | 425 | **GAL1** (42.51881N 2.19395E 1081m) Augen gneiss | | | | | | | | | | |
| | | GAL1-1 | 60.6 | 4.5 | 3.6 | 5.4 | 0.8 | 15934.3 | 478.0 | 0.79 | 31.3 | 2.5 |
| | | GAL1-2 | 59.9 | 8.1 | 3.9 | 9.1 | 0.5 | 22677.0 | 453.5 | 0.78 | 26.5 | 2.1 |
| | | GAL1-3 | 60.1 | 3.4 | 4.0 | 4.3 | 1.2 | 12565.9 | 439.8 | 0.78 | 30.8 | 2.5 |
| | | | | | | | | | | **Mean** | **29.5** | **1.6** |
| **ST profile** | 5 | **ST1** (42.50798N 2.17611E 1073m) Highly fractured leucocratic gneiss with silica veins | | | | | | | | | | |
| | | ST1-1* | 73.9 | 58.4 | 5.6 | 59.8 | 0.1 | 235977.6 | 2359.8 | 0.81 | 40.3 | 3.2 |
| | | ST1-2* | 63.4 | 70.8 | 3.3 | 71.6 | 0.0 | 227052.2 | 3405.8 | 0.80 | 32.8 | 2.4 |
| | | ST1-3* | 68.4 | 79.1 | 3.1 | 79.8 | 0.0 | 299652.6 | 2996.5 | 0.81 | 38.3 | 2.7 |
| | | ST1-4* | 62.8 | 83.8 | 8.5 | 85.8 | 0.1 | 358555.4 | 3585.6 | 0.80 | 43.2 | 3.1 |
| | 160 | **ST11** (42.51047N 2.17598E 1081m) Unaltered gneiss with biotite | | | | | | | | | | |
| | | ST11-1* | 60.9 | 63.0 | 19.7 | 67.7 | 0.3 | 136793.9 | 1504.7 | 0.79 | 21.2 | 1.7 |
| | | ST11-2* | 61.9 | 75.8 | 25.7 | 81.9 | 0.3 | 187572.8 | 1875.7 | 0.78 | 24.3 | 1.9 |
| | | ST11-3* | 76.4 | 53.7 | 25.6 | 59.8 | 0.5 | 120320.0 | 1443.8 | 0.83 | 20.2 | 1.6 |
| | | ST11-5* | 55.6 | 42.2 | 13.0 | 45.3 | 0.3 | 101862.1 | 1018.6 | 0.75 | 24.9 | 2.0 |
| | | | | | | | | | | **Mean** | **22.7** | **1.6** |
| | 175 | **ST14** (42.50748N 2.16848E 1139m) Granite with biotite and few evidences of chloritization | | | | | | | | | | |
| | | ST14-1 | 71.4 | 21.9 | 2.5 | 22.5 | 0.1 | 64392.2 | 965.9 | 0.83 | 28.5 | 2.3 |
| | | ST14-2 | 55.9 | 17.6 | 3.5 | 18.5 | 0.2 | 38848.5 | 1165.5 | 0.77 | 22.5 | 1.8 |
| | | ST14-3 | 66.3 | 36.7 | 4.4 | 37.7 | 0.1 | 107444.1 | 1074.4 | 0.78 | 30.1 | 2.4 |
| | | ST14-4 | 59.4 | 32.7 | 10.3 | 35.2 | 0.3 | 88734.6 | 1064.8 | 0.79 | 26.6 | 2.1 |
| | | | | | | | | | | **Mean** | **26.9** | **2.3** |
| | 1900 | **ST12** (42.51239N 2.14921E 1552m) Granite with biotite | | | | | | | | | | |
| | | ST12-1* | 65.2 | 16.7 | 12.2 | 19.6 | 0.7 | 55653.7 | 1113.1 | 0.80 | 29.4 | 2.3 |
| | | ST12-2* | 72.9 | 15.8 | 11.4 | 18.6 | 0.7 | 63720.2 | 955.8 | 0.81 | 35.2 | 2.8 |
| | | ST12-3* | 74.0 | 23.5 | 8.8 | 25.6 | 0.4 | 73803.2 | 1291.6 | 0.81 | 29.4 | 2.4 |
| | | ST12-4* | 57.1 | 10.1 | 6.1 | 11.6 | 0.6 | 39116.4 | 1564.7 | 0.76 | 36.8 | 2.9 |
| | | ST12-5* | 72.6 | 26.1 | 17.4 | 30.3 | 0.7 | 94696.8 | 1420.5 | 0.81 | 32.0 | 2.6 |
| | | | | | | | | | | **Mean** | **32.6** | **1.5** |
| **ML sample** | 5000 | **ML1** (42.5898083N 2.184692E 1400m) Granite | | | | | | | | | | |
| | | ML1-1 | 60.0 | 31.0 | 39.0 | 40.4 | 1.3 | 120757.3 | 1207.6 | 0.81 | 30.5 | 2.4 |
| | | ML1-2 | 50.9 | 36.0 | 45.0 | 46.9 | 1.3 | 112014.0 | 1120.1 | 0.76 | 26.2 | 2.1 |
| | | ML1-3 | 61.8 | 17.1 | 25.0 | 23.1 | 1.5 | 60151.0 | 962.4 | 0.79 | 27.4 | 2.2 |
| | | ML1-4 | 66.0 | 18.6 | 12.0 | 21.6 | 0.7 | 70099.4 | 1261.8 | 0.79 | 34.1 | 2.7 |
| | | | | | | | | | | **Mean** | **29.6** | **2.2** |
| **PLA profile** | 600 | **PLA5** (42.49059N 2.13607E 1560m) Two mica fine grained gneiss | | | | | | | | | | |
| | | PLA5-1* | 59.8 | 29.6 | 10.8 | 32.2 | 0.4 | 72354.2 | 795.9 | 0.80 | 23.3 | 1.9 |
| | | PLA5-2* | 69.2 | 26.1 | 15.6 | 29.9 | 0.6 | 121524.9 | 1215.2 | 0.81 | 41.5 | 3.3 |
| | | PLA5-3* | 48.1 | 22.3 | 19.2 | 26.9 | 0.9 | 74554.5 | 1118.3 | 0.77 | 29.9 | 2.4 |
| | | PLA5-4* | 77.3 | 54.8 | 3.6 | 55.7 | 0.1 | 203870.1 | 2038.7 | 0.81 | 37.5 | 3.0 |

**Note :** Samples with * are also analysed for REE / Fd : Fault distance

## 5 Discussion

### 5.1 Footwall of the Têt fault

In order to interpret AHe ages from the footwall of the Têt fault DZ, we used apatites from the samples outside the DZ as reference. Samples outside the DZ (GAL3, ST4, ST8 and PLA2) show accurate AHe ages with weak intra-sample

variations (Table 1 and 2). All apatites from outside the DZ, including those from the work of Milesi et al. (2019), show similar REE patterns, consistent with S- type granite apatites, except for the fine-grained gneiss sample GAL3 displaying a REE pattern usually found in apatites from paragneiss lithologies (Henrichs et al., 2018). We define then two reference REE compositional-AHe age fields: one for the GAL profile and another for all the other profiles. eU values for all apatite grains





are mostly under 50 ppm with the exception of those from a leucogranite (sample ST4) which is a lithology generally rich in
uranium (Cuney, 2010; Ballouard et al., 2017; see Table 2 and Supplement Section S.5) but showing similar REE patterns than
other gneisses (ST8, PLA2).

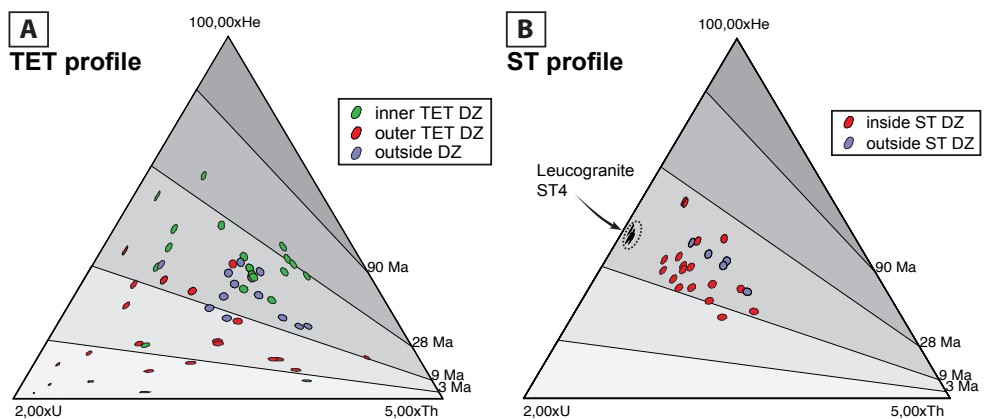

**Figure 10. U-Th-He diagrams using Helioplot (Vermeesch, 2010) for A) TET profile (this study and Milesi et al., 2019) and B) ST profile**

### 5.1.1 Nearby active hot spring clusters: TET and ST profiles

U-Th-He ternary diagrams show a larger intra- and inter-sample isotopic dispersion for apatites within the DZ
compared to those outside the DZ (Fig. 10). This is peculiarly evident for the TET profile where newly analysed samples yield
perfectly concordant results with previous ones (Milesi et al., 2019). On Figure 10A, we can identify three fields that partially
overlap: (1) defined by apatites outside the DZ (violet dots) with AHe ages between ~9 and 25 Ma interpreted to record the
regional cooling of the Carança massif (Fig. 2); (2) defined by mostly young apatites (< 9 Ma) from the outer DZ (red dots)
interpreted as apatites with partially reset AHe ages by hydrothermal activity (Milesi et al., 2019); and (3) a third field
corresponding to apatites from the inner DZ (green dots) with very scattered AHe ages but mostly above 20 Ma interpreted as
apatites contaminated by excess [4]He (Milesi et al., 2019). On Figure 10B, apatites from the outer and inner DZ along the ST
profile (red dots) yield less dispersed AHe ages, but mainly younger, compared to those from outside the DZ (violet dots).
Note that ST4 apatites retrieved from a leucogranite plot within a distinct compositional field. It is noticeable that along both
profiles, a rejuvenation of AHe ages is observed even if it is more pronounced along the TET profile.

TET and ST apatites from the outer DZ display similar depleted REE patterns compared to apatites sampled outside
the DZ, without fractionation between the light REE and heavy REE (Fig. 11). This global depletion of REE is more intense
for TET samples compared to ST ones. However, it must be pointed out that along the ST profile, the youngest apatites
correspond to the most REE depleted ones, consistently with the results from the TET profile (Fig. 11). The only three analysed
apatite grains from the inner DZ along the TET profile (sample TET1) do not exhibit this REE depletion (Milesi et al., 2019).
Such a small number of analyses does not allow further comment.





Intra- and inter-sample AHe age dispersion from samples within the DZ cannot be simulated by any thermal helium diffusion models in apatite (Flowers et al., 2009; Gautheron et al., 2009). In particular, young AHe ages cannot be modelled and do not fit the regional thermal history defined in section 2.1 (Supplement Section S.6). This result, in accordance with those of Milesi et al. (2019) for samples in the outer DZ on the TET profile, questions the origin of the AHe age scattering near hydrothermal circulation zones. We proposed that the young apatite grains were subjected to hydrothermal alteration that preferentially enhances $^4$He daughter loss and apatite rejuvenation rather than U-Th parent supply, which is inconsistent with the observed REE depletion (Figs. 10 and 11; Milesi et al., 2019). REE depletion without fractionation associated to hydrothermal alteration of apatite grains has already been evidenced experimentally (Harlov et al., 2005). These authors showed that the reacted region of treated apatite is depleted in REE+Y and Cl compared to the unreacted region and that REE have been carried by fluids circulating through nano-channels or nano-voids that developed within the lattice of reacted apatites. As $^4$He is mainly trapped in lattice defects (Zeitler et al., 2017), it is highly susceptible to dissolve in fluids and thus to be lost by the host apatites mainly by advection during hydrothermal alteration processes. Indeed, it is well-known that hydrothermal fluids are highly $^4$He enriched in highly fractured granitic environments even in the absence of U mineralization (e.g. Andrews and Lee, 1989; Paternoster et al., 2017). The variable REE loss between apatite grains from different samples or even at an intra-sample scale can also be related to the heterogeneity of hydrothermal fluid circulations (Fig. 5, Caine et al., 1996; Bense et al., 2013). Indeed, even at a thin section scale, unaltered domains can be distinguished from more altered domains, thus reflecting a variable imprint of fluid interaction. We cannot exclude a control of the chemistry and the temperature of fluid flow that can change with time (Favara et al., 2001; Cox et al., 2015). However, it is noticeable that apatite rejuvenation, AHe age dispersion and REE depletion are much more important near the Thuès hot springs than near the St-Thomas ones (Figs. 10 and 11). This may be related to more important hydrothermal fluid circulations near Thuès as exemplified by the greater number of hot springs near Thuès (10 springs) than in St-Thomas (4 springs). The specific topography around the Thuès area (Taillefer et al., 2017) or possible enhanced permeability at depth can favour longer-lived hydrothermal and higher fluid temperatures in Thuès (up to 73°C vs. < 60°C at St-Thomas). We cannot exclude the impact of the intersecting fault network in St-Thomas area (Fig. 5), which can be responsible of an increase of its overall permeability close to the surface and flow and heat diffusion in a larger volume of fractured rocks compared to the more channelized system observed in the Thuès-les-Bains hot spring cluster. This is highlighted by the localisation of the St Thomas and nearby Prats-Balaguer hot springs on secondary NW-SE and NE-SW faults cross-cutting or branched onto the Têt fault, which contrasts with the Thuès hot springs that are all adjacent to the Têt fault.




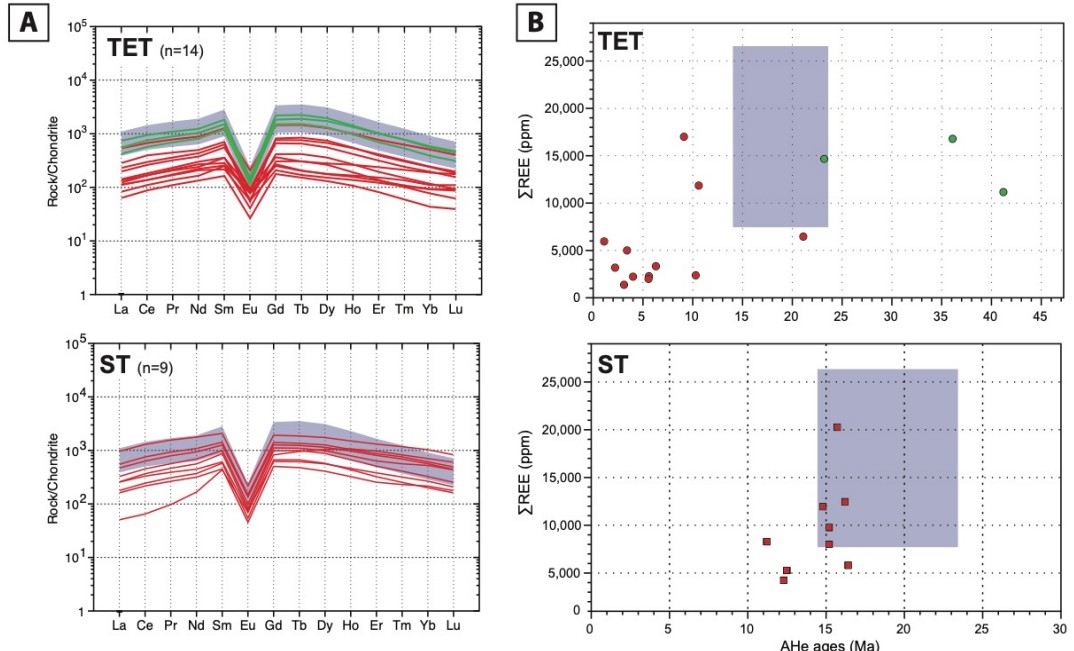

**Figure 11. A) REE patterns of samples within the DZ for ST and TET profile; blue field: compositional field of samples outside the DZ defined by 13 REE spectra (gneiss samples TET5, ST4, ST8 and PLA2). For TET profile, samples from the inner DZ (green) and outer DZ (red) are distinguished. B) Sum of REE contents as a function of AHe ages. The purple field corresponds to samples outside the DZ showing consistent REE patterns. The depletion of all REE is associated with a younging of AHe ages.**

### 5.1.2 Free hot spring areas: GAL and PLA profiles, CAR valley

Interestingly, we also obtain slightly rejuvenated and scattered AHe ages and depleted REE patterns for apatite grains from outer DZ footwall samples located in two areas away from hot springs clusters: Carança valley and Planès profile (Figs 12A and 13A).

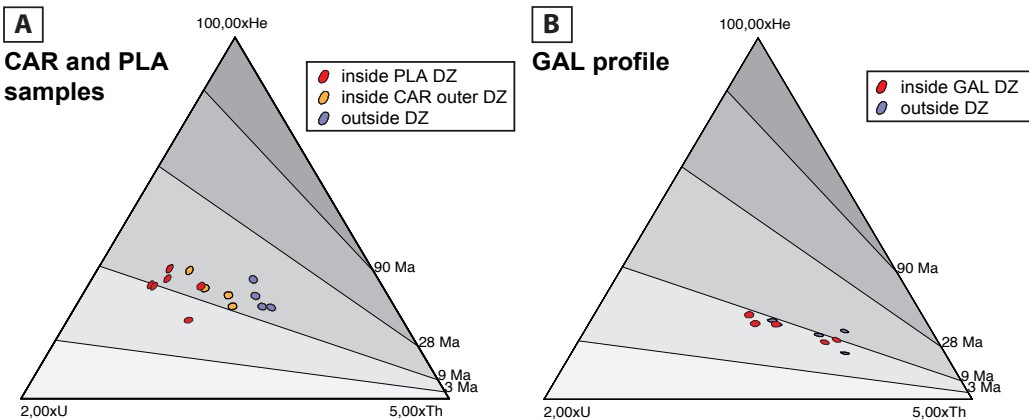

**Figure 12. U-Th-He diagrams using Helioplot (Vermeesch, 2010) for A) CAR and PLA samples (this study and Milesi et al., 2019) and B) GAL profile**



CAR7 apatites from the outer DZ at ~2 km west of the Thuès hot spring cluster exhibit dispersed and reset AHe ages but are also depleted in REE relatively to the outside DZ reference samples (Fig. 13). They are less rejuvenated, but their ages are consistent with AHe ages from the TET outer DZ and their REE depletion is less pronounced (Figs. 10 and 13). Yet again, dispersed AHe ages cannot be simulated by any helium thermal diffusion models (Flowers et al., 2009; Gautheron et al., 2009) and young ones peculiarly are not consistent with the regional thermal history (see section 2.1 and Supplement Section S.6).

Consequently, despite the lack of hot springs in the Carança valley, CAR7 apatites appear to record some thermal resetting as in the TET outer DZ. This suggests that hydrothermal circulations took place in the Carança valley and are no more active. The lack of surface hydrothermal circulations today may be related to the sealing of fluid-filled fractures in agreement with the occurrence of calcite and/or silica filled fractures in this area (Fig. 5D–E; Milesi et al., 2019). A resulting question is whether or not this hydrothermal system can be linked to that of Thuès-les-bains. If so, the present data on TET and CAR

samples would suggest the presence of an intense and recent hydrothermal flow over a larger segment of the fault than today and thus a larger thermal anomaly around Thuès than that indicated by numerical modelling (Taillefer et al., 2018). In this respect, the lack of surface hydrothermal activity in the Carança valley might then be very recent and a hidden geothermal system could still exist there nowadays.

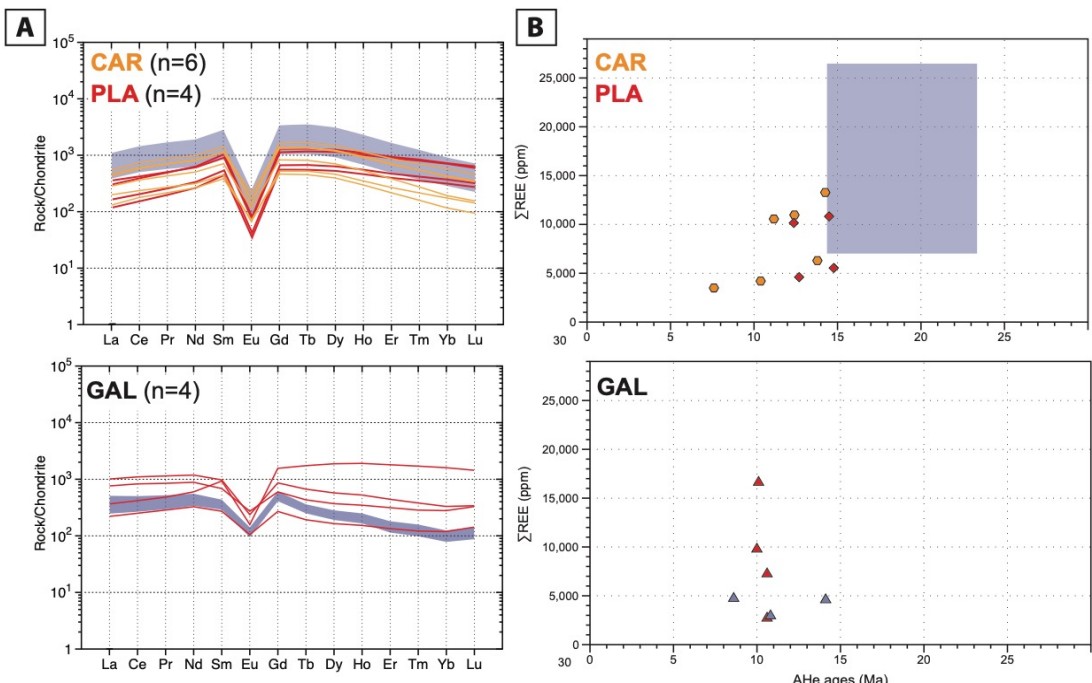

**Figure 13. A) REE patterns of samples within the DZ; blue field: compositional field of samples outside the DZ defined by 13 REE**
**spectra (gneiss samples TET5, ST4, ST8 and PLA2). B) Sum of REE contents as a function of AHe ages. The purple field and triangles correspond to samples outside the DZ showing consistent REE patterns. The depletion of all REE is associated with a youning of AHe ages.**

In the Planès area which should correspond to a recharge area of the hydrothermal system with cold surface water circulations (Taillefer et al., 2018), apatite grains from sample PLA3 in the outer DZ yield similar results to the ones from St-



Thomas hot spring cluster with variously depleted REE patterns and ages of 13.6 ± 0.6 Ma (Figs. 7 and 13). These young ages
at 1600 m of altitude do not fit the regional thermal history depicted above in Figure 2. The geothermal anomaly around St-
Thomas might have extended 1.5-2 km to the west, which is also inconsistent with the numerical thermal modelling (Taillefer
et al., 2018). Alternatively, a very circumscribed hydrothermal system may have been active on the plateau independently of
that of St Thomas. The Planès area, such as the Carança valley, may also correspond to a hidden geothermal system. Both
locations might then be potential sites for future geothermal exploration.

At the foot of Gallinàs peak (GAL profile) where numerical models propose the occurrence of a geothermal anomaly
(Taillefer et al., 2018), our results do not evidence significant AHe ages scattering while we can note that REE content is more
variable within the outer DZ (Fig. 13). Although the number of analysed grains is limited, the homogeneity of the data suggests
no significant hydrothermal circulations within the DZ in the last 10 Ma. This does not support the presence of a significant
thermal anomaly in this area since 10 Ma, in contrast to what the numerical models suggest (Taillefer et al., 2018). The presence
of calcite-quartz filled fractures and chloritisation observed in the samples from this area might support more ancient (> 10Ma)
hydrothermal circulations. Moreover, the thermal modelling of GAL samples from both outside the DZ and in the outer DZ
with QTQt software share a common history with the TET samples collected outside the DZ (Fig. 14). The modelled T-t path
suggests a rapid cooling between 30 and 24 Ma followed by a quiescent period and a second phase of cooling around 10 Ma
(Supplement Section S.6 for model parameters). This model shows a good fit and is compatible with previous regional thermal
histories indicating rapid cooling in the late Oligocene-early Miocene (Maurel et al., 2008; Milesi et al., 2019). At the scale of
the eastern Pyrenees, a second exhumation phase is suggested by Gunnell et al. (2009) with a 2 km uplift in the Tortonian and
could account for a reactivation of the Têt fault around 10 Ma. The good fit of the model reinforces the proposal of lack of a
significant recent thermal anomaly along this segment of the Têt fault.

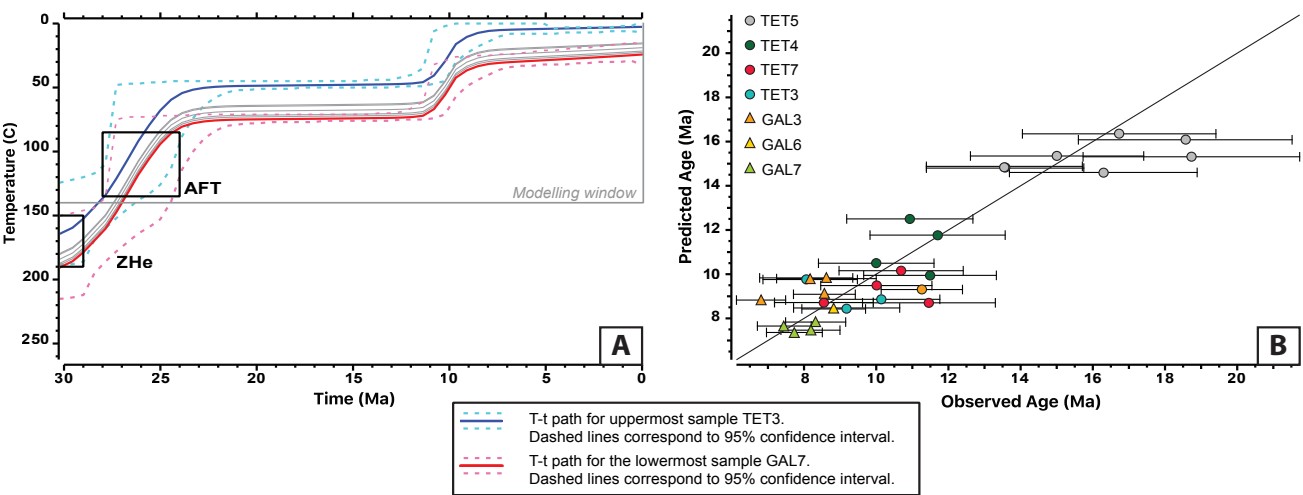

**Figure 14. A) Thermal modelling of the Têt fault footwall made with QTQt software (Gallagher, 2012) using TET outside DZ samples
and GAL DZ and outside DZ samples and computed with the Gautheron et al. (2009) diffusion parameters. AFT and ZHe constraints**





**are from Maurel et al. (2008). B) Predicted age vs Observed age graph for each apatite grain. 1:1 diagonal line corresponds to an ideal fit.**

As fluid flows through fractured rocks is a highly heterogeneous process, even at the thin section-scale (Fig. 6),

variable $^4$He loss by fluids advection can account for AHe ages dispersion even in areas distant from actual hot springs. This is supported by the global positive correlation between AHe rejuvenation and REE loss in samples from the Têt fault outer DZ footwall. Therefore, the combination of AHe dating and REE analyses allowed us evidencing hidden geothermal systems, maybe representing extension of the actual surficial ones, and thus to better constrain the geometry and intensity of recent geothermal anomalies along the Têt fault (Fig. 15). This result questions the interpretation of AHe ages in many cases

when potential hydrothermal alteration is not considered.

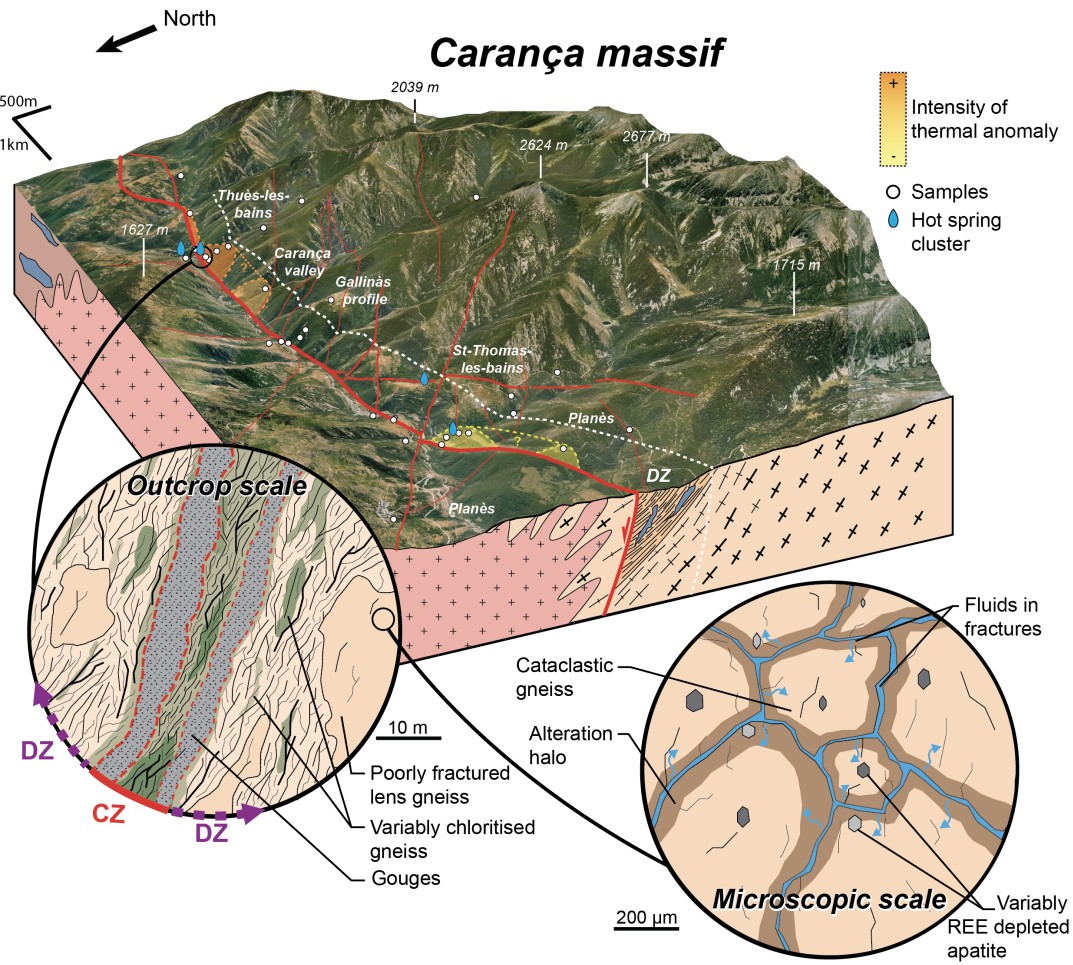

**Figure 15. Bloc 3D synthetic of the surface distribution of the thermal anomaly in the Têt fault footwall, based on AHe dating and REE analyses with zooms on the fault zone at outcrop and microscopic scale.**



## 5.2 Hanging wall of the Têt fault

Outside the hanging wall DZ, apatites from sample ML1 and ST12 yield AHe ages consistent with previously
published AHe ages in the Mont Louis massif (Maurel et al., 2008) and REE patterns from sample ST12 are compatible with
those of apatites from granitic lithology (Sha and Chappell, 1999), with a moderate LREE depletion.

Within the hanging wall DZ, samples nearest the Têt fault (TETHW, ST1 and GAL8) show a scattering and mainly
an ageing of AHe ages, as in the footwall inner DZ (Milesi et al., 2019). These scattered AHe ages (Fig. 3; Table 2) cannot be
properly modelled with He diffusion models in apatite (Flowers et al., 2009; Gautheron et al., 2009, see Supplement Section
S.6). Apatite REE content appears less variable within the hanging wall DZ, except for a single apatite grain from TETHW
sample that yields a depleted REE pattern and the youngest AHe age. This result may be related to the influence of the nearby
Canaveilles hot spring cluster in the hanging wall DZ just to the NE of the TET profile (Fig. 5) and to the same process of $^4$He
loss described in the footwall outer DZ (e.g. sample CAR7).

Samples from distal hanging wall DZ yield less scattered AHe than those closer to the Têt fault with the exception of
sample PLA5. However, all samples including PLA5 exhibit very similar REE patterns. The AHe ages scattering from all
these samples but PLA5 can be properly modelled with He diffusion models in apatite (See Supplement Section S.6). Therefore
samples from the distal hanging wall DZ (GAL1, GAL2, ST11 and ST14) are used in combination with those outside DZ
(ML1 and ST12) for modelling the thermal history of the hanging wall. The T-t path (Fig. 16) shows a rapid cooling between
35 and 30 Ma in the temperature range of 150-60°C and low cooling since 30 Ma. Slow cooling rates can account for the intra-
sample variation of AHe ages in the hanging wall away from the Têt fault in accordance with previous works (Fitzgerald et
al., 2006; Maurel et al., 2008; Brown et al., 2013). PLA5 sample is peculiar because it is the only sample located at a lower
elevation than the Têt fault surface trace (Fig. 5). It is therefore located lower than the geothermal system that we evidence in
the Têt fault footwall in the Planès area. This peculiar location may account for the AHe age scattering and ageing within this
sample, in addition to the age dispersion due to the above mentioned regional slow cooling.





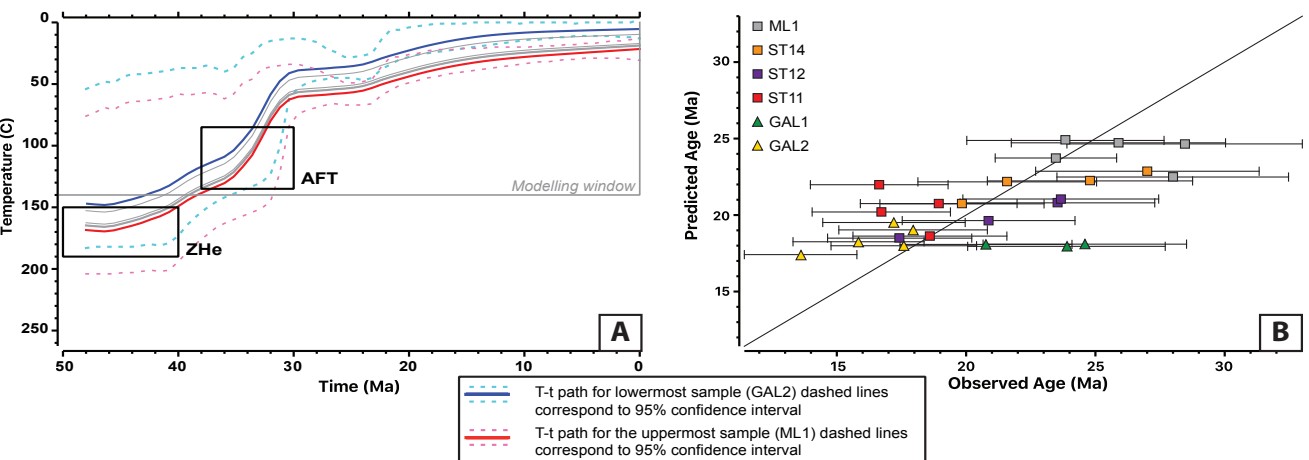

**Figure 16. A) Thermal modelling of the Têt fault hanging wall computed with QTQt software (Gallagher, 2012) and using the Gautheron et al. (2009) diffusion parameters. AFT and ZHe constraints are from Maurel et al. (2008). B) Predicted age vs Observed age graph for each apatite grain. 1:1 diagonal line corresponds to an ideal fit.**

In the hanging wall, disturbed AHe ages are observed mainly close to the Têt fault. Therefore, we consider that AHe age dispersion within the hanging wall near the Têt fault is indicative of hydrothermal circulations within this part of the

hanging wall despite the small number of samples does not allow a straightforward interpretation. This question the impermeable character of the hanging wall rocks (Taillefer et al., 2017). However, we show that moving away from the fault all samples within the DZ except one share a common thermal history with rocks far away from the fault (Fig. 16). This result indicates that they lack hydrothermal perturbation and is in accordance with the proposal of Taillefer et al. (2018) that hydrothermal circulations are mainly restricted to the footwall of the Têt fault.

**6 Conclusion**

This study confirms that the apatite (U-Th)/He thermochronometer can be impacted by local low-temperature hydrothermal circulations. In addition, we also highlight the mobility of REE during interactions between apatites and hydrothermal fluids (< 130°C), demonstrating that apatite grains can no longer be considered as a closed system in such a context. [4]He trapped within apatite lattice defects may be easily dissolved in hydrothermal fluids, circulating even through

nano-channels within the crystals lattice, and its loss may be enhanced rather by fluids advection than only by thermal diffusion as classically considered. As fluid flows through fractured rocks is a highly heterogeneous process, even at the thin section-scale, variable [4]He loss by fluids advection can account for AHe age dispersion. This is supported by the global positive correlation between AHe rejuvenation and REE loss in samples from the Têt fault outer DZ footwall. This process can then also partly account for AHe age dispersion in many cases as paleo-hydrothermal fluid circulations took place in various

magmatic, metamorphic and tectonic settings. This question the interpretation of AHe ages in many cases when potential hydrothermal alteration, peculiarly in the absence of obvious paleo-geothermal systems, is not carefully considered. As a





corollary, AHe thermochronology combined with REE analysis is an efficient tool to track active and paleo- geothermal systems. As an exploration tool, the use of (U-Th)/He thermochronology appears very complementary to other tools as, for example, hydrothermal fluid chemical analyses, fluid circulation numerical simulations or electrical methods. Moreover, it is

a cost-effective tool as it allows constraining such models without the need for drilling.

**Acknowledgments**

This work was funded by THERMOFAULT, a project supported by the Region Occitanie (France) involving TLS Geothermics (main sponsor), Géosciences Montpellier and the TelluS Program of CNRS/INSU. Thanks for technical support to Doriane Delmas and Christophe Nevado for thin section preparation, Lucie Koeller and Léa Causse for ICP-MS analyses
and Cyprien Astoury for apatite separation. Thanks to inhabitants of Thuès-Entre-Valls for their welcome during the field trip.

**Data availability**

The data that support the findings of this study are available in Supplement Section, more details upon request from the corresponding author, Gaétan Milesi (gaetan.milesi@umontpellier.fr).

**Author contribution**

The number of authors is quite large because significant analytical work was produced. Michael Bonno supervised the helium analyses, Olivier Bruguier and Céline Martin the ICPMS analyses, Audrey Taillefer did the field and numerical models of fluid and heat transfer, Mathieu Bellanger co-wrote the project and participate to its funding, Philippe Münch participated to the analysis of the thermochronological and REE data, Patrick Monié and Roger Soliva are the PhD directors and scientific advisors, and Gaétan Milesi is the PhD student who did the main part of the field and laboratory work.

**Competing interests**

The authors declare that they have no conflict of interest.

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
