# Peer review of "Tracking geothermal anomalies along a crustal fault using (U-Th)/He apatite thermochronology and REE analyses, the example of the Têt fault (Pyrenees, France)"

_Solid Earth, 2020_

## Referee Comment (RC1) · Meinert Rahn (Referee) · 17 May 2020

Review of

**Tracking geothermal anomalies along a crustal fault using (U-Th)/He apatite thermochronology and REE analyses, the example of the Têt fault (Pyrenees, France)**

submitted by Milesi G, Monié P, Münch P, Soliva R, Taillefer A, Bruguier O, Bellanger M, Bonno M, Martin C.

reviewed by Meinert Rahn
* * *
The submitted manuscript and supplementary material presents 99 new (U-Th)/He single grain ages and 63 REE spectra on apatites from the geothermal fields along the Têt fault in the Pyrenees. The study is closely linked to the recently published study Milesi et al. 2019, which already presented a restricted data set from the TET profile, a study that here is extended by new samples and samples along more profiles (STA, GAL, PAL) and a small number of more isolated samples. Samples were taken from the hanging wall and footwall of the fault at variable distance, divided into samples inside a deformation zone (DZ) around the fault and outside of it.

The study concludes that He ages and REE contents can be distinctly influenced by hydrothermal activity, mainly dependent on rock permeability, fault activity and thermal evolution. Apatite ages closest to the fault show a large age scatter and strong depletion of REE contents if compared to samples far away from the fault, which show the original REE patterns and the regional cooling history (as a result of exhumation and differential uplift along the fault).

Ages and REE support the conclusions and I see no substantial or harsh mismatches, even though the reader is left behind by some question marks concerning the ideas of the authors of the processes taking place. Since I have not major drawbacks; I propose that the paper needs minor revision of the text.

Points that the authors should have a closer look to are the following:

1. Normal o rhorizontal distance to the fault plane: The authors should clarify in their paper, whether the cited distances are horizontal distances or distances normal to the fault plane.
2. When the authors refer to Flowers et al. or Gautheron et al. diffusion models, then they should carefully choose their wording. The diffusion models are ok, but it is the fact that the AHe ages scatter too much to be modelled by a common thermal history.
3. The model the authors present for the fault plane suggests that the plane seems to have clearly restrict geothermal activity to the footwall. However, there are faults that crosscut the Têt fault into the hanging wall, and AHe ages next to the fault show considerable age scatter. So, the influence reaches beyond the fault plane. Note, that the rocks are by far not impermeable, in particular if they have undergone tectonic shearing.
4. The reference to Henrichs et al. (2018) to explain a common apatite REE pattern is problematic to me. First, you do not give any information about the degree of metamorphism, second, the paragneisses should contain apatites of different origin and different REE patterns in the beginning. Does the Henrichs et al. study refer to new-grown apatites or detrital apatites? Quoting this study should be checked again. In general, we do not need to know the original REE pattern, we simply need to have differences, but of course, in a paragneiss, there might not be a unique reference REE pattern.
5. The authors do calculate mean ages for some samples, for others they do not (Tables 1 and 2). They should clearly state on the basis of what they decide that a sample has internally consistent ages, but another one shows age scattering. Methodically, this is not clear yet, but would be very important to clarify the next steps. In addition, this may lead to the possibility to independently talk about the thermal history of the footwall on one side and the hanging wall on the other. The authors point to this in their abstract (lines 24/25), but the paper does not discuss the individual thermal histories. Perhaps, it would be wise to do this first.

6. If there is one chapter that I would recommend strong rewriting for, it is the Conclusions chapter, which at the moment caused the highest density of question marks to me.
7. I highly recommend to the authors to not speculate about the processes that have taken place and led to AHe age rejuvenation or increase or REE depletion. There is little evidence presented that would be able to narrow down potential processes that took place. Such work could be done on the same samples, but not based on the her presented evidence.

In my detailed comments below, I have noted the locations of a considerable number of minor issues that definitely need clarification, more than a simple rewording, but clarification with respect to the issue that are listed above. The entire paper, including the figures, tables and references, need some fine-tuning to make the study internally sound. In some cases, this means to delete some side tracks, which deviate more than they provide answers, but in some other cases, we need a more pronounced argumentation to better emphasize the point that should be made.

Detailed comments:

| | |
|---|---|
| Line 12 | "around" instead of "nearby" |
| Line 12 | Since you are mentioning the two hot spring clusters for the first time; I would suggest to shift the bracket from line 17 right after "clusters" |
| Line 12 | "in-between" |
| Lines 12/13 | delete "in an attempt" |
| Line 16 | to be concordant with line 12: "nearby the two hot spring clusters" |
| Line 17 | "resettings" instead of "resetting", we do not know whether the reset occurred at the same time. |
| Line 18 | "around the Thuès-les-Bains …" |
| Lines 20 and 22 | You mention phases of exhumation for footwall and hanging wall. If I compare the numbers (30-24 Ma, footwall, versus 35-30 Ma, hanging wall), the logic consequence would be that the fault has changed from a thrust fault to a normal fault with time. It also means that the background cooling signal at the fault should be different on both sides of the fault. This point is not taken up in your discussion. |
| Line 21 | "little evidence" instead of "few evidences", and then "has" instead of "have" |
| Line 23 | I think you mean "extent" rather than "distribution" |
| Line 24 | You claim that the data from this study provide "new constraints on the tectonic uplift of the footwall and hanging wall massifs". I do not see such a discussion and corresponding conclusions in the paper. I also refer to the fact that these issues have already been addressed in the Terra Nova paper. Here you should restrict yourself to what is new in THIS paper. |
| Line 27 | "Global" instead of "World" |
| Line 31 | Start sentence with "Heat production…" |
| Line 31 | Suggestion: add "e.g." after "provided" |
| Line 33 | Add comma after "Anatolia" |
| Line 33 | I assume that the here quoted temperature is a "downhole temperature", if so, add "downhole" |
| Line 41 | "along" instead of "around" |
| Line 42 | Suggestion: "host rocks and the geometry of the fault zone" |
| Line 43 | "mineralization" or "mineralisation" (see line 46)? |
| Line 45 | Suggestion: "In places where no heat flow data are available…" |
| Line 45 | Bracket: "(no boreholes)" |
| Line 46 | "tuff" instead of "tuffa" |
| Line 46 | The "however" at the end of the line could be deleted. It is not really a counterargument… |
| Line48 | Start sentence with "The past decades revealed …" |

| | |
|---|---|
| Lines 51-53 | Suggestion: "In this study, we propose an extended analysis of the (U-Th)/He system in apatite (AHe), sensitive in a temperature range between …." |
| Line 53 | Can you check the quoted temperature range? You quote two references, but these references quote no range from 30 to 90°C. other people quote a smaller range of 80 to 40°C. |
| Lines53/54 | Suggestion: "in association with Rare Earth element (REE) analysis on dated apatites" |
| Line 54 | The term "hidden thermal system" is used frequently, but is not self-explaining, as you obviously refer to a geothermal system in the subsurface. This should be explained, as this is important to understand your reasoning in the Discussion chapter (see e.g. lines 469 or 492). |
| Line 55 | Suggestion: "In a previous study (Milesi et al., 2019), we showed that …" (see also comment to line 56. |
| Line 56 | "AHe age scattering" |
| Line 56 | Suggestion: Use "illustrate" or "prove" instead of "evidence" |
| Line 56 | "In this previous study" (see comment to line 55) |
| Line 57 | "The present study …" |
| Line 58 | "to test these tools", as you refer to AHe dating and REE analysis |
| Line 63 | It is not clear what you mean by "main part". Does this mean "central part", "inner part". |
| Line 63 | "of the Pyrenees" |
| Line 66 | "were (re-)activated" |
| Line 67 | delete bracket after "Later," |
| Line 68 | "led to the formation" |
| Line 68 | "such as the Cerdagne …" |
| Line 71 | Figure 1: In figure B, the white abbreviations "C. b" and "R.b" should be changed to "CB" and "RB" as mentioned in the figure caption. Figure C: I note that four samples from Maurel et al. (2008) in the Mont Louis granite (and one in the Conflent basin) are shown as black squares instead of grey squares (see legend). The rectangle shown with dashed white line is probably the area shown in figure 5A. If yes, refer to it rather than "Our study". Legend: "Glacial" instead of "Glaciary", add "basins" after "Cerdagne". |
| Line 72 | Suggestion: "the outline of the Pyrenees are shown …" |
| Lines 75/76 | Suggestion: "C) Local map with locations of previously published AHe samples" |
| Line 76 | "represented by drops" |
| Line 77 | Suggestion: The Têt fault (eastern Pyrenees) is 100 km long and runs across …". I did not understand the necessity of the word "accident"… |
| Line 77 | "Palaeozoic" instead of "Palaezoic" |
| Line 77 | The here mentioned "Axial Zone" has not been introduced. Important? |
| Lines 80/81 | Suggestion: "provide important age constraints to the regional thermal history…" |
| Line 81 | "of the eastern Pyrenees" |
| Line 83 | Here, for the first time, the term "hanging wall" is mentioned in the text after the abstract, but the reader has no idea from your introduction, which side is the hanging wall, as you have not described the geometry of the fault in 3D. Thrust fault, normal fault, nothing was said about this. |
| Line 85 | You may delete the word "increasing", as you talk about a spectrum. |
| Line 86 | "Pyrenean" instead of "pyrenean" |
| Lines 86/87 | "This is consistent with fission track and zircon (U-Th)/He ages …" |
| Line 89 | Figure 2: Strictly speaking, the modelling window stops at 40 or 30°C as the (U-Th)/He method is no longer sensitive to temperature changes below 40 or 30°C (dependent on what reference you are choosing, see comment to line 53) |
| Line 90 | Suggestion: "of the Caranç a and Canigou massifs (Fig. 1C) in the …" |
| Line 99 | Not clear what you mean by "with a differential estimated at …". Do you refer to a "difference in total exhumation"? Please, clarify! |

| | |
|---|---|
| Line 100 | The statement "was mainly exhumed, and rapidly cooled," is surprising. What we measure with thermochronology is "cooling". And this cooling can be interpreted in terms of exhumation. Here, it sounds as if it would be the other way around (which could be the case if exhumation is modelled and the corresponding cooling due to erosion estimated by the model). I would suggest to refer to the original citations in the quoted papers. |
| Line 100 | "in relation with the Têt fault" |
| Line 102 | It is not clear what you mean by "from a large consensus". It sounds as if the community has debated on this, which would mean that papers could be cited. |
| Line 102 | "but the last period" |
| Line 105/106 | The statement "It must be stressed that some authors consider that any vertical displacement occurred …. First of all, you only quote one source for this (Petit and Mouthereau, 2012), which looks strange if you state "some authors". Then "any vertical displacement occurred" is not clear. Suggestion: "Petit and Mouthereau (2012) suggest that vertical displacement occurred during …" |
| Line 110 | Suggestion: "on the basis of multi-disciplinary (Gunnell et al., 2009) and thermochronological studies (Fitzgerald et al. 1999)." |
| Line 111 | Suggestion: "Today, the Têt fault shows no evidence of …" |
| Line 113 | "suggests incision rates in the range of 1 to 25 m/m.y. …" |
| Line 116 | "surrounded" instead of "characterized"? |
| Line 121 | You should explain what you mean by "a multi-core pattern". Is this the same as "a hot spring cluster"? |
| Line 125 | The term "half-thickness" is not clear. Do you mean the distance from fault to outer rim of the DZ? It would be important to know, whether you refer to a horizontal distance of to normal distance to the fault. Depending on the geometry of the fault plane and the local topography, this might cause significant differences between horizontal and normal distance to the fault. |
| Line 130 | The term "fault rocks" comes unexpectedly. So far, we were talking about a fault and surrounding rocks. Please, explain or rephrase. |
| Line 141 | Suggestion: "may be related to the occurrence of impermeable metasediments in the hanging wall at the surface" |
| Line 143 | The here cited "Prats fault near St-Thomas" is not shown in figures 1 and 3. So, how important is it? Note also that you commonly use a hyphen between "St" and "Thomas". |
| Line 144 | Do you mean "can also increase permeability and localise channelized fluid upflow" Or: "can also increase permeability and localized upflow of channelized fluid"? |
| Line 144 | Figure 3: "Têt Fault" or "Têt fault" (see e.g. "Py fault" in this figure and writing in figure 1)? Should you use same colours as in figure 1? What is the meaning of the green zone in the hanging wall? What is the meaning of "CMNC"? |
| Line 147 | Suggestion: "Figure 4 shows the numerical modelling results of …" |
| Line 148 | ", which takes…" |
| Line 148 | "discontinuities, but does not …" |
| Line 153 | "Gallinàs peak" is shown in figure 4, but also in figure 5 as "Puig Gallinàs" with a noted elevation of 2624 m… |
| Line 154 | "At this locality" instead of "In the latter location" |
| Line 155 | The statement here seems to be in contradiction to what is said in line 149. Please, clarify. |
| Line 157 | Suggestion: "suggesting that this area is a recharge zone". It does not make sense to state that you "suggest" that something is "interpreted"… |
| Line 159 | Figure 4: The colours in the scale to the right end with dark blue, but the figure also shows patches where the dark blue changes to white. What do the white patches mean? In addition, the figure shows areas surrounded by a thin black line and it is not clear how you defined the black line. What does the black line separate? |

| Line 159 | Figure caption: I do not understand the numbers at the ned of the line. I assume that these numbers should be "$7.1^{-15}$ m$^2$ and $5.1^{-15}$ m$^2$". Note that I only refer to one digit after the comma, or is the second relevant? Presumably not as the range given her is much larger… |
| Line 161 | "Thuès-les-Bains" instead of "Thuès" |
| Line 162 | Suggestion: "with respect to sample position." |
| Line 168 | Suggestion: "The here presented thermochronological study extends the study area along the footwall Têt fault to the west, up to the potential Planès recharge zone." |
| Line 170 | "in a way different from numerical modelling." |
| Line 170 | "We also incorporated samples from the hanging wall for comparison." |
| Line 174 | "Palaeozoic" |
| Line 175 | "providing" instead of "representing" |
| Line 178 | Suggestion: "the Carança valley (CAR), at the foot of Gallinàs peak (GAL profile) and further west in the…" |
| Line 179 | "which is a topographic high" |
| Line 182 | "we selected the freshest rocks". Is this a good strategy to find the most altered apatites, the strongest impact of geothermal activity, the strongest geochemical changes? I could understand that you would state that you had to obtain samples that you could carry along and that you could make thin sections from. |
| Line 185 | "Thuès-les-Bains" |
| Line 191 | "crossing the St-Thomas hot spring cluster" |
| Lines 191/192 | "The hot springs at St-Thomas …" |
| Line 192 | Reference to figure 4? |
| Lines 193/194 | Suggestion: "The dimension of the DZ in St-Thomas with a width of 700 m is larger than in …of this fault network." |
| Line 195 | The here quoted samples and their elevation are samples from the footwall. |
| Line 197 | "around the St-Thomas hot springs" |
| Line 197 | "at 250 and 400 m distance" |
| Line 200 | "distance" of 835 and 1750 m normal to the Têt fault." See also comment to line 125. Clarify what distance you mean exactly. |
| Lines 201/202 | "are collected at 5, 160, 175, and 1900 m normal distance from the Têt fault, respectively, and…". See also previous comment. |
| Line 203 | delete "also" |
| Line 206 | Figure 5: The weak point of this figure is that it does not fit to figure 15 with respect to the fault lines. One example: In figure 5, a prominent fault line passes just south of Puig Gallinàs, while in figure 4, the peak is shown N of the Prats fault, and in figure 15, it seems that the fault passes north of the peak. It is also interesting to note that some NW-SE Palaeozoic faults are offset along the Têt fault dextrally (e.g. near CAR7), but some go across it (e.g. next to St-Thomas). Knowing that the fault has been active in the Cenozoic, a fault passing through would mean that there was no lateral offset during this faulting. Furthermore, profile (3) should have two faults at its SE end, but figure B does not show these faults. |
| Line 206 | Figure caption: "showing four sampling profiles" (as in figure B, they are called "profiles". Suggestion: "Grey colours show samples from previous…" |
| Line 211 | The title should be "Areas with no hot springs: …" The same would apply to the titles of chapters 4.1.2 (line 315) and 5.1.2 (line 439). |
| Line 212 | Suggestion: "The GAL profile has been placed across the thermal anomaly…" |
| Lines 212/213 | In my print, the reference "Taillefer et al., 2018" has a blue font. Why? |
| Line 213 | Start sentence with "The samples GAL3, …" |
| Line 215 | delete "sample" after "GAL3" |
| Line 216 | "the same lithology as GAL3" |
| Line 218 | "from the Tête fault, respectively (Fig. 5)." I would avoid to use the term "core", as you have not defined such a "core". Or do you mean the DZ instead? |

| | |
|---|---|
| Line 219 | "from the inner DZ" |
| Lines 221/222 | Suggestion: "near the valley bottom at Carança" (if there is a place with this name). |
| Line 224 | The term "retrogression" is commonly used in terms of marine retrogression. I think what you mean is "alteration" (or retrograde metamorphic overprint?). |
| Line 225 | Suggestion: "fractures at the outcrop scale" |
| Line 226 | Suggestion: "In the PLA profile, two dated samples are located …". Since you refer to an elevation of 1500 m, there are only two samples |
| Line 228 | "fractured with local presence of …" |
| Line 228 | The reference to figure 5 is probably wrong. How can we see iron oxides in figure 5? |
| Line 233 | Figure caption: I am missing a general intro such as "Outcrop and microscopic images of crystalline rocks near Têt fault, eastern Pyrenees". |
| Line 234 | In figure D, I see mainly dark fillings (chlorite?) for the fractures. Do you mean "silicate fillings or do you refer to quartz? |
| Line 235 | Figure E: Again use "quartz" instead of silica", as the optical properties clearly indicate the presence of quartz. According to text, this should be sample CAR7, if yes, add this information to all figure parts. |
| Line 237 | "fractured" instead of "fracturated" |
| Lines 238/239 | "(enlargement of dashed rectangle in figure H)" |
| Line 243 | Suggestion: "mineral concentrates were gained by…" |
| Line 243 | "with no evidence" |
| Line 244 | "for apatite grain photos" |
| Line 245 | Start sentence with "Each single grain was packed in …" |
| Line 247 | "achieved by" instead of "with" |
| Line 253 | add comma after "procedure" |
| Line 255 | "were analysed in-between four …" |
| Line 263 | Does "mn" means "minute"? If so, better use "min" |
| Lines 268/269 | Here, the abbreviations LREE and HREE are introduced. You should then use them systematically (see further comments below). |
| Lines 271/272 | I am not sure whether this sentence should be included. If yes, start with "Caused by technical problems, some apatites have not…" |
| Line 274 | "whether" instead of "if" |
| Line 276 | Do you mean "modelled" instead of "modelised"? |
| Lines 280-288 | In this paragraph I am missing a description of the AHe ages… |
| Line 281 | "is observed" instead of "can be evidenced" |
| Lines 282/283 | "a marked EU anomaly (Fig. 8) |
| Lines283/284 | Suggestion: "… 1999). For fine-grained gneiss samples of the GAL profile, this anomaly is less pronounced." |
| Line 284 | "apatites of GAL3 have…" |
| Line 284 | "consistently" instead of "consistent" |
| Line 285 | "and flatter REE patterns" |
| Line 286 | Suggestion: "are attributed to co-genetic growth of epidote, …" |
| Line 287 | "in the GAL samples" rather than "in our samples", this would be more specific. |
| Line 288 | The term "at different scales" is difficult to understand. What scales are you referring to? |
| Line 289 | Figure 7: In profile GAL, there seems to be a difference with the distances from the Têt fault for the samples of the hanging wall with respect to figure 5. The labels and numbers along the axes and the font in the legends are on the edge of readability. One may simplify by labelling the axis only once on the left side and bottom of the figure. |
| Line 290 | Figure caption: I think, what you define here for the grey area is the outer damage zone (as the inner is yellow). I would start the sentence with "An inner DZ (yellow)…" |
| Line 291 | "for the TET and GAL profiles" |

| Line 291 | The reference to Milesi et al. (2019) is only half way correct. As written, the reader has to believe that Milesi et al. (2019) also contains samples from the GAL profile, which is not true. |
|---|---|
| Line 293 | "analysed" instead of "dated" |
| Line 295 | "and are younger" |
| Lines 296/297 | Suggestion: "… (Milesi et al., 2019). Age dispersion is evident for this sample." |
| Line 297 | "of the now analysed grains" |
| Line 299 | Suggestion: "age dispersion from … to …" |
| Line 304 | Suggestion: "grains in each sample (ST15 and ST16, Table 1)." |
| Line 304 | Start next sentence with "The gneiss samples …" |
| Line 306 | add comma after "respectively" |
| Line 306 | delete "comprised" |
| Line 307 | Here, the reference should also be table "1" instead of "2". |
| Lines 307/308 | The statement "that could be indicative of the occurrence of optically undetected tiny Th-rich inclusions within these apatites" is weakly supported. It is a possible explanation, but you refer to this sample as being a paragneiss, and in paragneisses, the apatites could be of very heterogeneous origin. Thus, the explanation you give could be true for any of the paragneiss samples in your data set. In addition, the initial REE patterns of these apatites could be very different. |
| Line 310 | The statement "A mean age of … is retained for sample ST2." Is unusual. The statement is only given for this sample, while for no other sample. Delete sentence? |
| Line 315 | Title should be "Areas with no hot springs: GAL and PLA profiles, Car valley" (see also comment to line 211). |
| Lines 316/317 | Suggestion: "… are fine gneisses from outside the outer Têt fault DZ and show both limited AHe age dispersion with … and …, respectively." |
| Line 317 | The "However" is not needed, there is no contrast to be mentioned. |
| Line 320 | "HREE" instead of "heavy REE" |
| Lines 321/322 | I have serious doubts about the statement "geochemical and paragenetic conditions of growth during medium grade metamorphism of pelitic rocks". I have checked the study of Henrichs et al. (2018), and my impression is that they talk about new-grown apatites rather detrital ones. At least they refer to the fact that they see a change in REE mobility starting with the uppermost greenschist facies. We do not know, however, what the metamorphic grade of the here sampled gneisses is. Thus, I have two critical questions that should be answered in the paper, if you want to refer to Henrichs et al. study: First, what is the metamorphic overprint your gneisses (in particular the paragneisses) have undergone during Pyrenean metamorphism? Second, do we talk about detrital or new-grown apatites (and what does the Henrichs et al. study talk about)? |
| Line 323 | I would start the paragraph with "Sample CAR7 from the …", because all CAR samples come from the Carança valley, so this information is redundant. |
| Line 327 | "observed for apatites" |
| Line 329 | "Further" instead of "Farther" |
| Line 334 | No comments to Figure 8 and its caption |
| Line 336 | Table 1: The caption (or "Note" line below the table should explain the meaning of "Ft". I do not understand, why for some samples, you report mean ages, while for others, no mean age is given. What is the argument behind the selection? The Ft value for TET8-5 is given with three digits, while all other Ft values only with two digits. The Th/U value for TET1-12 is given as "<0.1" (see also "<0.1" for the age "Error". According to the data for U and Th, the value should be 0.02, if one uses one more digit, similar to ST4-1, where you report a "Th/U" value of "0.0", and the ratio derived from the data would be 0.02 as well, if expressed with two digits after the comma. Since the number of digits is dependent on the data for U and Th, the number of digits may vary, and I would make a difference. |

| | |
|---|---|
| Line 342 | "Apatites from the hanging wall" |
| Line 343 | The statement suggests that you believe that your samples are "S-type granites". Can you clarify this? |
| Line 344 | The subchapter number should be 4.2.1, and I would recommend to adapt the title to line 292. |
| Line 345 | "In the TET profile, …" |
| Line 349 | I would recommend to quote the age (14.8 ± 0.7 Ma). |
| Line 351 | "In the ST profile, …" |
| Line 355 | "mainly by lithology" |
| Line 358-361 | I would argue that this paragraph is at the wrong place. Sample ML1 is closest to the GAL profile (Fig. 5). |
| Lines 360/361 | Suggestion: from the same sample, and a mean AHe age from granite ST12 …" |
| Line 362 | The subchapter number should be 4.2.2, and I would recommend to adapt the title to line 315. |
| Line 367 | For completion, I would add a sentence such as "No REE patterns have been determined for these samples." (in reference to lines 271/272, I guess) |
| Line 368 | Suggestion: "A single gneiss sample (PLA5) from ..:" |
| Line 370 | Suggestion: "exhibits a REE pattern with constant LREE values (Fig. 8) |
| Line 370 | The statement "due to the presence of Th-rich inclusions" cannot be valid, as grain PLA5-4 has a very low Th value of 3.6 ppm. See also comment to lines 307/308). |
| Line 371 | The bracket could be reduced to "(Table 2)", as the grain number is already mentioned two lines above. |
| Line 371 | Suggestion: "Consequently, this apatite grain was discarded from further discussion." |
| Line 372 | No comment to figure 9 and its caption. |
| Line 375 | Table 2: Here again, you should clarify the argument on which you report mean ages for some samples and not for others. And you should explain the abbreviation "Ft" somewhere (either caption of "Note" line at the bottom). |
| Line 378 | Suggestion: "…DZ, we combined these data with those from samples outside the …" |
| Line 379 | "as a reference" |
| Line 379 | It is not clear, what you mean by "accurate ages". "Accurate" with respect to what? |
| Line 379 | "low" instead of "weak" |
| Line 380 | "variation" instead of "variations" |
| Line 381 | "S-type granite", no empty space after "S" |
| Line 382 | Again, the reference to Henrichs et al. (2018) seems to be weak. Is their statement that all apatites in paragneisses show the same REE patterns? At least, they note that a homogenization does depend on the metamorphic grade. And if the input of REE along with detrital apatite varies strongly at the beginning, the REE patterns must look differently, even if the apatites homogenize their REE in medium- to high-grade paragneisses. Furthermore, the amount of new epidote, which depends on the amount of Ca available in the paragneisses will also influence the redistribution of REE. Thus, my recommendation would be: Please, read the Henrichs et al. study carefully, and then decide, whether it is worthwhile citing it. |
| Line 382 | "We then define …" |
| Line 383 | "for all other profiles" |
| Line 387 | Figure 10: Several of the ellipses in these plots are so tiny that one cannot see the colour inside. Thus, I suggest to the authors to make the triangles bigger, they may even overlap, as the ST profile data are concentrated in the centre. |
| Line 388 | "… profile samples." |
| Lines 392/393 | Suggestion: "…, we identify three fields that overlap partially:" |
| Line 396 | "above 20 Ma, which are interpreted as …" |
| Line 400 | Suggestion: "observed, more pronounced along …" |
| Line 401 | "similarly" instead of "similar" |

| Lines 401-403 | The statement "depleted REE patterns compared to apatites sampled outside the DZ" is not clear. Do you suggest that the apatites outside the DZ have undergone REE depletion? Aren't the apatites outside the DZ the reference material for the apatites within the DZ? The confusion for me gets worse, when you talk about "global depletion" in line 402. What do you mean by "global"? If all REE patterns are depleted, what reference do we have? I assume that there is a misunderstanding caused by the current text, which should be fixed by some re-writing. |
|---|---|
| Line 403 | Suggestion: "Note, however, that along the ST profile, …" |
| Line 404 | "consistent" instead of "consistently" |
| Line 405 | "do not exhibit REE depletion" |
| Line 406 | Suggestion: "Due to the small numbers of analyses, the result is not interpreted any further." Is this what you mean? |
| Lines 407/408 | This is the first time that you refer to Flowers et al. (2009) and Gautheron et al. (2009) and state that the intra-and inter-sample age dispersion cannot be simulated. Here, the statement is correct, as you leave the door open, that a simulation would work, if the single apatite grains would be modelled individually. There are, however, places in the text (see e.g. lines 503/504), where the wording has to be changed. Nevertheless, for this passage, I would recommend to change to "cannot be simulated by a common thermal history using existing diffusion models for apatite" |
| Lines 408/409 | Suggestion: "In particular, the young AHe ages do not fit to the regional …" It is not the fact that there are young ages, but the spread in ages that does not allow a common thermal history to be modelled. |
| Lines 409-411 | Suggestion: "In accordance with the results for samples in the outer DZ of the TET profile, they raise questions concerning the origin of the AHe age scattering near the hydrothermal circulation zones (Milesi et al., 2019)." |
| Line 411 | "were subject to" |
| Line 412 | Suggestion: "enhances $^4$He diffusion and apatite rejuvenation rather than U-Th incorporation, …" |
| Lines 411/412 | The statement "Which is inconsistent with the observed REE depletion" cannot be assessed, as there are some intermediate steps missing in our argumentation. The claim is; I guess, that REE depletion would run parallel to U and Th depletion. Is this true? And what is the argument for it? I do not doubt the argument, but is has to be stated. |
| Line 415 | "reaction zone" instead of "reacted zone"? |
| Line 416 | "have been carried away"? |
| Line 416 | "lattice of reacting apatite" |
| Line 418 | "to be lost from the host apatite" |
| Line 418 | The statement "mainly by advection during hydrothermal alteration processes" Is not sufficiently precise. Within the apatite, the process is diffusion, outside of the apatite, it is advection. |
| Line 420 | According to the reference list, it should be "Andrews and Lee, 1979" |
| Line 420 | Start sentence with "Variable REE loss" |
| Line 422 | "at thin-section scale" |
| Line 423 | After "fluid interaction" you could add a reference to figure 6H. |
| Lines 423/424 | Start sentence with "The chemistry and temperature of the fluid flow may change with time", then add a comma after bracket, and continue with "…, however, it is .." |
| Line 425 | "prominent" instead of "important" |
| Lines 425/426 | "hot springs than near St-Thomas (…)" |
| Line 426 | "pronounced" rather than "important" |
| Line 427 | "larger" instead of "greater" |
| Line 429 | "We cannot exclude an impact" |
| Line 430 | "at the St-Thomas areas" |
| Line 430 | Suggestion: "which may be responsible for an increase of rock permeability" |

| | |
|---|---|
| Line 431 | "heat dispersion into" |
| Line 432 | observed within the" |
| Line 432 | Suggestion: "This is independently supported by the …" |
| Line 433/434 | One may add references to figure 4 after "Têt fault" in line 433 and to figure 5 after "Têt fault" in line 434. |
| Line 434 | "that are all located adjacent to" |
| Line 435 | Figure 11: In the caption you refer to "blue" and "purple field"s. In my print-out, the "blue" and "purple" seem to be of the same colour. What about having the diagrams on the right with the same x-axis scale? |
| Line 435 | Figure caption: "for the TET and ST profiles" |
| Line 436 | "For the TET profile, " |
| Line 438 | The "consistent" again raises the question of the reference. "consistent" to or with what? How about using "elevated" instead? |
| Line 438 | "contents" instead of "patterns" |
| Line 438 | "The depleting of REE is …" |
| Line 439 | Better use title "Area with no hot springs: …" (see also comment to lines 315 and 362) |
| Line 440 | Suggestion: "Slightly rejuvenated and scattered AHe ages and depleted REE patterns are also obtained for…" |
| Line 441 | "hot spring clusters: Carança valley (CAR) and Planès profiles (PLA, Figs. 12 and 13A) |
| Line 443 | Again, some of the ellipses are very small, so that the colour inside is hard to see, one may shift the triangles so that they overlap with each other. |
| Line 444 | "GAL profile samples" |
| Lines 447/448 | Here, the wording has to be adapted to be more precise. It is not the fault of the diffusion models that you cannot simulate a common thermal history. For wording, see comment to lines 4087/408. The diffusion models work ok, if applied to one single grain, but they cannot come up with a common thermal history. |
| Line 449 | "and young ones peculiarly": Here, again, it is not the young ages that cause the problem, but the spread of ages. The young grains have a correct thermal history and the old ones as well, but it is not the history of a closed system, but of in-growth or loss of He. |
| Line 451 | "no longer active" |
| Line 452 | "circulation" instead of "circulations" |
| Line 456 | Do you mean "the end of surface hydrothermal activity"? |
| Lines 456/457 | Suggestion: "… recent, while a hidden geothermal system could still exist nowadays in the subsurface. |
| Lines 458-461 | Figure 13 and caption: Several details seem to be at odds with this figure: First, the start of the caption is equal to figure 11, but these are NOT the samples from "within the DZ" as stated in Line 459. Second, in lines 460/461, you state that the purple filed and triangles correspond to samples outside the DZ. Why is there no purple field for the GAL profile, three triangles for this diagram, but no triangles for the Car and PLA plot? Third, you mention in line 462 that there is an associated younging of AHe ages, but this is only visible for the CAR samples, not for the PLA and not for the GAL either. This caption has to be revised completely, I propose. |
| Line 466 | "do not fit to the regional ..:" |
| Line 466 | "depicted in Figure 2." |
| Line 467 | The statement "which is also inconsistent with the numerical thermal modelling" is problematic. The modelling in figure 4 shows the situation as of today, but the AHe and REE data integrate over the entire geothermal activity period. The hydrothermal activity might have stopped at 13 Ma. |
| Line 468 | I am not sure, what you mean by "very circumscribed", do you mean "more local" or "more regional"? |
| Line 469 | "St-Thomas" |

| Line 470 | "might then represent potential …" |
| Line 472 | "we note" instead of "we can note" |
| Line 473 | reference to figure 13 or 13A? |
| Lines 474/475 | I would delete the first part of the sentence "This does not support … since 10 Ma", and connect the rest to the sentence before: "… within the DZ in the last 10 Ma, in contrast to what …" |
| Line 477 | "circulation" instead of "circulations" |
| Line 477 | "Moreover, thermal modelling …" |
| Line 479 | "suggests rapid cooling" |
| Line 483 | "supports" instead of "reinforces" |
| Line 485 | "using" instead of "made with" and then "and" instead of "using" |
| Line 487 | "suggests an" instead of "corresponds to an" as these represent no unique solutions. |
| Line 489 | "even at thin-section scale" |
| Line 490 | You are mentioning "He loss", but could it also be gain to explain the unusually old single grains next to the fault? |
| Line 490 | I think what you mean is not "actual" but "present-day hot springs" |
| Line 492 | Start sentence with "The combination of ..:" |
| Line 493 | "extensions" |
| Lines 494/495 | Suggestion: "This result questions a straight-forward interpretation of AHe ages, if potential …" |
| Line 496 | Figure 15: Should the scale of intensity (top right) not be called "intensity of present-day thermal anomaly"? For the "outcrop scale": "Poorly fractured gneiss lenses" |
| Line 497 | Figure caption: Start text with "Synthetic 3D block of …" |
| Line 497 | Suggestion: "of the present-day thermal anomalies" |
| Line 499 | "samples" instead of "sample" (as there are ML1 and ST12) |
| Line 499 | add reference to Table 2 after "AHe ages"? |
| Line 501 | "S-typ granites" instead of "granitic lithology"? |
| Line 502 | "similar to" instead of "as in" |
| Line 502 | "closest to" instead of "nearest" |
| Line 503 | Suggestion: "The AHe age scatter (Fig. 7, Table 2) cannot …" I am sure it should be figure 7 instead of figure 3… |
| Lines 503/504 | Here again, the wording has to be adapted to the comments to line 407/408. |
| Line 505 | "Apatite REE contents appear less …" |
| Lines 506/507 | I do not understand the statement on the Canaveilles hot spring cluster in the hanging wall. The hydrothermal anomaly on the other side of the fault is much closer. You draw a picture of a completely impermeable fault plane, which for several clusters and profiles is shown to not be true. I would rather go for a picture, in which the fault plane is permeable, in particular along existing fractures systems that cross the Têt fault. By such a picture, it is very easy to also explain the Canaveilles hot spring cluster, which is located very close to a prominent fault linked to the Têt fault (Fig. 5). Of course, these neighbouring faults may have undergone some reactivation and local increase of permeability, even if surrounded by less permeable metasediments. Such lenses of metasediments (Fig. 5) may even help to channel and concentrate fluid flow. |
| Line 507 | "just NE of …" |
| Line 509 | "those close to the Têt fault" |
| Line 511 | Here again, I would suggest a rewording to better clarify what you mean by "can be properly modelled with He diffusion models in apatite". It is not the models that are wrong, but it is the fact that even within one sample fluid circulation may have led to very different thermal histories. |
| Line 512 | "outside the DZ" |
| Line 513 | "shows rapid cooling" |
| Line 514 | According to figure 16, it should be "range of 150 to 50°C" (perhaps even 40°C). |

| | |
|---|---|
| Line 514 | "slows" instead of "low" |
| Line 514 | "Low" instead of "slow", as this time, you refer to "rates". |
| Line 514 | Suggestion: "may account for" instead of "can account for" |
| Line 516 | "The PLA5 sample" or "Sample PLA5" |
| Line 517 | Suggestion: "lower than the geothermal system along the Têt fault footwall …" |
| Lines 518/519 | The last sentence suggests that everything can be explained by regional slow cooling. Does thie mean that geothermal activity is no explanation for the scattering? Why not? See, e.g., hot springs in Canaveilles on the other side of the fault. |
| Line 522 | Suggestion: "corresponds to a perfect match." |
| Line 523 | delete "mainly" |
| Line 525 | "even though" instead of "despite" |
| Line 525 | I think the statement that the small numbers do not allow "a straightforward interpretation" is not the issue. The small number allows many interpretations to be supported by the data. The low number of data is not sensitive enough to discredit many of them. So, my suggestion would be "does not allow a clear" or "unique interpretation". |
| Line 525 | "questions" instead of "question" |
| Line 527 | Do you mean "all samples outside the DZ". If not, I do not understand the statement… |
| Line 527 | "thermal history together with rocks ..:" |
| Line 528 | "perturbation, in accordance with …" |
| Line 529 | "that hydrothermal circulation is mainly restricted" |
| Lines 529/530 | I would disagree with this statement. First, several profiles show the influence of the geothermal activity on AHe ages also in the hanging wall. Furthermore, the Canaveilles hot spring cluster clearly indicates that there are hot springs on the other side. Even if you suggest a completely impermeable fault plane, AHe ages might be influenced by re-activation of fluids on the other side of the fault by the increase in temperature in the rocks of the footwall. |
| Line 531 | "influenced" rather than "impacted" |
| Line 532 | Start sentence with "We also highlight …", "in addition" and "also" are redundant. |
| Line 532 | "interaction" instead of "interactions" |
| Line 533 | The here quoted temperature values of "<130°C" has not been quoted in the paper. It is important? If yes, you have to explain, where this temperature has been derived from. |
| Line 533 | "considered a closed system" |
| Line 534 | "released into" instead of "dissolved". The transport within the fluid is not the critical process. |
| Lines 534-536 | I would not make these statements here. First, you have not discussed the processes that caused He gain and loss and REE loss in this paper. The conclusions should not come up with new (not discussed) issues. Second, I would suggest that the processes could be addressed in a next paper, if you have evidence or modelling results to distinguish between potential processes. But in this paper, the focus clearly is on the hydrothermal activity and how to detect it; there is little evidence that was presented about the processes, certainly nothing about "nano-channels within the crystal lattice". So, consequently take these statements out. |
| Line 536 | "As fluid flow through …" |
| Line 536 | "even at thin-section scale" |
| Line 537 | "Again, the "$^4$He loss by fluid advection" is not the limiting process, but diffusion is much more crucial as Flowers et al. or Gautheron et al. have shown. |
| Line 537 | The "global positive correlation" does not exist. It exists for some of the data sets (profiles), but by far not all of them. |
| Lines 538/539 | Suggestion: Start sentence with "The same process may also account for …" |
| Line 539 | "in other cases" |

| | |
|---|---|
| Line 539 | "fluid circulation" instead of "fluid circulations" |
| Line 539 | "may occur" instead of "took place", I would suggest to make this more general, not only relevant for the past. |
| Line 540 | "questions" instead of "question" |
| Line 540 | Suggestion: "AHe ages in cases, for which potential hydrothermal …" |
| Line 541 | Do you mean "active" instead of "obvious"? |
| Line 542 | "palaeo-geothermal" instead of "paleo- geothermal" |
| Line 543 | Suggestion: "As an exploratory tool, (U-Th)/He thermochronology may be applied complementary to …" |
| Line 544 | "geoelectrical" instead of "electrical" |
| Line 545 | It is not clear, what "models" you are referring to. |
| Line 549 | "by Doriane" |
| Line 550 | "during field work." |
| Line 552 | Suggestion: Data supporting the findings of this study are available in the Supplementary section. Additional details may be obtained upon request …" |
| Line 556 | "models on" |
| Line 557 | "co-wrote" |
| Line 558 | "participated in" |
| References: | I was unable to locate the reference Sutherland et al. (2012) in the text. |
| | Missing in the list is the reference Farley et al. (1996), found in line 254. |
| | The reference Shipton & Cowie is at the wrong place. |
| | I found strange spellings, spaces or upper case words in lines 677, 683, 703, 742, and 771/772 that should be corrected. |

Meinert Rahn, May 17, 2020.

---

## Referee Comment (RC2) · Cecile Gautheron (Referee) · 26 May 2020

**Cecile Gautheron (Referee)**

cecile.gautheron@u-psud.fr

Received and published: 26 May 2020

This contribution presents new apatite (U-Th)/He (AHe) thermochronological data associated with rare earth element (REE) content in the same dated apatite along four transects cutting the Tet fault in the Pyrenees (France). The aim of the papers is to investigate the possible impact of geothermal fluids on affecting AHe age, using the evolution of REE along the samples.

The ms presents interesting new data on two mains zones (free fluids and hot fluids

zones). However, the authors have already published similar contribution in 2019, where they present the impact of hot fluid on AHe data. Only at the very end of the text, the authors show that when hot fluids are not present, it doesn't affect the AHe data. It is difficult to really understand what the message of the contribution is. Is that a methodological paper? If yes, the authors need to go further as the impact of hot fluids in fault affecting the AHe system has been already proposed. In addition, the authors present new thermal history modeling of the area, but they don't really discuss the implications for the eastern part of the Pyrenees. I strongly suggest that the authors propose further investigation on the exhumation of the eastern Pyrenees and compare with the other thermochronological data. Or the authors could investigate past geothermal anomalies and better present the new result of their contribution (AHe data are not affected in area where fluids are absent).

Below are some additional comments:

Abstract. Add France after Pyrenees. The passage with the second sentence of the paragraph is odd. I suggest "in order to investigate the evolution of the geothermal gradient and fluid flow, we used AHe + geochemical analysis to ...

Paragraph 20: the authors gave sentences about exhumation for two zones, without using the data. What is the implication of the results? It is very unexploited.

Paragraph 25: it opens new perspective to what exactly? This could be a good angle to go to deeper investigation in the ms

Paragraph 30: I am not sure to understand why the presence of water is required to heat production. Perhaps, heat advection need water, but water don't produce heat.

Paragraph 50: about AHe temperature sensitivity range go higher than 90°C depending on the damage dose. I suggest adding 40-120°C and add Ault et al 2019 reference

Paragraph 55: the goal of the paper "the study wants to test this tool both in areas lacking of hot fluids ...". Please rephrase better to see what is new in the study. Why
above 60°C?

Paragraph 60: and in general, more recent citations on low temperature thermochronological data of the Pyrenees are missing. Please add other papers than Verges et al. 1995.

Paragraph 65: remove the ) between Later, ) two minor...

Figure 1: Homogenization of the scale bar for each figure (a, b, c) could be good. What is the white square on fig c?

paragraph 80, line 3: add Ma after the numbers 300.3 $\pm$ 3.1 and 291.2 $\pm$ 2.8 Reference of Ar/Ar ages are missing It is ZFT and not ZFt. Please replace

Figure 2: give the elevation or difference of elevation between samples. The addition of the AHe observed / predicted versus elevation could be nice.

Paragraph 85: it should be Apatite (U-Th)/He age yielded a large range of age between ... and not apatite yielded a large range of AHe age. Add error on AHe ages

Paragraph 95: could also add error on Ar ages

Paragraph 100: please correct, it is not the low temperature data that reveal that the Canigou massif was exhumed and cooled but thermal modeling. What do you refer by rapidly cooled? Add values, like that it will be more homogenous with other given exhumation rates (see Paragraph 110)

Paragraph 110: add more recent references about uplift and erosion in the Pyrenees, e.g.:. Vacherat et al. 2014, 2016, Ternois et al., 2019 etc for example

Section 2.2: since when hot fluids are circulating in the tet fault?

Figure 3: color legend is missing. What is the brown line at the bottom, in the footwall?

Figure 4: could the authors add on the figure, the location of the samples? It can be useful I am not really sure about the purpose of this figure. What the authors wants to
show?

Paragraph 145: please add a little more information on how the numerical modeling has been done and with which data

Paragraph 180: the aim of this sampling ... was to track the effects of recent hydrothermal circulation... but what is new in this study? The authors have already published this type of study, so it is important to go further.

Paragraph 210: why is the reference Taillefer et al., 2018, in bleu and italic?

Paragraph 245: because the authors also measured the Sm content, they can use the value to add them and calculate the (U-Th-Sm)/He age

Paragraph 260: the raw REE data should be add in the AHe data table for simplicity. Please add the raw values and not normalized to chondrite.

Figure 7: it will be nice, if the authors add directly on the graphic, the free hot spring and with hotspring.

Paragraph 295: about eU (U+0.234xTh+0.0045xSm; Gastil et al., 1967), the presentation of the value could more simply presented 12 ppm instead of 12.3 ppm etc. What about the variation in the Th/U ratio? It can help to see if the Th and U have fractionated

Figure 8 et 9: both figures are really difficult to read. They look very similar. Perhaps so diagram can go in the supplementary diagram and only light/heavy REE ratio could be presented.

Paragraph 410-415: It will be interesting to add CI measurement or compare the REE data from the apatite to discuss more about the dissolution / recrystallisation process. The reference Zeitler et al 2017 is good, but just is a summary of other studies (Shuster et al., 2006; Flowers et al., 2009; Gautheron et al., 2009; Gerin et al., 2017, Idelman et al., 2018, McDannell et al., 2018), that can be cited or for simplicity the reference Ault et al 2019 resumed all of other recent studies.
Figure 11: are you sure about the sum of REE because 20000 ppm is 20%. It seems that it is chondritic normalized values. Please verify the value as 1000 ppm of Sm is quite a lot and if it is the case, it will strongly change the AHe age. What about AHe age vs Th/U, AHe age vs light/heavy REE ratio? It will be better to add also the data outside of the DZ rather than the purple square

Figure 13: same comment than for Fig 11

Figure 15: scale of the microscopic scale seems not be correct as the apatite crystal are really too small, or the scale

Paragraph 525: the last line that describe the fact that if not fluids are shown, it doesn't impact the AHe ages can be the angle of the paper. This is the main new result of the paper and it appears at the end. It is a shame, because it is very interesting. The authors could present this results in the beginning, and not focus to much about the influence of hot fluids, as it has been already published.

Supplementary: please report the raw REE content in ppm

Table 2; you can add directly the REE measurement in the same table

SED

---

## Author Comment (AC1) · 30 Jun 2020

**Point per point reply Reviewer Comments 1**

The submitted manuscript and supplementary material presents 99 new (U-Th)/He single grain ages and 63 REE spectra on apatites from the geothermal fields along the Têt fault in the Pyrenees. The study is closely linked to the recently published study Milesi et al. 2019, which already presented a restricted data set from the TET profile, a study that here is extended by new samples and samples along more profiles (STA, GAL, PAL) and a small number of more isolated samples. Samples were taken from the hanging wall and footwall of the fault at variable distance, divided into samples inside a deformation zone (DZ) around the fault and outside of it.
The study concludes that He ages and REE contents can be distinctly influenced by hydrothermal activity, mainly dependent on rock permeability, fault activity and thermal evolution. Apatite ages closest to the fault show a large age scatter and strong depletion of REE contents if compared to samples far away from the fault, which show the original REE patterns and the regional cooling history (as a result of exhumation and differential uplift along the fault).
Ages and REE support the conclusions and I see no substantial or harsh mismatches, even though the reader is left behind by some question marks concerning the ideas of the authors of the processes taking place. Since I have not major drawbacks; I propose that the paper needs minor revision of the text.

Points that the authors should have a closer look to are the following:

1. Normal or horizontal distance to the fault plane: The authors should clarify in their paper, whether the cited distances are horizontal distances or distances normal to the fault plane.

   We now present both horizontal and normal distances. The dip of the Têt fault can be measured at Thuès-les-Bains (TET) and in the Carança valley (CAR), so there the normal distance can be easily calculated. For GAL, ST and PLA profiles, normal distances can be estimated according to the mean dip measured further east. To be consistent, damage zone thicknesses were also recalculated according to the normal distance. Then now, to be complete, we indicate both horizontal and normal distances in tables 1 and 2 based on the dip of the Têt fault but we use horizontal distance in the manuscript. This is now indicated lines 134-135 and lines 207-208.

2. When the authors refer to Flowers et al. or Gautheron et al. diffusion models, then they should carefully choose their wording. The diffusion models are ok, but it is the fact that the AHe ages scatter too much to be modelled by a common thermal history.

   We took this comment into account and rephrased the sentences (see lines 422-423, 463-465, 522-526, 533-534 and specific comments below).

3. The model the authors present for the fault plane suggests that the plane seems to have clearly restrict geothermal activity to the footwall. However, there are faults that crosscut the Têt fault into the hanging wall, and AHe ages next to the fault show considerable age scatter. So, the influence reaches beyond the fault plane. Note, that the rocks are by far not impermeable, in particular if they have undergone tectonic shearing.

   We agree, although the larger fluid flow is into the footwall as shown by the number of hot springs (Taillefer et al., 2017) and fluid flow numerical modeling (Taillefer et al., 2018), but also the size of the thermal perturbation recorded by thermochronology (this study). First of all, it is well known that the hanging wall scattering of ages is larger than in the footwall due

to the lower exhumation rate (Maurel et al., 2008). This is particularly well seen in St-Thomas profile. Secondly, we cannot exclude a contribution of fluids from the footwall relief as also suggested by Taillefer et al. (2018), especially in the Têt profile since REE patterns are depleted, which is not the case for the other profiles. REE global depletion is mainly observed in the footwall block and only in sample TETHW from the hanging wall, in the vicinity of the Thuès-les-Bains hot springs. It is also worth considering that Têt fault gouges are not a perfect impermeable barrier for a series of reasons (crystalline rock juxtaposition, composition of the fault core, fault intersections) presented and discussed in Taillefer et al. (2017). This is now better explained in the manuscript (lines 526-531).

4. The reference to Henrichs et al. (2018) to explain a common apatite REE pattern is problematic to me. First, you do not give any information about the degree of metamorphism, second, the paragneisses should contain apatites of different origin and different REE patterns in the beginning. Does the Henrichs et al. study refer to new-grown apatites or detrital apatites? Quoting this study should be checked again. In general, we do not need to know the original REE pattern, we simply need to have differences, but of course, in a paragneiss, there might not be a unique reference REE pattern.

   We agree with the comment and now we just mention that the apatite protolith is different (see lines 304-305). Let's just precise that these apatites are new grown apatites formed during the variscan barrovian metamorphism at temperatures close to 600°C (Hoÿm de Marien et al., 2019). They appear unaffected by a pyrenean metamorphic overprint, the effects of which are mainly localized along shear zones.

5. The authors do calculate mean ages for some samples, for others they do not (Tables 1 and 2). They should clearly state on the basis of what they decide that a sample has internally consistent ages, but another one shows age scattering. Methodically, this is not clear yet, but would be very important to clarify the next steps. In addition, this may lead to the possibility to independently talk about the thermal history of the footwall on one side and the hanging wall on the other. The authors point to this in their abstract (lines 24/25), but the paper does not discuss the individual thermal histories. Perhaps, it would be wise to do this first.

   The calculation of mean ages has been deleted in Tables and text. Only individual ages are considered for QTQt modelling. We have improved the discussion based on the thermal histories of both the footwall and the hanging wall (lines 497 to 504 and 553 to 556). However, this not the main topic of the paper and we do not develop further the exhumation history of eastern Pyrenees. This will be the topic of a new paper in progress.

6. If there is one chapter that I would recommend strong rewriting for, it is the Conclusions chapter, which at the moment caused the highest density of question marks to me.

   The conclusion has been completely rewritten to highlight the main results of this work (see lines 558 to 588).

7. I highly recommend to the authors to not speculate about the processes that have taken place and led to AHe age rejuvenation or increase or REE depletion. There is little evidence presented that would be able to narrow down potential processes that took place. Such work could be done on the same samples, but not based on the her presented evidence.

   We have toned down the too speculative sentences in the manuscript as suggested, and also have revised the conclusion and the discussion in that sense.

**Detailed comments:**

Line 12 "around" instead of "nearby"

Line 12 Since you are mentioning the two hot spring clusters for the first time; I would suggest to shift the bracket from line 17 right after "clusters"

Line 12 "in-between"

Lines 12/13 delete "in an attempt"

Line 16 to be concordant with line 12: "nearby the two hot spring clusters"

Line 17 "resettings" instead of "resetting", we do not know whether the reset occurred at the same time.

Line 18 "around the Thuès-les-Bains …"

All done (lines 12 to 19).

Lines 20 and 22 You mention phases of exhumation for footwall and hanging wall. If I compare the numbers (30-24 Ma, footwall, versus 35-30 Ma, hanging wall), the logic consequence would be that the fault has changed from a thrust fault to a normal fault with time. It also means that the background cooling signal at the fault should be different on both sides of the fault. This point is not taken up in your discussion.

We agree with this remark. During the Eocene collision, the northern units of the hanging-wall were thrust southward onto the Canigou-Carança massif (Ternois et al., 2019) but it is still unknown if the Têt fault was activated at this time. Nonetheless, the main objective of the paper is not to discuss about the exhumation history of eastern Pyrenees. We have rephrased the sentence (line 20) and discussed and improved the thermal evolution only (see lines 497 to 504 and 553 to 556).

Line 21 "little evidence" instead of "few evidences", and then "has" instead of "have"

Done (lines 21-22).

Line 23 I think you mean "extent" rather than "distribution"

"distribution" has been replaced by "extent" (line 24).

Line 24 You claim that the data from this study provide "new constraints on the tectonic uplift of the footwall and hanging wall massifs". I do not see such a discussion and corresponding conclusions in the paper. I also refer to the fact that these issues have already been addressed in the Terra Nova paper. Here you should restrict yourself to what is new in THIS paper.

We have added more details about tectonics on the basis of thermal modelling (lines 497 to 504 and 553 to 556), and also added a paragraph on the tectonic/uplift history of our studied area in the conclusion (lines 583 to 588). Following the general comment of reviewer #1, we have rewritten the conclusion and highlighted the new results of this work.

Line 31 Start sentence with "Heat production…"

Line 31 Suggestion: add "e.g." after "provided"

Line 33 Add comma after "Anatolia"

Line 33 I assume that the here quoted temperature is a "downhole temperature", if so, add "downhole"

Line 42 Suggestion: "host rocks and the geometry of the fault zone"

Line 43 "mineralization" or "mineralisation" (see line 46)?

Line 45 Suggestion: "In places where no heat flow data are available…"

Line 45 Bracket: "(no boreholes)"

Line 46 "tuff" instead of "tuffa"

Line 46 The "however" at the end of the line could be deleted. It is not really a counterargument…

Line48 Start sentence with "The past decades revealed …"

Lines 51-53 Suggestion: "In this study, we propose an extended analysis of the (U-Th)/He system in apatite (AHe), sensitive in a temperature range between …."

All done (see lines 32 to 53).

Line 53 Can you check the quoted temperature range? You quote two references, but these references quote no range from 30 to 90°C. other people quote a smaller range of 80 to 40°C.

As suggested by reviewer 2, we have changed for a large range between 40°C and 120°C and added a new reference (Ault et al., 2019). This range is larger than the previous one because it considers the damage dose.

Lines53/54 Suggestion: "in association with Rare Earth element (REE) analysis on dated apatites"

Done (Line 54).

Line 54 The term "hidden thermal system" is used frequently, but is not self-explaining, as you obviously refer to a geothermal system in the subsurface. This should be explained, as this is important to understand your reasoning in the Discussion chapter (see e.g. lines 469 or 492).

We propose to replace "hidden" by "blind" in the whole text. This term is widely accepted in geothermal system studies (e.g. Coolbaugh et al., 2006; Faulds and Hinz, 2015, Gorynski et al., 2014) since it does not necessarily refer to a subsurface geothermal system.

Line 55 Suggestion: "In a previous study (Milesi et al., 2019), we showed that …" (see also comment to line 56.

Line 56 "AHe age scattering"

Line 56 Suggestion: Use "illustrate" or "prove" instead of "evidence"

Line 56 "In this previous study" (see comment to line 55)

Line 57 "The present study …"

Line 58 "to test these tools", as you refer to AHe dating and REE analysis

All done (lines 56 to 60).

Line 63 It is not clear what you mean by "main part". Does this mean "central part", "inner part".

"main" has been replaced by "axial" (line 65).

Line 63 "of the Pyrenees"

Done (line 65).

Line 71 Figure 1: In figure B, the white abbreviations "C. b" and "R.b" should be changed to "CB" and "RB" as mentioned in the figure caption. Figure C: I note that four samples from Maurel et al. (2008) in the Mont Louis granite (and one in the Conflent basin) are shown as black squares instead of grey squares (see legend). The rectangle shown with dashed white line is probably the area shown in figure 5A. If yes, refer to it rather than "Our study". Legend: "Glacial" instead of "Glaciary", add "basins" after "Cerdagne".

Abbreviations, legend and colours have been corrected on Figure 1.

Line 66 "were (re-)activated"

Line 67 delete bracket after "Later,"

Line 68 "led to the formation"

Line 68 "such as the Cerdagne …"

Line 72 Suggestion: "the outline of the Pyrenees are shown …"

Lines 75/76 Suggestion: "C) Local map with locations of previously published AHe samples"

Line 76 "represented by drops"

Line 77 Suggestion: The Têt fault (eastern Pyrenees) is 100 km long and runs across …". I did not understand the necessity of the word "accident"…

Line 77 "Palaeozoic" instead of "Palaezoic"

All done (lines 71 to 82).

Line 77 The here mentioned "Axial Zone" has not been introduced. Important?

Now, this nomenclature has been introduced line 65 with references.

Lines 80/81 Suggestion: "provide important age constraints to the regional thermal history…"

Line 81 "of the eastern Pyrenees"

Done (lines 85 to 86).

Line 89 Figure 2: Strictly speaking, the modelling window stops at 40 or 30°C as the (U-Th)/He method is no longer sensitive to temperature changes below 40 or 30°C (dependent on what reference you are choosing, see comment to line 53)

We agree that the closure temperature range for AHe is between 40°C and 120°C following Ault et al. (2019). The thermal history modelling takes into account the present-day temperature of the rock. A thermal history can therefore be modelled between 40°C and surface temperature.

Line 83 Here, for the first time, the term "hanging wall" is mentioned in the text after the abstract, but the reader has no idea from your introduction, which side is the hanging wall, as you have not described the geometry of the fault in 3D. Thrust fault, normal fault, nothing was said about this.

To clarify, "normal" and "(north to the fault)" have been added (line 88). "(Fig. 1C)" after the Mont-Louis massif was added to specify its location on the map (line 89). "In the early Eocene, the balanced cross-sections of Ternois et al. (2019) suggest the Eocene thrusting of Aston-Mont-Louis unit onto the Canigou massif, in agreement with thermochronological data (Maurel et al. 2008)" has been added at the beginning of section 2.1 to specify past motion of the fault (lines 68 to 70).

Line 85 You may delete the word "increasing", as you talk about a spectrum.

Line 86 "Pyrenean" instead of "pyrenean"

Lines 86/87 "This is consistent with fission track and zircon (U-Th)/He ages …"

Line 90 Suggestion: "of the Carança and Canigou massifs (Fig. 1C) in the …"

Line 99 Not clear what you mean by "with a differential estimated at …". Do you refer to a "difference in total exhumation"? Please, clarify!

All done (lines 90 to 108).

Line 100 The statement "was mainly exhumed, and rapidly cooled," is surprising. What we measure with thermochronology is "cooling". And this cooling can be interpreted in terms of exhumation. Here, it sounds as if it would be the other way around (which could be the case if exhumation is modelled and the corresponding cooling due to erosion estimated by the model). I would suggest to refer to the original citations in the quoted papers.

This sentence has been clarified: "Thermal modelling using low temperature thermochronometers revealed that the Canigou massif was rapidly cooled ~20°C/Ma between 30–25 Ma (Fig. 2) in relation with the Têt fault normal activity (Maurel et al., 2008; Milesi et al., 2019)" (lines 107 to 110).

Line 100 "in relation with the Têt fault"

Done (line 109).

Line 102 It is not clear what you mean by "from a large consensus". It sounds as if the community has debated on this, which would mean that papers could be cited.

"from a large consensus" has been removed and "Maurel et al., (2008)" citation has been added (line 110).

Line 102 "but the last period"

Done (lines 110-111).

Line 105/106 The statement "It must be stressed that some authors consider that any vertical displacement occurred …. First of all, you only quote one source for this (Petit and Mouthereau, 2012), which looks strange if you state "some authors". Then "any vertical displacement occurred" is not clear. Suggestion: "Petit and Mouthereau (2012) suggest that vertical displacement occurred during …"

We have changed by "Petit and Mouthereau (2012) consider that any vertical displacement on the Têt fault occurred during the Plio-Quaternary" (lines 114-115).

Line 110 Suggestion: "on the basis of multi-disciplinary (Gunnell et al., 2009) and thermochronological studies (Fitzgerald et al. 1999)."

Done (line 118).

Line 111 Suggestion: "Today, the Têt fault shows no evidence of …"

The word "clear" has been left because there are evidences of seismic activity around the Têt fault without enough precise location to say if it is or not related to the Têt fault zone (Souriau and Pauchet, 1998).

Line 113 "suggests incision rates in the range of 1 to 25 m/m.y. …"

Done (lines 122-123).

Line 116 "surrounded" instead of "characterized"?

The core zone is not surrounded by, but formed of breccias, cataclasite and gouges (i.e. fault rocks, which define fault cores, e.g. Caine et al., 1996). Consequently, we have replaced "characterized by" by "formed" (line 127).

Line 121 You should explain what you mean by "a multi-core pattern". Is this the same as "a hot spring cluster"?

"a multi-core pattern" is a fault zone composed of smaller cataclastic fault cores, alternating with fractured gneiss, referred to as lenses when the fault cores are linked. This sentence has been clarified by adding a short sentence referring to a multi-core fault zone: "with small cataclastic fault zone and deformed gneiss lenses within the CZ" (lines 130 to 131).

Line 125 The term "half-thickness" is not clear. Do you mean the distance from fault to outer rim of the DZ? It would be important to know, whether you refer to a horizontal distance of to normal distance to the fault. Depending on the geometry of the fault plane and the local topography, this might cause significant differences between horizontal and normal distance to the fault.

The term "half-thickness" defines the thickness of the DZ in one block of the fault (footwall). We have clarified this in the sentence by adding "the half-thickness of the DZ is estimated at 400 m (horizontal distance)" (lines 134 to 135) but also later in the manuscript with the sentence: "The distance from the Têt fault is a horizontal distance throughout the manuscript." (lines 207-208) for the reasons explained above in the point number 1. We also have added the "distance normal to the fault plane" in the Table 1 and Table 2.

Line 130 The term "fault rocks" comes unexpectedly. So far, we were talking about a fault and surrounding rocks. Please, explain or rephrase.

We have clarified that "fault rocks" (see comment line 116 above) are highly crushed or sheared rocks (breccias, cataclasites and gouges) by tectonic deformation processes, and localized into fault zones. They define the fault Core Zone (CZ) dimension (e.g. Fossen et al., 2015).

Line 141 Suggestion: "may be related to the occurrence of impermeable metasediments in the hanging wall at the surface"

Done (lines 149-150).

Line 143 The here cited "Prats fault near St-Thomas" is not shown in figures 1 and 3. So, how important is it? Note also that you commonly use a hyphen between "St" and "Thomas".

The Prats fault is considered as a subsidiary fault (Fig. 3). The Prats fault has been added in figure 3. These subsidiary faults are not shown in Figures 1 because they are not considered as major structures at a regional scale. However, at the local scale such faults can impact the fluid circulations.

Line 144 Do you mean "can also increase permeability and localise channelized fluid upflow" Or: "can also increase permeability and localized upflow of channelized fluid"?

We have modified with "can also increase permeability and localise channelized fluid upflow" (lines 152 to 153).

Line 144 Figure 3: "Têt Fault" or "Têt fault" (see e.g. "Py fault" in this figure and writing in figure 1)? Should you use same colours as in figure 1? What is the meaning of the green zone in the hanging wall? What is the meaning of "CMNC"?

Capital letter has been deleted at the beginning of the word "fault". "CMNC" has also been deleted, the kinematics of this mylonitic zone is not the scope of this paper. We used the same colours than in Figure 1 for lithologies as specified in the figure caption.

Line 147 Suggestion: "Figure 4 shows the numerical modelling results of …"
Line 148 ", which takes…"
Line 148 "discontinuities, but does not …"

All done (lines 157-158).

Line 153 "Gallinàs peak" is shown in figure 4, but also in figure 5 as "Puig Gallinàs" with a noted elevation of 2624 m…

It was a mistake. The altitude and the name have been changed line 165 and in the Figure 4.

Line 159 Figure 4: The colours in the scale to the right end with dark blue, but the figure also shows patches where the dark blue changes to white. What do the white patches mean? In addition, the

figure shows areas surrounded by a thin black line and it is not clear how you defined the black line. What does the black line separate?

This figure is a zoom of regional fluid circulation model from Taillefer et al. (2018). Clear patches are graphic effects due to the topographic shadow of the DEM. The significance of the black line around thermal anomaly has been clarified: "The dotted black lines correspond to the limit of the modelled surface temperature anomaly (modified after Taillefer et al., 2018)" (lines 173-174).

Line 159 Figure caption: I do not understand the numbers at the ned of the line. I assume that these numbers should be "7.1-15 m2 and 5.1-15 m2". Note that I only refer to one digit after the comma, or is the second relevant? Presumably not as the range given her is much larger…

Numbers have been corrected: $7.10^{-15}$ m2 and $5.10^{-15}$ (line 171). The permeability of the fault takes into account the displacement on the fault (line 172).

Line 154 "At this locality" instead of "In the latter location"

Line 155 The statement here seems to be in contradiction to what is said in line 149. Please, clarify.

Line 157 Suggestion: "suggesting that this area is a recharge zone". It does not make sense to state that you "suggest" that something is "interpreted"…

Line 161 "Thuès-les-Bains" instead of "Thuès"

Line 162 Suggestion: "with respect to sample position."

Line 168 Suggestion: "The here presented thermochronological study extends the study area along the footwall Têt fault to the west, up to the potential Planès recharge zone."

Line 170 "in a way different from numerical modelling."

Line 170 "We also incorporated samples from the hanging wall for comparison."

Line 174 "Palaeozoic"

Line 175 "providing" instead of "representing"

Line 178 Suggestion: "the Carança valley (CAR), at the foot of Gallinàs peak (GAL profile) and further west in the…"

Line 179 "which is a topographic high"

All done (lines 165 to 194).

Line 182 "we selected the freshest rocks". Is this a good strategy to find the most altered apatites, the strongest impact of geothermal activity, the strongest geochemical changes? I could understand that you would state that you had to obtain samples that you could carry along and that you could make thin sections from.

This strategy was followed in order to obtain apatite grains suitable for U, Th and He analyses. Indeed, in the most altered parts, apatite grains are highly fractured, highly coated with oxides and frequently inclusion-rich, thus not suitable for analyses. Even if we agree that we may have missed the strongest impact, we carefully collected in different places of the damage zone in order to estimate the various impacts and our results confirmed our choice. We added also a sentence and a reference to describe hydrothermal alteration in our samples "Figure 6 shows that the alteration and fluid circulation in the DZ are localized along fractures, are very heterogeneous at the scale of the rock sample, and generally weather limited volumes of poorly permeable rocks around fractures (e.g. McCay et al., 2019)" (lines 197 to 200).

Line 185 "Thuès-les-Bains"

Line 191 "crossing the St-Thomas hot spring cluster"

Lines 191/192 "The hot springs at St-Thomas …"

All done (lines 202 to 210).

Line 192 Reference to figure 4?

We now refer to Figure 3 (line 210).

Lines 193/194 Suggestion: "The dimension of the DZ in St-Thomas with a width of 700 m is larger than in …of this fault network."

Done (lines 210-211).

Line 195 The here quoted samples and their elevation are samples from the footwall.

We have modified the sentence "We collected 6 samples in the footwall along this profile including gneisses, granites and mylonites (Figs. 6 A, B and C), with elevations between 1107 m (ST16) and 1480 m (ST8)" (lines 212-213).

Line 197 "around the St-Thomas hot springs"

Line 197 "at 250 and 400 m distance"

Done (line 216).

Line 200 "distance" of 835 and 1750 m normal to the Têt fault." See also comment to line 125. Clarify what distance you mean exactly.

Lines 201/202 "are collected at 5, 160, 175, and 1900 m normal distance from the Têt fault, respectively, and…". See also previous comment.

We have clarified this in the manuscript with the sentence: "The distance from the Têt fault is a horizontal distance throughout the manuscript" in lines 207-208, see the comment above. We have also added the "normal distance to the fault core surface" in the Table 1 and Table 2.

Line 203 delete "also"

Done.

Line 206 Figure 5: The weak point of this figure is that it does not fit to figure 15 with respect to the fault lines. One example: In figure 5, a prominent fault line passes just south of Puig Gallinàs, while in figure 4, the peak is shown N of the Prats fault, and in figure 15, it seems that the fault passes north of the peak. It is also interesting to note that some NW-SE Palaeozoic faults are offset along the Têt fault dextrally (e.g. near CAR7), but some go across it (e.g. next to St-Thomas). Knowing that the fault has been active in the Cenozoic, a fault passing through would mean that there was no lateral offset during this faulting. Furthermore, profile (3) should have two faults at its SE end, but figure B does not show these faults.

Although Figure 15 is a schematic view of the area (now indicated in the caption), it is right that it did not fit enough the precise map shown in Figure 5. Fault traces presented in Figure 15 have been carefully revised with respect to both Figure 5 and a new and more precise fault trace interpretation of the area done using spot 6/7 images, which is presented in a paper now under revision in Tectonophysics (Taillefer et al., 2020). This analysis reveals that the NW-SE faults are mainly crosscutting or abutting (and not clearly displaced dextrally) along the Tet fault. Since this fault population analysis is not the focus of the paper, here we just present the results and properly refer to the work published Taillefer et al. (2017). Figure 5a and b have also been corrected with both the kinematic marks in the map, and the missing faults and arrows on the profiles.

Line 206 Figure caption: "showing four sampling profiles" (as in figure B, they are called "profiles". Suggestion: "Grey colours show samples from previous…"

Line 211 The title should be "Areas with no hot springs: …" The same would apply to the titles of chapters 4.1.2 (line 315) and 5.1.2 (line 439).

Line 212 Suggestion: "The GAL profile has been placed across the thermal anomaly…"

Lines 212/213 In my print, the reference "Taillefer et al., 2018" has a blue font. Why?

Line 213 Start sentence with "The samples GAL3, …"

Line 215 delete "sample" after "GAL3"

Line 216 "the same lithology as GAL3"

Line 218 "from the Tête fault, respectively (Fig. 5)." I would avoid to use the term "core", as you have not defined such a "core". Or do you mean the DZ instead?

Line 219 "from the inner DZ"

Lines 221/222 Suggestion: "near the valley bottom at Carança" (if there is a place with this name).

Line 224 The term "retrogression" is commonly used in terms of marine retrogression. I think what you mean is "alteration" (or retrograde metamorphic overprint?).

Line 225 Suggestion: "fractures at the outcrop scale"

Line 226 Suggestion: "In the PLA profile, two dated samples are located …". Since you refer to an elevation of 1500 m, there are only two samples

Line 228 "fractured with local presence of …"

Line 228 The reference to figure 5 is probably wrong. How can we see iron oxides in figure 5?

All done (see lines 225 to 244).

Line 233 Figure caption: I am missing a general intro such as "Outcrop and microscopic images of crystalline rocks near Têt fault, eastern Pyrenees".

We have added a general sentence to introduce Figure 6: "Outcrop and microscopic images of crystalline rocks near the Têt fault (eastern Pyrenees) showing brittle deformation gradient. Outcrop and microscopic images of crystalline rocks near the Têt fault (eastern Pyrenees) showing brittle deformation gradient (lines 250-251).

Line 234 In figure D, I see mainly dark fillings (chlorite?) for the fractures. Do you mean "silicate fillings or do you refer to quartz?

The content of these fractures is colloidal silica, based on field criteria (chlorite does not striate the hammer), but it is also possibly composed of quartz, similar to quartz veins observed in thin-section view in Figure 6E, that may undergone strike slip. We do not have thin-sections of these veins on Figure 6D so we preferred to mention silica for the description.

Line 235 Figure E: Again use "quartz" instead of silica", as the optical properties clearly indicate the presence of quartz. According to text, this should be sample CAR7, if yes, add this information to all figure parts.

We now refer to quartz for sample CAR7.

Line 237 "fractured" instead of "fracturated"

Lines 238/239 "(enlargement of dashed rectangle in figure H)"

Line 243 Suggestion: "mineral concentrates were gained by…"

Line 243 "with no evidence"

Line 244 "for apatite grain photos"

Line 245 Start sentence with "Each single grain was packed in …"

Line 247 "achieved by" instead of "with"

Line 253 add comma after "procedure"

Line 255 "were analysed in-between four …"

Line 263 Does "mn" means "minute"? If so, better use "min"

Lines 268/269 Here, the abbreviations LREE and HREE are introduced. You should then use them systematically (see further comments below).

Lines 271/272 I am not sure whether this sentence should be included. If yes, start with "Caused by technical problems, some apatites have not…"

Line 274 "whether" instead of "if"

Line 276 Do you mean "modelled" instead of "modelised"?

Lines 280-288 In this paragraph I am missing a description of the AHe ages…

Line 281 "is observed" instead of "can be evidenced"

All done (see lines 252 to 300).

Lines 282/283 "a marked EU anomaly (Fig. 8)

We did the grammatical correction, but we left Europium to not confuse with eU values (line 301).

Lines283/284 Suggestion: "… 1999). For fine-grained gneiss samples of the GAL profile, this anomaly is less pronounced."

Line 284 "apatites of GAL3 have…"

Line 284 "consistently" instead of "consistent"

Line 285 "and flatter REE patterns"

Line 286 Suggestion: "are attributed to co-genetic growth of epidote, …"

Line 287 "in the GAL samples" rather than "in our samples", this would be more specific.

All done (lines 302 to 305).

Line 288 The term "at different scales" is difficult to understand. What scales are you referring to?

"at different scale" has been deleted.

Line 289 Figure 7: In profile GAL, there seems to be a difference with the distances from the Têt fault for the samples of the hanging wall with respect to figure 5. The labels and numbers along the axes and the font in the legends are on the edge of readability. One may simplify by labelling the axis only once on the left side and bottom of the figure.

On the x-axis "horizontal distance from the Têt fault" has been added to clarify. The size of the letters has been enlarged for the readability of the figure.

Line 290 Figure caption: I think, what you define here for the grey area is the outer damage zone (as the inner is yellow). I would start the sentence with "An inner DZ (yellow)…"

This sentence has been rephrased "AHe ages as a function of the horizontal distance from the Têt fault. Grey area shows the Damage Zone of the Têt fault; an inner DZ (yellow) is only distinguished for the TET and GAL profiles according to Milesi et al. (2019)" (lines 306-307).

Line 291 "for the TET and GAL profiles"

Done (line 307).

Line 291 The reference to Milesi et al. (2019) is only half way correct. As written, the reader has to believe that Milesi et al. (2019) also contains samples from the GAL profile, which is not true.

See answer to comment line 290.

Line 293 "analysed" instead of "dated"

Line 295 "and are younger"

Lines 296/297 Suggestion: "… (Milesi et al., 2019). Age dispersion is evident for this sample."

Line 297 "of the now analysed grains"

Line 299 Suggestion: "age dispersion from … to …"

Line 304 Suggestion: "grains in each sample (ST15 and ST16, Table 1)."

Line 304 Start next sentence with "The gneiss samples …"

Line 306 add comma after "respectively"

Line 306 delete "comprised"

Line 307 Here, the reference should also be table "1" instead of "2".

All done (lines 309 to 320).

Lines 307/308 The statement "that could be indicative of the occurrence of optically undetected tiny Th-rich inclusions within these apatites" is weakly supported. It is a possible explanation, but you refer to this sample as being a paragneiss, and in paragneisses, the apatites could be of very heterogeneous origin. Thus, the explanation you give could be true for any of the paragneiss samples in your data set. In addition, the initial REE patterns of these apatites could be very different.

We agree on the fact that in a paragneiss, the origin of apatites can be multiple which can be responsible for variable chemical compositions. However, in this sample, the variability of Th content is much higher than in other samples and Th/U ratios vary from one order of magnitude between the different apatite populations of apatite in this sample (Table 1). Moreover, only the Th-rich grains from this sample also show an enrichment in LREE that is consistent with monazite or allanite inclusions. As these inclusions were not detected, we cannot ensure their existence, but we still prefer not to consider these grains in the rest of the study. In order to be less affirmative we rephrased the following sentence : "In the ST2 sample, two grains (ST2-5 and ST2-6, table 1) giving the youngest AHe ages of the sample (12.1 ± 0.7 Ma and 9.2 ± 0.6 Ma) show anomalously high Th contents, with high Th/U ratios compared to the other grains that can be indicative of the occurrence of optically undetected tiny Th-rich inclusions within these apatites (see supplement section S.5). By contrast with other apatite grains of this study, these Th-rich apatites are enriched in light REE (Fig. 8) consistent with the presence of tiny monazite or allanite inclusions which are common LREE-rich phases in granites (e.g. Förster, 1998). Therefore, these disturbed REE patterns are not taken into account in the discussion and the corresponding AHe ages are discarded" (lines 323 to 329).

Line 310 The statement "A mean age of … is retained for sample ST2." Is unusual. The statement is only given for this sample, while for no other sample. Delete sentence?

Line 315 Title should be "Areas with no hot springs: GAL and PLA profiles, Car valley" (see also comment to line 211).

Lines 316/317 Suggestion: "… are fine gneisses from outside the outer Têt fault DZ and show both limited AHe age dispersion with … and …, respectively"

Line 317 The "However" is not needed, there is no contrast to be mentioned.

Line 320 "HREE" instead of "heavy REE"

All done (lines 328 to 337).

Lines 321/322 I have serious doubts about the statement "geochemical and paragenetic conditions of growth during medium grade metamorphism of pelitic rocks". I have checked the study of Henrichs et al. (2018), and my impression is that they talk about new-grown apatites rather detrital ones. At least they refer to the fact that they see a change in REE mobility starting with the uppermost greenschist facies. We do not know, however, what the metamorphic grade of the here sampled gneisses is. Thus, I have two critical questions that should be answered in the paper, if you want to refer to Henrichs et al. study: First, what is the metamorphic overprint your gneisses (in particular the paragneisses) have undergone during Pyrenean metamorphism? Second, do we talk about detrital or new-grown apatites (and what does the Henrichs et al. study

The apatites are new-grown apatites formed during the barrovian variscan metamorphism at temperatures close to 600°C (Hoÿm de Marien et al., 2019). Detrital apatites are no longer present. The samples do not show evidence of a pyrenean metamorphic overprint that is mainly localized along the shear zones. There is no temperature estimate for this overprint but it does not exceed the range 350-400°C as along the Merens fault (Mc Caig and Miller, 1986; Mezger et al., 2018). We have rephrased "With the exception of one HREE enriched apatite in sample GAL7, apatite grains from the outer DZ and outside have the same REE patterns (Fig. 8), thus indicating similar geochemical and paragenetic conditions of growth during variscan barrovian metamorphism (Hoÿm de Marien et al., 2019)" (lines 336 to 339).

Line 323 I would start the paragraph with "Sample CAR7 from the …", because all CAR samples come from the Carança valley, so this information is redundant.
Line 327 "observed for apatites"
Line 329 "Further" instead of "Farther"
All done (lines 340 to 346).

Line 334 No comments to Figure 8 and its caption
Line 336 Table 1: The caption (or "Note" line below the table should explain the meaning of "Ft". I do not understand, why for some samples, you report mean ages, while for others, no mean age is given. What is the argument behind the selection? is given with three digits, while all other Ft values only with two digits. The Th/U value for TET1-12 is given as "<0.1" (see also "<0.1" for the age "Error"). According to the data for U and Th, the value should be 0.02, if one uses one more digit, similar to ST4-1, where you report a "Th/U" value of "0.0", and the ratio derived from the data would be 0.02 as well, if expressed with two digits after the comma. Since the number of digits is dependent on the data for U and Th, the number of digits may vary, and I would make a difference.

The mean age has been deleted for all samples (see answer of general comment number 5). Values in Table 1 and Table 2 have been corrected.

Line 343 The statement suggests that you believe that your samples are "S-type granites". Can you clarify this?

The sentence has been rephrased "Apatites from the hanging wall display basically REE patterns similar to those of the footwall suggesting similar protolith compositions." (lines 359-360).

Line 344 The subchapter number should be 4.2.1, and I would recommend to adapt the title to line 292.
Line 345 "In the TET profile, …"
Line 349 I would recommend to quote the age (14.8 ± 0.7 Ma).
Line 351 "In the ST profile, …"
Line 355 "mainly by lithology"
Line 358-361 I would argue that this paragraph is at the wrong place. Sample ML1 is closest to the GAL profile (Fig. 5).
Lines 360/361 Suggestion: from the same sample, and a mean AHe age from granite ST12 …"
Line 362 The subchapter number should be 4.2.2, and I would recommend to adapt the title to line 315.
Line 367 For completion, I would add a sentence such as "No REE patterns have been determined for these samples." (in reference to lines 271/272, I guess)
Line 368 Suggestion: "A single gneiss sample (PLA5) from ..:"

Line 370 Suggestion: "exhibits a REE pattern with constant LREE values (Fig. 8)

All done (lines 361 to 386).

Line 370 The statement "due to the presence of Th-rich inclusions" cannot be valid, as grain PLA5-4 has a very low Th value of 3.6 ppm. See also comment to lines 307/308).

It was a mistake; the sentence has been deleted.

Line 371 The bracket could be reduced to "(Table 2)", as the grain number is already mentioned two lines above.

Line 371 Suggestion: "Consequently, this apatite grain was discarded from further discussion."

Line 372 No comment to figure 9 and its caption.

Line 375 Table 2: Here again, you should clarify the argument on which you report mean ages for some samples and not for others. And you should explain the abbreviation "Ft" somewhere (either caption of "Note" line at the bottom).

Line 378 Suggestion: "…DZ, we combined these data with those from samples outside the …"

Line 379 "as a reference"

Line 379 It is not clear, what you mean by "accurate ages". "Accurate" with respect to what?

Line 379 "low" instead of "weak"

Line 380 "variation" instead of "variations"

Line 381 "S-type granite", no empty space after "S"

All done (lines 386 to 396).

Line 382 Again, the reference to Henrichs et al. (2018) seems to be weak. Is their statement that all apatites in paragneisses show the same REE patterns? At least, they note that a homogenization does depend on the metamorphic grade. And if the input of REE along with detrital apatite varies strongly at the beginning, the REE patterns must look differently, even if the apatites homogenize their REE in medium- to high-grade paragneisses. Furthermore, the amount of new epidote, which depends on the amount of Ca available in the paragneisses will also influence the redistribution of REE. Thus, my recommendation would be: Please, read the Henrichs et al. study carefully, and then decide, whether it is worthwhile citing it.

This part has been rephrased: "…except for the fine-grained gneiss sample GAL3 displaying a REE pattern usually found in apatites from paragneiss lithologies (Henrichs et al., 2018)" (lines 396-397).

Line 382 "We then define …"

Line 383 "for all other profiles"

Line 387 Figure 10: Several of the ellipses in these plots are so tiny that one cannot see the colour inside. Thus, I suggest to the authors to make the triangles bigger, they may even overlap, as the ST profile data are concentrated in the centre. .

Line 388 "… profile samples."

Lines 392/393 Suggestion: "…, we identify three fields that overlap partially:"

Line 396 "above 20 Ma, which are interpreted as …"

Line 400 Suggestion: "observed, more pronounced along …"

Line 401 "similarly" instead of "similar"

Lines 401-403 The statement "depleted REE patterns compared to apatites sampled outside the DZ" is not clear. Do you suggest that the apatites outside the DZ have undergone REE depletion? Aren't the apatites outside the DZ the reference material for the apatites within the DZ? The confusion for me gets worse, when you talk about "global depletion" in line 402. What do you mean by "global"? If all REE patterns are depleted, what reference do we have? I assume that there is a misunderstanding caused by the current text, which should be fixed by some re-writing.

Line 403 Suggestion: "Note, however, that along the ST profile, …"

Line 404 "consistent" instead of "consistently"

Line 405 "do not exhibit REE depletion"

Line 406 Suggestion: "Due to the small numbers of analyses, the result is not interpreted any further." Is this what you mean?

Lines 407/408 This is the first time that you refer to Flowers et al. (2009) and Gautheron et al. (2009) and state that the intra-and inter-sample age dispersion cannot be simulated. Here, the statement is

correct, as you leave the door open, that a simulation would work, if the single apatite grains would be modelled individually. There are, however, places in the text (see e.g. lines 503/504), where the wording has to be changed. Nevertheless, for this passage, I would recommend to change to "cannot be simulated by a common thermal history using existing diffusion models for apatite"

Lines 408/409 Suggestion: "In particular, the young AHe ages do not fit to the regional …" It is not the fact that there are young ages, but the spread in ages that does not allow a common thermal history to be modelled.

Lines 409-411 Suggestion: "In accordance with the results for samples in the outer DZ of the TET profile, they raise questions concerning the origin of the AHe age scattering near the hydrothermal circulation zones (Milesi et al., 2019)."

Line 411 "were subject to"

All done (line 397 to 426).

Line 412 Suggestion: "enhances 4He diffusion and apatite rejuvenation rather than U-Th incorporation, …"

We left the word "loss" here because it can be due to diffusion but also to the opening of apatite under hydrothermal conditions, probably concomitant to REE depletion. "parent supply" has been changed by "incorporation" (lines 427).

Lines 411/412 The statement "Which is inconsistent with the observed REE depletion" cannot be assessed, as there are some intermediate steps missing in our argumentation. The claim is; I guess, that REE depletion would run parallel to U and Th depletion. Is this true? And what is the argument for it? I do not doubt the argument, but is has to be stated.

We have added a sentence in order to more clearly state our argumentation : "Indeed, as U and Th have very close solubilities to REE, it is likely that the behaviour of these elements when apatites interacted with fluids was similar to that of REE (Cramer and Nesbitt, 1983; Gieré, 1990),…"(lines 428 to 430).

Line 415 "reaction zone" instead of "reacted zone"?
Line 416 "have been carried away"?
Line 416 "lattice of reacting apatite"
Line 418 "to be lost from the host apatite"

All done (lines 432 to 435).

Line 418 The statement "mainly by advection during hydrothermal alteration processes" Is not sufficiently precise. Within the apatite, the process is diffusion, outside of the apatite, it is advection.

We agree that we cannot state precisely which process is the main one accounting for He loss. We have removed "main" from the sentence.

Line 420 According to the reference list, it should be "Andrews and Lee, 1979"
Line 420 Start sentence with "Variable REE loss"
Line 422 "at thin-section scale"
Line 423 After "fluid interaction" you could add a reference to figure 6H.
Lines 423/424 Start sentence with "The chemistry and temperature of the fluid flow may change with time", then add a comma after bracket, and continue with "…, however, it is .."
Line 425 "prominent" instead of "important"
Lines 425/426 "hot springs than near St-Thomas (…)"
Line 426 "pronounced" rather than "important"
Line 427 "larger" instead of "greater"
Line 429 "We cannot exclude an impact"
Line 430 "at the St-Thomas areas"
Line 430 Suggestion: "which may be responsible for an increase of rock permeability"
Line 431 "heat dispersion into"
Line 432 observed within the"
Line 432 Suggestion: "This is independently supported by the …"
Line 433/434 One may add references to figure 4 after "Têt fault" in line 433 and to figure 5 after "Têt fault" in line 434.

Line 434 "that are all located adjacent to"
All done (lines 437 to 451).
Line 435 Figure 11: In the caption you refer to "blue" and "purple field"s. In my print-out, the "blue" and "purple" seem to be of the same colour. What about having the diagrams on the right with the same x-axis scale?
Now, it has been homogenized with purple colour for all the figure and common x scale for ST and TET graphs.
Line 435 Figure caption: "for the TET and ST profiles"
Line 436 "For the TET profile, "
Line 438 The "consistent" again raises the question of the reference. "consistent" to or with what? How about using "elevated" instead?
Line 438 "contents" instead of "patterns"
Line 438 "The depleting of REE is …"
Line 439 Better use title "Area with no hot springs: …" (see also comment to lines 315 and 362)
Line 440 Suggestion: "Slightly rejuvenated and scattered AHe ages and depleted REE patterns are also obtained for…"
Line 441 "hot spring clusters: Carança valley (CAR) and Planès profiles (PLA, Figs. 12 and 13A)
All done (lines 452 to 459).
Line 443 Again, some of the ellipses are very small, so that the colour inside is hard to see, one may shift the triangles so that they overlap with each other.
We have enlarged the triangles in Figure 12.
Line 444 "GAL profile samples"
Done (line 467).
Lines 447/448 Here, the wording has to be adapted to be more precise. It is not the fault of the diffusion models that you cannot simulate a common thermal history. For wording, see comment to lines 4087/408. The diffusion models work ok, if applied to one single grain, but they cannot come up with a common thermal history.
We have rephrased: "Yet again, AHe age dispersion cannot be simulated by a common thermal history regardless diffusion model for apatite…" (lines 463-464).
Line 449 "and young ones peculiarly": Here, again, it is not the young ages that cause the problem, but the spread of ages. The young grains have a correct thermal history and the old ones as well, but it is not the history of a closed system, but of in-growth or loss of He.
We deleted this sentence and added "suggesting opening of apatite system" (line 464).
Line 451 "no longer active"
Line 452 "circulation" instead of "circulations"
Line 456 Do you mean "the end of surface hydrothermal activity"?
Lines 456/457 Suggestion: "… recent, while a hidden geothermal system could still exist nowadays in the subsurface.
All done (lines 469-476).
Lines 458-461 Figure 13 and caption: Several details seem to be at odds with this figure: First, the start of the caption is equal to figure 11, but these are NOT the samples from "within the DZ" as stated in Line 459. Second, in lines 460/461, you state that the purple filed and triangles correspond to samples outside the DZ. Why is there no purple field for the GAL profile, three triangles for this diagram, but no triangles for the Car and PLA plot? Third, you mention in line 462 that there is an associated younging of AHe ages, but this is only visible for the CAR samples, not for the PLA and not for the GAL either. This caption has to be revised completely, I propose.
The caption has been thoroughly revised following this comment (see lines 486 to 490).
Line 466 "do not fit to the regional …"
Done (line 480).
Line 466 "depicted in Figure 2."
We have deleted "above" (line 480).

Line 467 The statement "which is also inconsistent with the numerical thermal modelling" is problematic. The modelling in figure 4 shows the situation as of today, but the AHe and REE data integrate over the entire geothermal activity period. The hydrothermal activity might have stopped at 13 Ma.

We agree. We have changed by "which is inconsistent with the present-day numerical thermal modelling (Taillefer et al., 2018)" (lines 480-481). We have also modified an other sentence of the paragraph to introduce that hydrothermal activity might have stopped earlier: "Alternatively, a more local hydrothermal system may have been active on the plateau independently of that of St-Thomas. The Planès area, such as the Carança valley, may also correspond to a blind geothermal system or a paleo-system. Both locations might then represent potential sites for future geothermal exploration" (lines 482 to 485).

Line 468 I am not sure, what you mean by "very circumscribed", do you mean "more local" or "more regional"?

We have changed "very circumscribed" by "more local" (line 482).

Line 469 "St-Thomas"

Line 470 "might then represent potential …"

Line 472 "we note" instead of "we can note"

Line 473 reference to figure 13 or 13A?

Lines 474/475 I would delete the first part of the sentence "This does not support … since 10 Ma", and connect the rest to the sentence before: "… within the DZ in the last 10 Ma, in contrast to what …"

Line 477 "circulation" instead of "circulations"

Line 477 "Moreover, thermal modelling …"

Line 479 "suggests rapid cooling"

Line 483 "supports" instead of "reinforces"

Line 485 "using" instead of "made with" and then "and" instead of "using"

Line 487 "suggests an" instead of "corresponds to an" as these represent no unique solutions.

Line 489 "even at thin-section scale"

Line 490 You are mentioning "He loss", but could it also be gain to explain the unusually old single grains next to the fault?

Line 490 I think what you mean is not "actual" but "present-day hot springs"

Line 492 Start sentence with "The combination of ..:"

Line 493 "extensions"

Lines 494/495 Suggestion: "This result questions a straight-forward interpretation of AHe ages, if potential …"

Line 496 Figure 15: Should the scale of intensity (top right) not be called "intensity of present-day thermal anomaly"? For the "outcrop scale": "Poorly fractured gneiss lenses"

Line 497 Figure caption: Start text with "Synthetic 3D block of …"

Line 497 Suggestion: "of the present-day thermal anomalies"

Line 499 "samples" instead of "sample" (as there are ML1 and ST12)

Line 499 add reference to Table 2 after "AHe ages"?

Line 501 "S-typ granites" instead of "granitic lithology"?

Line 502 "similar to" instead of "as in"

Line 502 "closest to" instead of "nearest"

All done (lines 483 to 522).

Line 503 Suggestion: "The AHe age scatter (Fig. 7, Table 2) cannot …" I am sure it should be figure 7 instead of figure 3…

Yes, it was a mistake. The number of the figure and the sentence have been changed.

Lines 503/504 Here again, the wording has to be adapted to the comments to line 407/408.

We have changed with "cannot be properly modelled with a common thermal history" (lines 523-524).

Line 505 "Apatite REE contents appear less …"

Done (line 524).

Lines 506/507 I do not understand the statement on the Canaveilles hot spring cluster in the hanging wall. The hydrothermal anomaly on the other side of the fault is much closer. You draw a picture of a completely impermeable fault plane, which for several clusters and profiles is shown to not be true. I would rather go for a picture, in which the fault plane is permeable, in particular along existing fractures systems that cross the Têt fault. By such a picture, it is very easy to also explain the Canaveilles hot spring cluster, which is located very close to a prominent fault linked to the Têt fault (Fig. 5). Of course, these neighbouring faults may have undergone some reactivation and local increase of permeability, even if surrounded by less permeable metasediments. Such lenses of metasediments (Fig. 5) may even help to channel and concentrate fluid flow.

We have rephrased: "This result can be due to the hydrothermal influence of the Thuès-les-Bains hot springs in the footwall (Fig. 5) and to the same process of $^4$He loss described in the footwall outer DZ (e.g. sample CAR7). This suggests that the fault is not an impermeable barrier for the fluids. We cannot exclude that fluid circulation in the hanging-wall played also a role as attested by the presence of the Canaveilles hot springs just to the North-East of the TET profile. In this specific location (TET profile, gneiss-gneiss contact, Fig. 3) both the hanging-wall topographic gradient and the presence of permeable rocks might have contributed to a hydrothermal cell into the hanging-wall." (lines 526 to 531).

Line 507 "just NE of …"

We have clarified "just to the North-East of the TET profile" (line 529).

Line 509 "those close to the Têt fault"

Line 511 Here again, I would suggest a rewording to better clarify what you mean by "can be properly modelled with He diffusion models in apatite". It is not the models that are wrong, but it is the fact that even within one sample fluid circulation may have led to very different thermal histories.

Line 512 "outside the DZ"

Line 513 "shows rapid cooling"

Line 514 According to figure 16, it should be "range of 150 to 50°C" (perhaps even 40°C).

Line 514 "slows" instead of "low"

Line 514 "Low" instead of "slow", as this time, you refer to "rates".

Line 514 Suggestion: "may account for" instead of "can account for"

Line 516 "The PLA5 sample" or "Sample PLA5"

Line 517 Suggestion: "lower than the geothermal system along the Têt fault footwall …"

All done (lines 529 to 540).

Lines 518/519 The last sentence suggests that everything can be explained by regional slow cooling. Does this mean that geothermal activity is no explanation for the scattering? Why not? See, e.g., hot springs in Canaveilles on the other side of the fault.

We have rephrased: "This peculiar location may account for the AHe age scattering and ageing within this sample outside the DZ, in addition to the more pronounced age dispersion compared to the footwall due to the above mentioned regional slow cooling" (lines 541 to 542).

Line 522 Suggestion: "corresponds to a perfect match."

We have rephrased with "an ideal fit' (lines 545-546).

Line 523 delete "mainly"

Line 525 "even though" instead of "despite"

Line 525 I think the statement that the small numbers do not allow "a straightforward interpretation" is not the issue. The small number allows many interpretations to be supported by the data. The low number of data is not sensitive enough to discredit many of them. So, my suggestion would be "does not allow a clear" or "unique interpretation".

Line 525 "questions" instead of "question"

All done (lines 547-549).

Line 527 Do you mean "all samples outside the DZ". If not, I do not understand the statement…

It was a mistake now corrected.

Line 527 "thermal history together with rocks ..:"

Line 528 "perturbation, in accordance with …"

Line 529 "that hydrothermal circulation is mainly restricted"

All done (lines 553 to 555).

Lines 529/530 I would disagree with this statement. First, several profiles show the influence of the geothermal activity on AHe ages also in the hanging wall. Furthermore, the Canaveilles hot spring cluster clearly indicates that there are hot springs on the other side. Even if you suggest a completely impermeable fault plane, AHe ages might be influenced by re-activation of fluids on the other side of the fault by the increase in temperature in the rocks of the footwall.

See comments for lines 506/507.

Line 531 "influenced" rather than "impacted"

Line 532 Start sentence with "We also highlight …", "in addition" and "also" are redundant.

Line 532 "interaction" instead of "interactions"

Line 533 The here quoted temperature values of "<130°C" has not been quoted in the paper. It is important? If yes, you have to explain, where this temperature has been derived from.

Line 533 "considered a closed system"

Line 534 "released into" instead of "dissolved". The transport within the fluid is not the critical process.

Lines 534-536 I would not make these statements here. First, you have not discussed the processes that caused He gain and loss and REE loss in this paper. The conclusions should not come up with new (not discussed) issues. Second, I would suggest that the processes could be addressed in a next paper, if you have evidence or modelling results to distinguish between potential processes. But in this paper, the focus clearly is on the hydrothermal activity and how to detect it; there is little evidence that was presented about the processes, certainly nothing about "nano-channels within the crystal lattice". So, consequently take these statements out.

Line 536 "As fluid flow through …"

Line 536 "even at thin-section scale"

Line 537 "Again, the "4He loss by fluid advection" is not the limiting process, but diffusion is much more crucial as Flowers et al. or Gautheron et al. have shown.

Line 537 The "global positive correlation" does not exist. It exists for some of the data sets (profiles), but by far not all of them.

Lines 538/539 Suggestion: Start sentence with "The same process may also account for …"

Line 539 "in other cases"

Line 539 "fluid circulation" instead of "fluid circulations"

Line 539 "may occur" instead of "took place", I would suggest to make this more general, not only relevant for the past.

Line 540 "questions" instead of "question"

Line 540 Suggestion: "AHe ages in cases, for which potential hydrothermal …"

Line 541 Do you mean "active" instead of "obvious"?

Line 542 "palaeo-geothermal" instead of "paleo- geothermal"

Line 543 Suggestion: "As an exploratory tool, (U-Th)/He thermochronology may be applied complementary to …"

Line 544 "geoelectrical" instead of "electrical"

Line 545 It is not clear, what "models" you are referring to.

As suggested, we have totally rewritten the conclusion see lines 558 to 588.

Line 549 "by Doriane"

Line 550 "during field work."

Line 552 Suggestion: Data supporting the findings of this study are available in the Supplementary section. Additional details may be obtained upon request …"

Line 556 "models on"

Line 557 "co-wrote"

Line 558 "participated in"

All done (lines 590 to 595).

References: I was unable to locate the reference Sutherland et al. (2012) in the text.
Missing in the list is the reference Farley et al. (1996), found in line 254.
The reference Shipton & Cowie is at the wrong place.
I found strange spellings, spaces or upper case words in lines 677, 683, 703, 742, and 771/772 that should be corrected.
All references have been checked.

---

## Author Comment (AC2) · 30 Jun 2020

**Point per point reply Reviewer Comments 2**

The ms presents interesting new data on two mains zones (free fluids and hot fluids zones). However, the authors have already published similar contribution in 2019, where they present the impact of hot fluid on AHe data. Only at the very end of the text, the authors show that when hot fluids are not present, it doesn't affect the AHe data. It is difficult to really understand what the message of the contribution is. Is that a methodological paper? If yes, the authors need to go further as the impact of hot fluids in fault affecting the AHe system has been already proposed. In addition, the authors present new thermal history modeling of the area, but they don't really discuss the implications for the eastern part of the Pyrenees. I strongly suggest that the authors propose further investigation on the exhumation of the eastern Pyrenees and compare with the other thermochronological data. Or the authors could investigate past geothermal anomalies and better present the new result of their contribution (AHe data are not affected in area where fluids are absent).

This paper, as explained in lines 183 to 186, is an extension of the study published in Milesi et al. (2019) allowing to better discuss at a larger scale the extent of the thermal anomalies based on AHe ages and REE patterns along four profiles across the Têt fault, in its footwall and hanging wall.

Abstract.
Add France after Pyrenees.
Done (line 10).
The passage with the second sentence of the paragraph is odd. I suggest "in order to investigate the evolution of the geothermal gradient and fluid flow, we used AHe + geochemical analysis to ...
The sentence has been changed (lines 11 to 13).
Paragraph 20: the authors gave sentences about exhumation for two zones, without using the data. What is the implication of the results? It is very unexploited.
As clearly stated now, the main goal of this paper deals with the effects of hydrothermal fluid circulations along a dormant fault on AHe ages. We use the cooling histories of both hanging wall and footwall determined far from the fault as references and do not intend to discuss the exhumation. (lines 19 to 23 and 25 to 26). This is beyond the topic of this paper and will be the topic of a new paper in progress.
Paragraph 25: it opens new perspective to what exactly? This could be a good angle to go to deeper investigation in the ms
The perspectives have been specified (line 25).
Paragraph 30: I am not sure to understand why the presence of water is required to heat production. Perhaps, heat advection need water, but water don't produce heat.
Hear we talk about the electricity production in well, referred to as "geothermal doublet". There the electricity production is a direct function of the water flow (discharge rate). This is now mentioned in lines 32 to 35 :" This type of system can show temperature and water discharge rate large enough for electricity production in "geothermal doublets", such as developed in the Basin and Range province (Blackwell et al., 2000; Faulds et al., 2010) or in western Anatolia, Turkey (Roche et al., 2018), with temperatures above 200°C (Bertani, 2012)."
Paragraph 50: about AHe temperature sensitivity range go higher than 90°C depending on the damage dose. I suggest adding 40-120°C and add Ault et al 2019 reference
Done (lines 53-54).
Paragraph 55: the goal of the paper "the study wants to test this tool both in areas lacking of hot fluids ...". Please rephrase better to see what is new in the study. Why above 60°C?
This sentence has been changed and completed (lines 55 to 61).

Paragraph 60: and in general, more recent citations on low temperature thermochronological data of the Pyrenees are missing. Please add other papers than Verges et al.1995.

Done. New references have been added: Gibson et al., 2007; Sinclair et al., 2005; Metcalf et al., 2009; Whitchurch et al. 2011; Fillon and van der Beek, 2012 (lines 65 to 67).

Paragraph 65: remove the ) between Later, ) two minor…

Done.

Figure 1: Homogenization of the scale bar for each figure (a, b, c) could be good. What is the white square on fig c?

Done.

paragraph 80, line 3: add Ma after the numbers 300.3_3.1 and 291.2_2.8 Reference of Ar/Ar ages are missing It is ZFT and not ZFt. Please replace

Done. Maurel et al. (2008) reference for Ar/Ar ages was added (lines 88 and 89).

Figure 2: give the elevation or difference of elevation between samples. The addition of the AHe observed / predicted versus elevation could be nice.

Done.

Paragraph 85: it should be Apatite (U-Th)/He age yielded a large range of age between and not apatite yielded a large range of AHe age. Add error on AHe ages

Done.

Paragraph 95: could also add error on Ar ages

Done (lines 101 to 102).

Paragraph 100: please correct, it is not the low temperature data that reveal that the Canigou massif was exhumed and cooled but thermal modeling. What do you refer by rapidly cooled? Add values, like that it will be more homogenous with other given exhumation rates (see Paragraph 110)

Done, cooling rate values have been added (lines 108 to 110).

Paragraph 110: add more recent references about uplift and erosion in the Pyrenees, e.g.:. Vacherat et al. 2014, 2016, Ternois et al., 2019 etc for example

Done (line 86 to 87).

Section 2.2: since when hot fluids are circulating in the tet fault?

In section 2.2, a sentence was added to discuss about the timing of the onset of hydrothermal loops (lines 168-169).

Figure 3: color legend is missing. What is the brown line at the bottom, in the footwall?

The colours of the lithologies have been modified in agreement with Figure. 1 and lithology is now indicated in the Figure. 3. The brown line at the bottom has been deleted.

Paragraph 145: please add a little more information on how the numerical modeling has been done and with which data

Some information has been added (see lines 157 to 171) but we cannot describe thoroughly the modelling, this is an other work already published (Taillefer et al., 2018) to which we refer.

Figure 4: could the authors add on the figure, the location of the samples? It can be useful I am not really sure about the purpose of this figure. What the authors wants to show?

The location of the sampling profiles has been added on this figure. The modelling and such a map helped us to locate the sampling profiles in "hot" and "cold" thermal anomalies areas along the fault.

Paragraph 180: the aim of this sampling… was to track the effects of recent hydrothermal circulation… but what is new in this study? The authors have already published this type of study, so it is important to go further.

In our previous work, we showed that actual hot (>60°C) water circulations have an effect on AHe ages. This work further shows that in areas devoid of present-day hot spring, apatite re-opening occurred in relation with recent or blind geothermal system thus affecting AHe ages that should therefore not be taken into account for regional thermal modelling. The re-opening of apatite grains can be traced thanks to combined REE analyses. We propose that combined U-Th/He and REE analyses is a useful tool to select grains devoid of hydrothermal imprint. The latter should be used

for apatite selection for thermal modelling. Moreover, this tool may be useful for exploration of blind geothermal systems. The conclusion has been rewritten to clearly state the original contributions of this work (see lines 558 to 588).

Paragraph 210: why is the reference Taillefer et al., 2018, in bleu and italic?

Done.

Paragraph 245: because the authors also measured the Sm content, they can use the value to add them and calculate the (U-Th-Sm)/He age

We cannot use Sm values, for different reasons. First, we did not measure the Sm for all apatite grains. Second, we did not use a solution spiked with Sm isotopes, therefore we cannot have the same accuracy than for U and Th measurement. Finally, adding Sm content would change AHe ages in a range of 0.02 Ma to 0.2 Ma, which is in the error values of (U-Th)/He ages.

Paragraph 260: the raw REE data should be add in the AHe data table for simplicity. Please add the raw values and not normalized to chondrite.

Raw REE data was added in the Table 1 and 2. Chondrite normalized REE data were replaced by measured values (in ppm) in the supplementary material too (Supplement section S5, Figure. 5).

Figure 7: it will be nice, if the authors add directly on the graphic, the free hot spring and with hotspring.

Done.

Paragraph 295: about eU (U+0.234xTh+0.0045xSm; Gastil et al., 1967), the presentation of the value could more simply presented 12 ppm instead of 12.3 ppm etc. What about the variation in the Th/U ratio? It can help to see if the Th and U have fractionated

Reference to Gastil et al., 1967 has been added line 313; Th/U ratio graphs are now showed in supplement section S.5. They do not help to discriminate whether the Th or U fractionated.

Figure 8 et 9: both figures are really difficult to read. They look very similar. Perhaps so diagram can go in the supplementary diagram and only light/heavy REE ratio could be presented.

These figures show very well AHe ages associated to the REE patterns for each apatite grain. As the combination of REE with AHe ages is very important in our work and as the measurements of REE represented a huge effort, we prefer to keep these figures in the manuscript.

Line 342 "Apatites from the hanging wall"

Done (line 357).

Paragraph 410-415: It will be interesting to add Cl measurement or compare the REE data from the apatite to discuss more about the dissolution / recrystallisation process. The reference Zeitler et al 2017 is good, but just is a summary of other studies (Shuster et al., 2006; Flowers et al., 2009; Gautheron et al., 2009; Gerin et al., 2017, Idelman et al., 2018, McDannell et al., 2018), that can be cited or for simplicity the reference Ault et al 2019 resumed all of other recent studies

We did not measure Cl content. At the regional scale, Cl values in apatite grains are available in Gunnell et al. (2009) and in our previous work (Milesi et al., 2019). These two studies show low Cl content in apatite in the DZ and outside the DZ, without important Cl intra-sample variability. This is why we did not measure it systematically. Ault et al. (2019) reference has been added in the discussion (lines 434-435).

Paragraph 525: the last line that describe the fact that if not fluids are shown, it doesn't impact the AHe ages can be the angle of the paper. This is the main new result of the paper and it appears at the end. It is a shame, because it is very interesting. The authors could present this results in the beginning, and not focus to much about the influence of hot fluids, as it has been already published.

We do not agree with this comment. On the contrary, the main result is that AHe ages have been impacted in areas without present-day fluid circulations (location CAR and PLA). This suggest that in areas where past or blind hydrothermal circulations are suspected, the apatite grains may have an hydrothermal imprint that accounts for the AHe age scattering. In order to overcome this issue, we propose that REE measurements can help to select non re-opened grains and exploration of blind hydrothermal systems.

Figure 11: are you sure about the sum of REE because 20000 ppm is 20%. It seems that it is chondritic normalized values. Please verify the value as 1000 ppm of Sm is quite a lot and if it is the case, it will strongly change the AHe age. What about AHe age vs Th/U, AHe age vs light/heavy REE ratio? It will be better to add also the data outside of the DZ rather than the purple square

Indeed, in the submitted version the values reported in Tables 1 and 2 were Chondritic normalized values. According to this comment, we have changed for REE content in ppm as we have changed in Table 1 and 2. Sm content ranges from 24.9 to 431.8 ppm in the footwall samples and in a range of 74.5 to 526.1 in the hanging wall samples. See comment paragraph 245 for the impact on AHe ages, and paragraph 295 for Th/U ratios. We have added the data from outside the DZ (see Figures 11 and 13).

Figure 13: same comment than for Fig 11

See comments for Fig.11.

Figure 15: scale of the microscopic scale seems not be correct as the apatite crystal are really too small, or the scale

The scale has been corrected.

Supplementary: please report the raw REE content in ppm

Done, see Table 2 in supplementary.

Table 2; you can add directly the REE measurement in the same table

The sums of REE have been added in Table 1 and Table 2.